# Sampling weights of deep neural networks

**Erik Lien Bolager**[+]    **Iryna Burak**[+]    **Chinmay Datar**[+†]    **Qing Sun**[+]    **Felix Dietrich**[+*]

Technical University of Munich

[+]School of Computation, Information and Technology; [†]Institute for Advanced Study

## Abstract

We introduce a probability distribution, combined with an efficient sampling algorithm, for weights and biases of fully-connected neural networks. In a supervised learning context, no iterative optimization or gradient computations of internal network parameters are needed to obtain a trained network. The sampling is based on the idea of random feature models. However, instead of a data-agnostic distribution, e.g., a normal distribution, we use both the input and the output training data to sample shallow and deep networks. We prove that sampled networks are universal approximators. For Barron functions, we show that the $L^2$-approximation error of sampled shallow networks decreases with the square root of the number of neurons. Our sampling scheme is invariant to rigid body transformations and scaling of the input data, which implies many popular pre-processing techniques are not required. In numerical experiments, we demonstrate that sampled networks achieve accuracy comparable to iteratively trained ones, but can be constructed orders of magnitude faster. Our test cases involve a classification benchmark from OpenML, sampling of neural operators to represent maps in function spaces, and transfer learning using well-known architectures.

## 1   Introduction

Training deep neural networks involves finding all weights and biases. Typically, iterative, gradient-based methods are employed to solve this high-dimensional optimization problem. Randomly sampling all weights and biases before the last, linear layer circumvents this optimization and results in much shorter training time. However, the drawback of this approach is that the probability distribution of the parameters must be chosen. Random projection networks [54] or extreme learning machines [30] involve weight distributions that are completely problem- and data-agnostic, e.g., a normal distribution. In this work, we introduce a data-driven sampling scheme to construct weights and biases close to gradients of the target function (cf. Figure 1). This idea provides a solution to three main challenges that have prevented randomly sampled networks to compete successfully against iterative training in the setting of supervised learning: deep networks, accuracy, and interpretability.

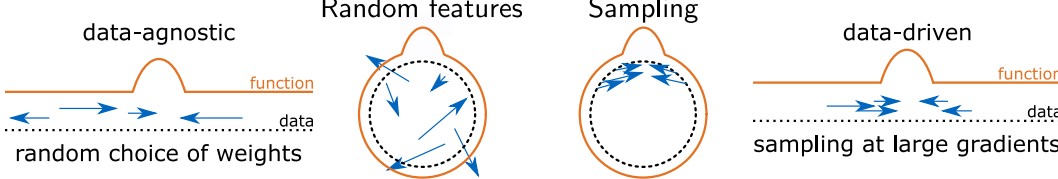

Figure 1: Random feature models choose weights in a data-agnostic way, compared to sampling them where it matters: at large gradients. The arrows illustrate where the network weights are placed.

---

[*]Corresponding author, `felix.dietrich@tum.de`.

37th Conference on Neural Information Processing Systems (NeurIPS 2023).

**Deep neural networks.** Random feature models and extreme learning machines are typically defined for networks with a single hidden layer. Our sampling scheme accounts for the high-dimensional ambient space that is introduced after this layer, and thus deep networks can be constructed efficiently.

**Approximation accuracy.** Gradient-based, iterative approximation can find accurate solutions with a relatively small number of neurons. Randomly sampling weights using a data-agnostic distribution often requires thousands of neurons to compete. Our sampling scheme takes into account the given training data points and function values, leading to accurate and width-efficient approximations. The distribution also leads to invariance to orthogonal transformations and scaling of the input data, which makes many common pre-processing techniques redundant.

**Interpretability.** Sampling weights and biases completely randomly, i.e., without taking into account the given data, leads to networks that do not incorporate any information about the given problem. We analyze and extend a recently introduced weight construction technique [27] that uses the direction between pairs of data points to construct individual weights and biases. In addition, we propose a sampling distribution over these data pairs that leads to efficient use of weights; cf. Figure 1.

## 2 Related work

**Regarding random Fourier features,** Li et al. [41] and Liu et al. [43] review and unify theory and algorithms of this approach. Random features have been used to approximate input-output maps in Banach spaces [50] and solve partial differential equations [16, 48, 10]. Gallicchio and Scardapane [28] provide a review of deep random feature models, and discuss autoencoders and reservoir computing (resp. echo-state networks [34]). The latter are randomly sampled, recurrent networks to model dynamical systems [6]. **Regarding construction of features,** Monte Carlo approximation of data-dependent parameter distributions is used towards faster kernel approximation [1, 59, 47]. Our work differs in that we do not start with a kernel and decompose it into random features, but we start with a practical and interpretable construction of random features and then discuss their approximation properties. This may also help to construct activation functions similar to collocation [62]. Fiedler et al. [23] and Fornasier et al. [24] prove that for given, comparatively small networks with one hidden layer, all weights and biases can be recovered exactly by evaluating the network at specific points in the input space. The work of Spek et al. [60] showed a certain duality between weight spaces and data spaces, albeit in a purely theoretical setting. Recent work from Bollt [7] analyzes individual weights in networks by visualizing the placement of ReLU activation functions in space. **Regarding approximation errors and convergence rates of networks,** Barron spaces are very useful [2, 20], also to study regularization techniques, esp. Tikohnov and Total Variation [40]. A lot of work [54, 19, 17, 57, 65] surrounds the approximation rate of $\mathcal{O}(m^{-1/2})$ for neural networks with one hidden layer of width $m$, originally proved by Barron [2]. The rate, but not the constant, is independent of the input space dimension. This implies that neural networks can mitigate the curse of dimensionality, as opposed to many approximation methods with fixed, non-trained basis functions [14], including random feature models with data-agnostic probability distributions. The convergence rates of over-parameterized networks with one hidden layer is considered in [19], with a comparison to the Monte Carlo approximation. In our work, we prove the same convergence rate for our networks. **Regarding deep networks,** E and Wojtowytsch [17, 18] discuss simple examples that are not Barron functions, i.e., cannot be represented by shallow networks. Shallow [15] and deep random feature networks [29] have also been analyzed regarding classification accuracy. **Regarding different sampling techniques,** Bayesian neural networks are prominent examples [49, 5, 26, 61]. The goal is to learn a good posterior distribution and ultimately express uncertainty around both weights and the output of the network. These methods are computationally often on par with or worse than iterative optimization. In this work, we directly relate data points and weights, while Bayesian neural networks mostly employ distributions only over the weights. Generative modeling has been proposed as a way to sample weights from existing, trained networks [56, 51]. It may be interesting to consider our sampled weights as training set in this context. In the lottery ticket hypothesis [25, 8], "winning" subnetworks are often not trained, but selected from a randomly initialized starting network, which is similar to our approach. Still, the score computation during selection requires gradient updates. Most relevant to our work is the weight construction method by Galaris et al. [27], who proposed to use pairs of data points to construct weights. Their primary goal was to randomly sample weights that capture low-dimensional structures. No further analysis was provided, and only a uniform distribution over the data pairs was used. We expand and analyze their setting here.

# 3 Mathematical framework

We introduce sampled networks, which are neural networks where each pair of weight and bias of all hidden layers is completely determined by two points from the input space. This duality between weights and data has been shown theoretically [60], here, we provide an explicit relation. The weights are constructed using the difference between the two points, and the bias is the inner product between the weight and one of the two points. After all hidden layers are constructed, we must only solve an optimization problem for the coefficients of a linear layer, mapping the output from the last hidden layer to the final output. We start to formalize this construction by introducing some notation.

Let $\mathcal{X} \subseteq \mathbb{R}^D$ be the input space with $\|\cdot\|$ being the Euclidean norm with inner product $\langle \cdot, \cdot \rangle$. Further, let $\Phi$ be a neural network with $L$ hidden layers, parameters $\{W_l, b_l\}_{l=1}^{L+1}$, and activation function $\phi : \mathbb{R} \to \mathbb{R}$. For $x \in \mathcal{X}$, we write $\Phi^{(l)}(x) = \phi(W_l \Phi^{(l-1)}(x) - b_l)$ as the output of the $l$th layer, with $\Phi^{(0)}(x) = x$. The two activation functions we focus on are the rectified linear unit (ReLU), $\phi(x) = \max\{x, 0\}$, and the hyperbolic tangent (tanh). We set $N_l$ to be the number of neurons in the $l$th layer, with $N_0 = D$ and $N_{L+1}$ as the output dimension. We write $w_{l,i}$ for the $i$th row of $W_l$ and $b_{l,i}$ for the $i$th entry of $b_l$. Building upon work of Galaris et al. [27], we now introduce sampled networks. The probability distribution to sample pairs of data points is arbitrary here, but we will refine it in Definition 2. We use $\mathcal{L}$ to denote the loss of our network we would like to minimize.

**Definition 1.** *Let $\Phi$ be a neural network with $L$ hidden layers. For $l = 1, \ldots, L$, let $\left(x_{0,i}^{(1)}, x_{0,i}^{(2)}\right)_{i=1}^{N_l}$ be pairs of points sampled over $\mathcal{X} \times \mathcal{X}$. We say $\Phi$ is a sampled network if the weights and biases of every layer $l = 1, 2, \ldots, L$ and neurons $i = 1, 2, \ldots, N_l$, are of the form*

$$w_{l,i} = s_1 \frac{x_{l-1,i}^{(2)} - x_{l-1,i}^{(1)}}{\|x_{l-1,i}^{(2)} - x_{l-1,i}^{(1)}\|^2}, \quad b_{l,i} = \langle w_{l,i}, x_{l-1,i}^{(1)} \rangle + s_2, \tag{1}$$

*where $s_1, s_2 \in \mathbb{R}$ are constants, $x_{l-1,i}^{(j)} = \Phi^{(l-1)}(x_{0,i}^{(j)})$ for $j = 1, 2$, and $x_{l-1,i}^{(1)} \neq x_{l-1,i}^{(2)}$. The last set of weights and biases are $W_{L+1}, b_{L+1} = \arg \min \mathcal{L}(W_{L+1} \Phi^{(L)}(\cdot) - b_{L+1})$.*

The constants $s_1, s_2$ are used to fix what values the activation function takes on when it is applied to the points $x^{(1)}, x^{(2)}$; cf. Figure 2. For ReLU, we set $s_1 = 1$ and $s_2 = 0$, so that $\phi\left(x^{(1)}\right) = 0$ and $\phi\left(x^{(2)}\right) = 1$. For tanh, we set $s_1 = 2s_2$ and $s_2 = \ln(3)/2$, which implies $\phi\left(x^{(1)}\right) = 1/2$ and $\phi\left(x^{(2)}\right) = -1/2$, respectively, and $\phi\left(1/2\left(x^{(1)} + x^{(2)}\right)\right) = 0$. This means that in a regression problem with ReLU, we linearly interpolate values between the two points. For classification, the tanh construction introduces a boundary if $x^{(1)}$ belongs to a different class than $x^{(2)}$. We will use this idea later to define a useful distribution over pairs of points (cf. Definition 2).

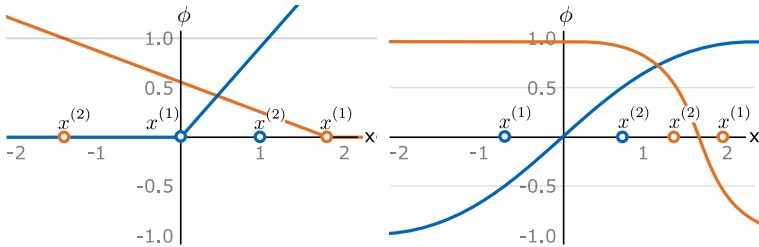

Figure 2: Placement of the point pairs $x^{(1)}, x^{(2)}$ for activation functions ReLU (left) and tanh (right). Two data pairs are chosen in each subfigure, resulting in two activation functions on each data domain.

The space of functions that sampled networks can approximate is not immediately clear. First, we are only using points in the input space to construct both the weights and the biases, instead of letting them take on any value. Second, there is a dependence between the bias and the direction of the weight. Third, for deep networks, the sampling space changes after each layer. These apparent restrictions require investigation into which functions we can approximate. We assume that the input space in Theorem 1 and Theorem 2 extends into its ambient space $\mathbb{R}^D$ as follows. Let $\mathcal{X}'$ be any compact subset of $\mathbb{R}^D$ with finite reach $\tau > 0$. Informally, such a set has a boundary that does not change too quickly [12]. We then set the input space $\mathcal{X}$ to be the space of points including $\mathcal{X}'$

and those that are at most $\epsilon_I$ away from $\mathcal{X}'$, given the canonical distance function in $\mathbb{R}^D$, where $0 < \epsilon_I < \tau$. In Theorem 1, we also consider $L$ layers to show that the construction of weights in deep networks does not destroy the space of functions that networks with one hidden layer can approximate, even though we alter the space of weights we can construct when $L > 1$.

**Theorem 1.** *For any number of layers $L \in \mathbb{N}_{>0}$, the space of sampled networks with $L$ hidden layers, $\Phi \colon \mathcal{X} \to \mathbb{R}^{N_L+1}$, with activation function ReLU, is dense in $C(\mathcal{X}, \mathbb{R}^{N_L+1})$.*

*Sketch of proof:* We show that the possible biases that sampled networks can produce are all we need inside a neuron, and the rest can be added in the last linear part and with additional neurons. We then show that we can approximate any neural network with one hidden layer with at most $6 \cdot N_1$ neurons — which is not much, considering the cost of sampling versus backpropagation. We then show that we can construct weights so that we preserve the information of $\mathcal{X}$ through the first $L-1$ layers, and then we use the arbitrary width result applied to the last hidden layer. The full proof can be found in Appendix A.1. We also prove a similar theorem for networks with tanh activation function and one hidden layer. The proof differs fundamentally, because tanh is not positive homogeneous.

We now show existence of sampled networks for which the $L^2$ approximation error of Barron functions is bounded (cf. Theorem 2). We later demonstrate that we can actually construct such networks (cf. Section 4.1). The Barron space [2, 20] is defined as

$$\mathcal{B} = \{f \colon f(x) = \int_\Omega w_2 \phi(\langle w_1, x \rangle - b) d\mu(b, w_1, w_2) \text{ and } \|f\|_{\mathcal{B}} < \infty\}$$

with $\phi$ being the ReLU function, $\Omega = \mathbb{R} \times \mathbb{R}^D \times \mathbb{R}$, and $\mu$ being a probability measure over $\Omega$. The Barron norm is defined as $\|f\|_{\mathcal{B}} = \inf_\mu \max_{(b, w_1, w_2) \in \mathrm{supp}(\mu)} \{|w_2|(\|w_1\|_1 + |b|)\}$.

**Theorem 2.** *Let $f \in \mathcal{B}$ and $\mathcal{X} = [0, 1]^D$. For any $N_1 \in \mathbb{N}_{>0}$, $\epsilon > 0$, and an arbitrary probability measure $\pi$, there exist sampled networks $\Phi$ with one hidden layer, $N_1$ neurons, and ReLU activation function, such that*

$$\|f - \Phi\|_2^2 = \int_\mathcal{X} |f(x) - \Phi(x)|^2 d\pi(x) < \frac{(3 + \epsilon)\|f\|_{\mathcal{B}}^2}{N_1}.$$

*Sketch of proof:* It quickly follows from the results of E et al. [20], which showed it for regular neural networks, and Theorem 1. The full proof can be found in Appendix A.2.

Up until now, we have been concerned with the space of sampling networks, but not with the distribution of the parameters. We found that putting emphasis on points that are close and differ a lot with respect to the output of the true function works well. As we want to sample layers sequentially, and neurons in each layer independently from each other, we define a layer-wise conditional definition underneath. The norms $\|\cdot\|_{\mathcal{X}_{l-1}}$ and $\|\cdot\|_\mathcal{Y}$, that defines the following densities, are arbitrary over their respective space, denoted by the subscript.

**Definition 2.** *Let $f \colon \mathbb{R}^D \to \mathbb{R}^{N_L+1}$ be Lipschitz-continuous and set $\mathcal{Y} = f(\mathcal{X})$. For any $l \in \{1, 2, \ldots, L\}$, setting $\epsilon = 0$ when $l = 1$ and otherwise $\epsilon > 0$, we define*

$$q_l^\epsilon\left(x_0^{(1)}, x_0^{(2)} \mid \{W_j, b_j\}_{j=1}^{l-1}\right) = \begin{cases} \dfrac{\|f(x_0^{(2)}) - f(x_0^{(1)})\|_\mathcal{Y}}{\max\{\|x_{l-1}^{(2)} - x_{l-1}^{(1)}\|_{\mathcal{X}_{l-1}}, \epsilon\}}, & x_{l-1}^{(1)} \neq x_{l-1}^{(2)} \\ 0, & \text{otherwise}, \end{cases} \tag{2}$$

*where $x_0^{(1)}, x_0^{(2)} \in \mathcal{X}$, $x_{l-1}^{(1)} = \Phi^{(l-1)}(x_0^{(1)})$, and $x_{l-1}^{(2)} = \Phi^{(l-1)}(x_0^{(2)})$, with the network $\Phi^{(l-1)}$ parameterized by sampled $\{W_j, b_j\}_{j=1}^{l-1}$. Then, using $\lambda$ as the Lebesgue measure, we define the integration constant $C_l = \int_{\mathcal{X} \times \mathcal{X}} q_l^\epsilon d\lambda$. The density $p_l^\epsilon$ to sample pairs of points for weights and biases in layer $l$ is equal to $q_l^\epsilon / C_l$ if $C_l > 0$, and uniform over $\mathcal{X} \times \mathcal{X}$ otherwise.*

Note that here, a distribution over pair of points is equivalent to a distribution over weights and biases, and the additional $\epsilon$ is a regularization term. Now we can sample for each layer sequentially, starting with $l = 1$, using the conditional density $p_l^\epsilon$. This induces a probability distribution $P$ over the full parameter space, which, with the given regularity conditions on $\mathcal{X}$ and $f$, is a valid probability distribution. For a complete definition of $P$ and proof of validity, see Appendix A.3.

Using this distribution also comes with the benefit that sampling and training are not affected by rigid body transformations (affine transformation with orthogonal matrix) and scaling, as long as the true function $f$ is equivariant w.r.t. to the transformation. That is, if $H$ is such a transformation, we say $f$ is equivariant with respect to $H$, if there exists a scalar and rigid body transformation $H'$ such that $H'(f(x)) = f(H(x))$ for all $x \in \mathcal{X}$, and invariant if $H'$ is the identity function. We also assume that norms $\|\cdot\|_{\mathcal{Y}}$ and $\|\cdot\|_{\mathcal{X}_0}$ in Equation (2) are chosen such that orthogonal matrix multiplication is an isometry.

**Theorem 3.** *Let $f$ be the target function and equivariant w.r.t. to a scalar and rigid body transformation $H$. If we have two sampled networks, $\Phi, \hat{\Phi}$, with the same number of hidden layers $L$ and neurons $N_1, \ldots, N_L$, where $\Phi \colon \mathcal{X} \to \mathbb{R}^{N_{L+1}}$ and $\hat{\Phi} \colon H(\mathcal{X}) \to \mathbb{R}^{N_{L+1}}$, then the following statements hold for all $x \in \mathcal{X}$:*

*(1) If $\hat{x}_{0,i}^{(1)} = H(x_{0,i}^{(1)})$ and $\hat{x}_{0,i}^{(2)} = H(x_{0,i}^{(2)})$ for all $i = 1, 2, \ldots, N_1$, then $\Phi^{(1)}(x) = \hat{\Phi}^{(1)}(H(x))$.*

*(2) If $f$ is invariant w.r.t. $H$, then for any parameters of $\Phi$, there exist parameters of $\hat{\Phi}$ such that $\Phi(x) = \hat{\Phi}(H(x))$, and vice versa.*

*(3) The probability measure $P$ over the parameters is invariant under $H$.*

*Sketch of proof:* Any neuron in the sampled network can be written as $\phi(\langle s_1 \frac{w}{\|w\|^2}, x - x^{(1)} \rangle - s_2)$. As we divide by the square of the norm of $w$, the scalar in $H$ cancels. There is a difference between two points in both inputs of $\langle \cdot, \cdot \rangle$, which means the translation cancels. Orthogonal matrices cancel due to isometry. When $f$ is invariant with respect to $H$, the loss function is also unchanged and lead to the same output. Similar argument is made for $P$, and the theorem follows (cf. Appendix A.3).

If the input is embedded in a higher-dimensional ambient space $\mathbb{R}^{D'}$, with $D < D'$, we sample from a subspace with dimension $\tilde{D} = \dim(\overline{\text{span}\{\mathcal{X}\}}) \leq D'$. All the results presented in this section still hold due to orthogonal decomposition. However, the standard approach of backpropagation and initialization allows the weights to take on any value in $\mathbb{R}^{D'}$, which implies a lot of redundancy when $\tilde{D} \ll D'$. The biases are also more relevant to the input space than when initialized to zero — potentially avoiding the issues highlighted by Holzmüller and Steinwart [32]. For these reasons, we have named the proposed method Sampling Where It Matters (SWIM), which is summarized in Algorithm 1. For computational reasons, we consider a random subset of all possible pairs of training points when sampling weights and biases.

We end this section with a time and memory complexity analysis of Algorithm 1. In Table 1, we list runtime and memory usage for three increasingly strict assumptions. The main assumption is that the dimension of the output is less than or equal to the largest dimension of the hidden layers. This is true for the problems we consider, and the difference in runtime without this assumption is only reflected in the linear optimization part. The term $\lceil N/M \rceil$, i.e., integer ceiling of $N/M$, is required because only a subset of points are considered when sampling. For the full analysis, see Appendix F.

## 4 Numerical experiments

We now demonstrate the performance of Algorithm 1 on numerical examples. Our implementation is based on the numpy and scipy Python libraries, and we run all experiments on a machine with 32GB system RAM (256GB in Section 4.3 and Section 4.4) and a GeForce 4x RTX 3080 Turbo GPU with 10GB RAM. The Appendix contains detailed information on all experiments. In Section 4.1 we compare sampling to random Fourier feature models regarding the approximation of a Barron function. In Section 4.2 we compare classification accuracy of sampled networks to iterative, gradient-based optimization in a classification benchmark with real datasets. In Section 4.3 we demonstrate that more specialized architectures can be sampled, by constructing deep neural architectures as PDE solution operators. In Section 4.4 we show how to use sampling of fully-connected layers for transfer learning. For the probability distribution over the pairs in Algorithm 1, we always choose the $L^\infty$ norm for $\|\cdot\|_{\mathcal{Y}}$ and for $l = 1, 2, \ldots, L$, we choose the $L^2$ norm for $\|\cdot\|_{\mathcal{X}_{l-1}}$. The code to reproduce the experiments from the paper, and an up-to-date code base, can be found at

https://gitlab.com/felix.dietrich/swimnetworks-paper,
https://gitlab.com/felix.dietrich/swimnetworks.

**Algorithm 1:** The SWIM algorithm, for activation function $\phi$, and norms on input, output of the hidden layers, and output space, $\|\cdot\|_{\mathcal{X}_0}, \|\cdot\|_{\mathcal{X}_l}$, and $\|\cdot\|_{\mathcal{Y}}$ respectively. Also, $\mathcal{L}$ is a loss function, which in our case is always $L^2$ loss, and $\arg\min \mathcal{L}(\cdot, \cdot)$ becomes a linear optimization problem.

---

**Constant :** $\epsilon \in \mathbb{R}_{>0}$, $\varsigma \in \mathbb{N}_{>0}$, $L \in \mathbb{N}_{>0}$, $\{N_l \in \mathbb{N}_{>0}\}_{l=1}^{L+1}$, and $s_1, s_2 \in \mathbb{R}$
**Data:** $X = \{x_i \colon x_i \in \mathbb{R}^D, i = 1, 2, \ldots, M\}$,
  $Y = \{y_i \colon f(x_i) = y_i \in \mathbb{R}^{N_{L+1}}, i = 1, 2, \ldots, M\}$
$\Phi^{(0)}(x) = x$;
**for** $l = 1, 2, \ldots, L$ **do**
  $\tilde{M} \leftarrow \varsigma \cdot \lceil \frac{N_l}{M} \rceil \cdot M$ ;
  $P^{(l)} \in \mathbb{R}^{\tilde{M}}$; $P_i^{(l)} \leftarrow 0 \quad \forall i$;
  $\tilde{X} = \{(x_i^{(1)}, x_i^{(2)}) \colon \Phi^{(l-1)}(x_i^{(1)}) \neq \Phi^{(l-1)}(x_i^{(2)})\}_{i=1}^{\tilde{M}} \sim \mathrm{Uniform}(X \times X)$;
  **for** $i = 1, 2, \ldots, \tilde{M}$ **do**
    $\tilde{x}_i^{(1)}, \tilde{x}_i^{(2)} \leftarrow \Phi^{(l-1)}(x_i^{(1)}), \Phi^{(l-1)}(x_i^{(2)})$;
    $\tilde{y}_i^{(1)}, \tilde{y}_i^{(2)} = f(x_i^{(1)}), f(x_i^{(2)})$;
    $P_i^{(l)} \leftarrow \dfrac{\|\tilde{y}_i^{(2)} - \tilde{y}_i^{(1)}\|_{\mathcal{Y}}}{\max\{\|\tilde{x}_i^{(2)} - \tilde{x}_i^{(1)}\|_{\mathcal{X}_{l-1}}, \epsilon\}}$;
  **end**
  $W_l \in \mathbb{R}^{N_l, N_{l-1}}, b_l \in \mathbb{R}^{N_l}$;
  **for** $k = 1, 2, \ldots, N_l$ **do**
    Sample $(x^{(1)}, x^{(2)})$ from $\tilde{X}$, with replacement and with probability proportional to $P^{(l)}$;
    $\tilde{x}^{(1)}, \tilde{x}^{(2)} \leftarrow \Phi^{(l-1)}(x^{(1)}), \Phi^{(l-1)}(x^{(2)})$;
    $W_l^{(k, \colon)} \leftarrow s_1 \dfrac{\tilde{x}^{(2)} - \tilde{x}^{(1)}}{\|\tilde{x}^{(2)} - \tilde{x}^{(1)}\|^2}$; $b_l^{(k)} \leftarrow \langle W_l^{(k, \colon)}, \tilde{x}^{(1)} \rangle + s_2$;
  **end**
  $\Phi^{(l)}(\cdot) \leftarrow \phi(W_l \Phi^{(l-1)}(\cdot) - b_l)$;
**end**
$W_{L+1}, b_{L+1} \leftarrow \arg\min \mathcal{L}(\Phi^{(L)}(X), Y)$;
**return** $\{W_l, b_l\}_{l=1}^{L+1}$

---

Table 1: Runtime and memory requirements for training sampled neural networks with the SWIM algorithm, where $N = \max\{N_0, N_1, N_2, \ldots, N_L\}$. Assumption (I) is that the output dimension is less than or equal to $N$. Assumption (II) adds that $N < M^2$, i.e., number of neurons and input dimension is less than the size of dataset squared. Assumption (III) requires a fixed architecture.

|  | Runtime | Memory |
| --- | --- | --- |
| Assumption (I) | $\mathcal{O}(L \cdot M(\min\{\lceil N/M \rceil, M\} + N^2))$ | $\mathcal{O}(M \cdot \min\{\lceil N/M \rceil, M\} + LN^2)$ |
| Assumption (II) | $\mathcal{O}(L \cdot M(\lceil N/M \rceil + N^2))$ | $\mathcal{O}(M \cdot \lceil N/M \rceil + LN^2)$ |
| Assumption (III) | $\mathcal{O}(M)$ | $\mathcal{O}(M)$ |

## 4.1 Illustrative example: approximating a Barron function

We compare random Fourier features and our sampling procedure on a test function for neural networks [64]: $f(x) = \sqrt{3/2}(\|x - a\| - \|x + a\|)$, with $x \in \mathbb{R}^D$ and the vector $a \in \mathbb{R}^D$ defined by $a_j = 2j/D - 1$, $j = 1, 2, \ldots, D$. The Barron norm of $f$ is equal to one for all input dimensions, and it can be represented exactly with a network with one infinitely wide hidden layer, ReLU activation, and weights uniformly distributed on a sphere of radius $D^{1/2}$. We approximate $f$ using networks $\hat{f}$ of up to three hidden layers. The error is defined by $e_{\text{rel}}^2 = \sum_{x \in \mathcal{X}} (f(x) - \hat{f}(x))^2 / \sum_{x \in \mathcal{X}} f(x)^2$. We compare this error over the domain $\mathcal{X} = [-1, 1]^2$, with $10{,}000$ points sampled uniformly, separately for training and test sets. For random features, we use $w \sim N(0, 1)$, $b \sim U(-\pi, \pi)$, as proposed in [54], and $\phi = \sin$. For sampling, we also use $\phi = \sin$ to obtain a fair comparison. We also observed similar accuracy results when repeating the experiment with the $\tanh$ function. The number of neurons $m$ is the same in each hidden layer and ranges from $m = 64$ up to $m = 4096$. Figure 3 shows results for $D = 5, 10$ (results are similar for $D = 2, 3, 4$, and sampled networks can be constructed as fast as the random feature method, cf. Appendix B). Random features here have

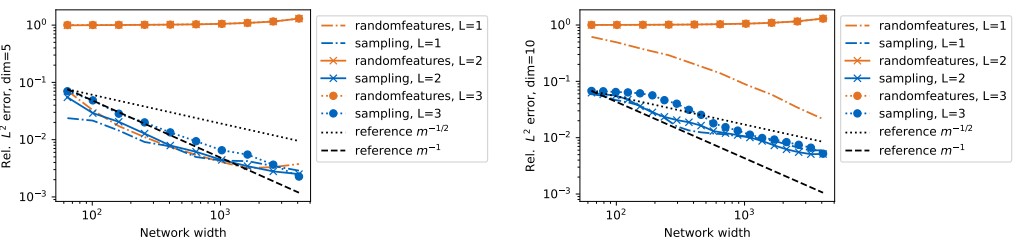

Figure 3: Relative $L^2$ approximation error of a Barron function (test set), using random features and sampling, both with sine activation. Left: input dimension $D = 5$. Right: input dimension $D = 10$.

comparable accuracy for networks with one hidden layer, but very poor performance for deeper networks. This may be explained by the much larger ambient space dimension of the data after it is processed through the first hidden layer. With our sampling method, we obtain accurate results even with more layers. The convergence rate for $D < 10$ seems to be faster than the theoretical rate.

## 4.2 Classification benchmark from OpenML

We use the "OpenML-CC18 Curated Classification benchmark" [4] with all its 72 tasks to compare our sampling method to the Adam optimizer [36]. With both methods, we separately perform neural architecture search, changing the number of hidden layers from 1 to 5. All layers always have 500 neurons. Details of the training are in Appendix C. Figure 4 shows the benchmark results. On all tasks, sampling networks is faster than training them iteratively (on average, 30 times faster). The classification accuracy is comparable (cf. Figure 4, second and third plot). The best number of layers for each problem is slightly higher for the Adam optimizer (cf. Figure 4, fourth plot).

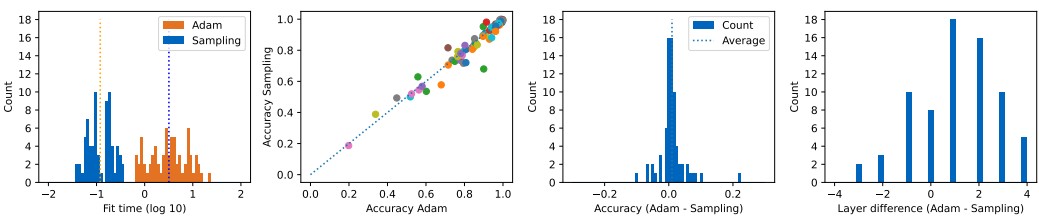

Figure 4: Fitting time, accuracy, and number of layers using weight sampling, compared to training with the Adam optimizer. The best architecture is chosen separately for each method and each problem, by evaluating 10-fold cross-validation error over 1-5 layers with 500 neurons each.

### 4.3 Deep neural operators

We sample deep neural operators and compare their speed and accuracy to iterative gradient-based training of the same architectures. As a test problem, we consider Burgers' equation, $\frac{\partial u}{\partial t} + u\frac{\partial u}{\partial x} = \nu\frac{\partial^2 u}{\partial^2 x}$, $x \in (0, 2\pi)$, $t \in (0, 1]$, with periodic boundary conditions and viscosity $\nu = 0.1$. The goal is to predict the solution at $t = 1$ from the initial condition at $t = 0$. Thus, we construct neural operators that represent the map $\mathcal{G} : u(x, 0) \rightarrow u(x, 1)$. We generate initial conditions by sampling five Fourier coefficients of lowest frequency and restoring the function values from these coefficients. Using a classical numerical solver, we generate 15000 pairs of $(u(x, 0), u(x, 1))$, and split them into the train (60%), validation (20%), and test sets (20%). Figure 5 shows samples from the generated dataset.

#### 4.3.1 Fully-connected network in signal space

The first baseline for the task is a fully-connected network (FCN) trained with tanh activation to predict the discretized solution from the discretized initial condition. We trained the classical version using the Adam optimizer and the mean squared error as a loss function. We also performed early stopping based on the mean relative $L^2$-error on the validation set. **For sampling**, we use Algorithm 1 to construct a fully-connected network with tanh as the activation function.

#### 4.3.2 Fully-connected network in Fourier space

Similarly to Poli et al. [53], we train a fully-connected network in Fourier space. For training, we perform a Fourier transform on the initial condition and the solution, keeping only the lowest frequencies. We always split complex coefficients into real and imaginary parts, and train a standard FCN on the transformed data. The reported metrics are in signal space, i.e., after inverse Fourier transform. **For sampling**, we perform exactly the same pre-processing steps.

#### 4.3.3 POD-DeepONet

The third architecture considered here is a variation of a deep operator network (DeepONet) architecture [44]. The original DeepONet consists of two trainable components: the trunk net, which transforms the coordinates of an evaluation point, and the branch net, which transforms the function values on some grid. The outputs of these nets are then combined into the predictions of the whole network $\mathcal{G}(u)(y) = \sum_{k=1}^{p} b_k(u)t_k(y) + b_0$, where $u$ is a discretized input function; $y$ is an evaluation point; $[t_1, \ldots, t_p]^T \in \mathbb{R}^p$ are the $p$ outputs of the trunk net; $[b_1, \ldots, b_p]^T \in \mathbb{R}^p$ are the $p$ outputs of the branch net; and $b_0$ is a bias. DeepONet sets no restrictions on the architecture of the two nets, but often fully-connected networks are used for one-dimensional input. POD-DeepONet proposed by Lu et al. [46] first assumes that evaluation points lie on the input grid. It performs proper orthogonal decomposition (POD) of discretized solutions in the train data and uses its components instead of the trunk net to compute the outputs $\mathcal{G}(u)(\xi_j) = \sum_{k=1}^{p} b_k(u)\psi_k(\xi_j) + \psi_0(\xi_j)$. Here $[\psi_1(\xi_j), \ldots, \psi_p(\xi_j)]$ are $p$ precomputed POD components for a point $\xi_j$, and $\psi_0(\xi_j)$ is the mean of discretized solutions evaluated at $\xi_j$. Hence, only the branch net is trained in POD-DeepONet. We followed Lu et al. [46] and applied scaling of $1/p$ to the network output. **For sampling**, we employ orthogonality of the components and turn POD-DeepONet into a fully-connected network. Let $\xi = [\xi_1, \ldots, \xi_n]$ be the grid used to discretize the input function $u$ and evaluate the output function $\mathcal{G}(u)$. Then the POD components of the training data are $\Psi(\xi) = [\psi_1(\xi), \ldots, \psi_p(\xi)] \in \mathbb{R}^{n \times p}$. If $b(u) \in \mathbb{R}^p$ is the output vector of the trunk net, the POD-DeepONet transformation can be written $\mathcal{G}(u)(\xi) = \Psi b(u) + \psi_0(\xi)$. As $\Psi^T\Psi = I_p$, we can express the output of the trunk net as $b(u) = \Psi^T(\mathcal{G}(u)(\xi) - \psi_0(\xi))$. Using this equation, we can transform the training data to sample a fully-connected network for $b(u)$. We again use tanh as the activation function for sampling.

#### 4.3.4 Fourier Neural Operator

The concept of a Fourier Neural Operator (FNO) was introduced by Li et al. [42] to represent maps in function spaces. An FNO consists of Fourier blocks, combining a linear operator in the Fourier space and a skip connection in signal space. As a first step, FNO lifts an input signal to a higher dimensional channel space. Let $v_t \in \mathbb{R}^{n \times d_v}$ be an input to a Fourier block having $d_v$ channels and discretized with $n$ points. Then, the output $v_{t+1}$ of the Fourier block is computed as $v_{t+1} = \phi(F_k^{-1}(R \cdot F_k(v_t)) + W \circ v_t) \in \mathbb{R}^{n \times d_v}$. Here, $F_k$ is a discrete Fourier transform keeping only

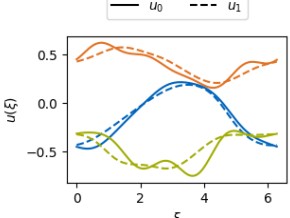

| | Model | width | layers | mean rel. $L^2$ error | Time | |
|---|---|---|---|---|---|---|
| Adam | FCN; signal space | 1024 | 2 | $4.48 \times 10^{-3}$ | 644s | GPU |
| | FCN; Fourier space | 1024 | 1 | $3.29 \times 10^{-3}$ | 1725s | |
| | POD-DeepONet | 2048 | 4 | $1.62 \times 10^{-3}$ | 4217s | |
| | FNO | n/a | 4 | $\mathbf{0.38 \times 10^{-3}}$ | 3119s | |
| Sampling | FCN; signal space | 4096 | 1 | $0.92 \times 10^{-3}$ | 20s | CPU |
| | FCN; Fourier space | 4096 | 1 | $1.08 \times 10^{-3}$ | 16s | |
| | POD-DeepONet | 4096 | 1 | $\mathbf{0.85 \times 10^{-3}}$ | 21s | |
| | FNO | 4096 | 1 | $0.94 \times 10^{-3}$ | 387s | |

Figure 5: Left: Samples of initial conditions $u_0$ and corresponding solutions $u_1$ for Burgers' equation. Right: Parameters of the best model for each architecture, the mean relative $L^2$ error on the test set, and the training time. We average the metrics across three runs with different random seeds.

the $k$ lowest frequencies and $F_k^{-1}$ is the corresponding inverse transform; $\phi$ is an activation function; $\cdot$ is a spectral convolution; $\circ$ is a $1 \times 1$ convolution with bias; and $R \in \mathbb{C}^{k \times d_v \times d_v}$, $W \in \mathbb{R}^{d_v \times d_v}$ are learnable parameters. FNO stacks several Fourier blocks and then projects the output signal to the target dimension. The projection and the lifting operators are parameterized with neural networks. **For sampling**, the construction of convolution kernels is not possible yet, so we cannot sample FNO directly. Instead, we use the idea of FNO to construct a neural operator with comparable accuracy. Similar to the original FNO, we normalize the input data and append grid coordinates to it before lifting. Then, we draw the weights from a uniform distribution on $[-1, 1]$ to compute the $1 \times 1$ lifting convolution. We first apply the Fourier transform to both input and target data, and then train a fully-connected network for each channel in Fourier space. We use skip connections, as in the original FNO, by removing the input data from the lifted target function during training, and then add it before moving to the output of the block. After sampling and transforming the input data with the sampled networks, we apply the inverse Fourier transform. After the Fourier block(s), we sample a fully-connected network that maps the signal to the solution.

The results of the experiments in Figure 5 show that sampled models are comparable to the Adam-trained ones. The sampled FNO model does not directly follow the original FNO architecture, as we are able to only sample fully-connected layers. This shows the advantage of gradient-based methods: as of now, they are applicable to much broader use cases. These experiments showcase one of the main advantages of sampled networks: speed of training. We run sampling on the CPU; nevertheless, we see a significant speed-up compared to Adam training performed on GPU.

## 4.4 Transfer learning

Training deep neural networks from scratch involves finding a suitable neural network architecture [21] and hyper-parameters [3, 66]. Transfer learning aims to improve performance on the target task by leveraging learned feature representations from the source task. This has been successful in image classification [35], multi-language text classification, and sentiment analysis [63, 9]. Here, we compare the performance of sampling with iterative training on an image classification transfer learning task. We choose the CIFAR-10 dataset [39], with 50000 training and 10000 test images. Each image has dimension $32 \times 32 \times 3$ and must be classified into one of the ten classes. We consider ResNet50 [31], VGG19 [58], and Xception [11], all pre-trained on the ImageNet dataset [55]. We freeze the weights of all convolutional layers and append one fully connected hidden layer and one output layer. We refer to these two layers as the classification head, which we sample with Algorithm 1 and compare the classification accuracy against iterative training with the Adam optimizer.

Figure 6 (left) shows that for a pre-trained ResNet50, the test accuracy using the sampling approach is higher than the Adam training approach for a width greater than 1024. We observe similar qualitative behavior for VGG19 and Xception (figures in Appendix E). Figure 6 (middle) shows that the sampling approach results in a higher test accuracy with all three pre-trained models. Furthermore, the deviation in test accuracy obtained with the sampling algorithm is very low, demonstrating that sampling is more robust to changing random seeds than iterative training. After fine-tuning the whole neural network with the Adam optimizer with a learning rate of $10^{-5}$, the test accuracies of sampled networks are close to the iterative approach. Thus, sampling provide a good starting point for fine-tuning the entire model. A comparison for the three models before and after fine-tuning is contained in Appendix E. Figure 6 (right) shows that sampling is up to two orders of magnitude faster than iterative training for

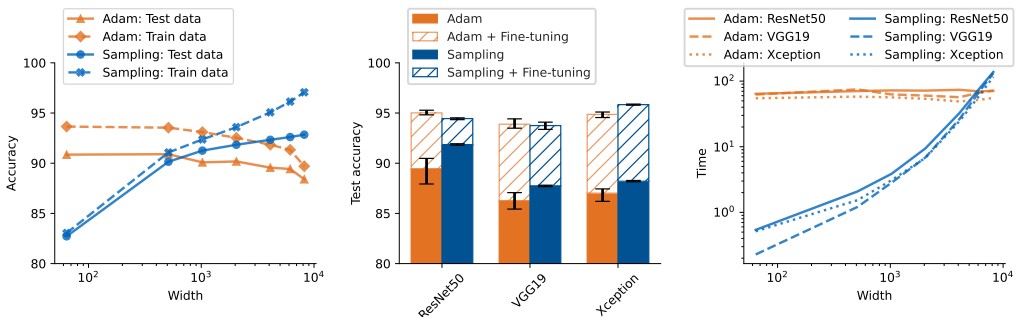

Figure 6: Left: Train and test accuracies with different widths for ResNet50 (averaged over 5 random seeds). Middle: Test accuracy with different models with and without fine-tuning (width = 2048). Right: Adam training and sampling times of the classification head (averaged over 5 experiments).

smaller widths, and around ten times faster for a width of 2048. In summary, Algorithm 1 is much faster than iterative training, yields a higher test accuracy for certain widths before fine-tuning, and is more robust with respect to changing random seeds. The sampled weights also provide a good starting point for fine-tuning of the entire model.

## 5    Broader Impact

Sampling weights through data pairs at large gradients of the target function offers improvements over random feature models. In terms of accuracy, networks with relatively large widths can even be competitive to iterative, gradient-based optimization. Constructing the weights through pairs of points also allows to sample deep architectures efficiently. Sampling networks offers a straightforward interpretation of the internal weights and biases, namely, which data pairs are important. Given the recent critical discussions around fast advancement in artificial intelligence, and calls to slow it down, publishing work that potentially speeds up the development (concretely, training speed) in this area by orders of magnitude may seem irresponsible. The solid mathematical underpinning of random feature models and, now, sampled networks, combined with much greater interpretability of the individual steps during network construction, should mitigate some of these concerns.

## 6    Conclusion

We present a data-driven sampling method for fully-connected neural networks that outperforms random feature models in terms of accuracy, and in many cases is competitive to gradient-based optimization. The time to obtain a trained network is orders of magnitude faster compared to gradient-based optimization. In addition, much fewer hyperparameters need to be optimized, as opposed to learning rate, number of training epochs, and type of optimizer.

Several open issues remain, we list the most pressing here. Many architectures like convolutional or transformer networks cannot be sampled with our method yet, and thus must still be trained with iterative methods. Implicit problems, such as the solution to PDE without any training data, are a challenge, as our distribution over the data pairs relies on known function values from a supervised learning setting. Iteratively refining a random initial guess may prove useful here. On the theory side, convergence rates for Algorithm 1 beyond the default Monte-Carlo estimate are not available yet, but are important for robust applications in engineering.

In the future, hyperparameter optimization, including neural architecture search, could benefit from the fast training time of sampled networks. We already demonstrate benefits for transfer learning here, which may be exploited for other pre-trained models and tasks. Analyzing which data pairs are sampled during training may help to understand the datasets better. We did not show that our sampling distribution results in optimal weights, so there is a possibility of even more efficient heuristics. Applications in scientific computing may benefit most from sampling networks, as accuracy and speed requirements are much higher than for many tasks in machine learning.

## Acknowledgments and Disclosure of Funding

We are grateful for discussions with Erik Bollt, Constantinos Siettos, Edmilson Roque Dos Santos, Anna Veselovska, and Massimo Fornasier. We also thank the anonymous reviewers at NeurIPS for their constructive feedback. E.B., I.B., and F.D. are funded by the Deutsche Forschungsgemeinschaft (DFG, German Research Foundation) - project no. 468830823, and also acknowledge association to DFG-SPP-229. C.D. is partially funded by the Institute for Advanced Study (IAS) at the Technical University of Munich. F.D. and Q.S. are supported by the TUM Georg Nemetschek Institute - Artificial Intelligence for the Built World.

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

# Appendix

Here we include full proofs of all theorems in the paper, and then details on all numerical experiments. In the last section, we briefly explain the code repository (URL in section Appendix F) and provide the analysis of the algorithm runtime and memory complexity.

## Contents

## A   Sampled networks theory

In this section, we provide complete proofs of the results stated in the main paper. In Appendix A.1 we consider sampled networks and the universal approximation property, followed by the result bounding $L_2$ approximation error between sampled networks and Barron functions in Appendix A.2. We end with the proof regarding equivariance and invariance in Appendix A.3.

We start by defining neural networks and sampled neural networks, and show more in depth the choices of the scalars in sampled networks for ReLU and tanh. Let the input space $\mathcal{X}$ be a subset of $\mathbb{R}^D$. We now define a fully connected, feed forward neural network; mostly to introduce some notation.

**Definition 3.** *Let $\phi\colon \mathbb{R} \to \mathbb{R}$ be non-polynomial, $L \geq 1$, $N_0 = D$, and $N_1, \ldots, N_{L+1} \in \mathbb{N}_{>0}$. A (fully connected) neural network with $L$ hidden layers, activation function $\phi$, and weight space $\Theta = \{W_l \in \mathbb{R}^{N_l, N_{l-1}}, b_l \in \mathbb{R}^{N_l}\}_{l=1}^{L+1}$, is a function $\Phi\colon \mathbb{R}^D \to \mathbb{R}^{N_{L+1}}$, of the form*

$$\Phi(x) = W_{L+1}x_L - b_{L+1}, \tag{3}$$

*where $x_l$ is defined recursively as $x_0 = x \in \mathcal{X}$ and*

$$x_l = \phi(W_l x_{l-1} - b_l), \quad l = 1, 2, \ldots, L. \tag{4}$$

The image after $l$ layers is denoted as $\mathcal{X}_l = \Phi^{(l)}(\mathcal{X})$.

As we study the space of these networks, we find it useful to introduce the following notation. The space of all neural networks with $L$ hidden layers, with $\boldsymbol{N} = [N_1, \ldots, N_L]$ neurons in the separate layers, is denoted as $\mathcal{F}_{L,\boldsymbol{N}}$, assuming the input dimension and the output dimension are implicitly given. If each hidden layer has $N$ neurons, and we let the input dimension and the output dimension be fixed, we write the space of all such neural networks as $\mathcal{F}_{L,N}$. We then let

$$\mathcal{F}_{\infty,N} = \bigcup_{L=1}^{\infty} \mathcal{F}_{L,N},$$

meaning the set of neural networks with $N$ width, and arbitrary depth. Similarly,

$$\mathcal{F}_{L,\infty} = \bigcup_{N=1}^{\infty} \mathcal{F}_{L,N},$$

for arbitrary width at fixed depth. Finally,

$$\mathcal{F}_{\infty,\infty} = \bigcup_{L=1}^{\infty} \bigcup_{N=1}^{\infty} \mathcal{F}_{L,N}.$$

We will now introduce *sampled* networks again, and go into depth the choice of constants. The input space $\mathcal{X}$ is a subset of Euclidean space, and we will work with canonical inner products $\langle \cdot, \cdot \rangle$ and their induced norms $\|\cdot\|$ if we do not explicitly specify the norm.

**Definition 4.** *Let $\Phi$ be an neural network with $L$ hidden layers. For $l = 1, \ldots, L$, let $(x_{0,i}^{(1)}, x_{0,i}^{(2)})_{i=1}^{N_l}$ be pairs of points sampled over $\mathcal{X} \times \mathcal{X}$. We say $\Phi$ is a sampled network if the weights and biases of every layer $l = 1, 2, \ldots, L$ and neurons $i = 1, 2, \ldots, N_l$, are of the form*

$$w_{l,i} = s_1 \frac{x_{l-1,i}^{(2)} - x_{l-1,i}^{(1)}}{\|x_{l-1,i}^{(2)} - x_{l-1,i}^{(1)}\|^2}, \quad b_{l,i} = \langle w_{l,i}, x_{l-1,i}^{(1)} \rangle + s_2 \tag{5}$$

*where $s_1, s_2 \in \mathbb{R}$ are constants related to the activation function, and $x_{l-1,i}^{(j)} = \Phi^{(l-1)}(x_{0,i}^{(j)})$ for $j = 1, 2$, assuming $x_{l-1,i}^{(1)} \neq x_{l-1,i}^{(2)}$. The last set of weights and biases are $W_{L+1}, b_{L+1} = \arg\min \mathcal{L}(W_{L+1}\Phi^{(L)}(\cdot) - b_{L+1})$, where $\mathcal{L}$ is a suitable loss.*

*Remark* 1. The constants $s_1, s_2$ can be fixed such that for a given activation function, we can specify values at specific points, e.g., we can specify what value to map the two points $x^{(1)}$ and $x^{(2)}$ to; cf. Figure 7. Also note that the points we sampled from $\mathcal{X} \times \mathcal{X}$ are sampled such that we have unique points after mapping the two sampled points through the first $l - 1$ layers. This is enforced by constructing the density of the distribution we use to sample the points so that zero density is assigned to points that map to the same output of $\Phi^{(l-1)}$ (see Definition 2).

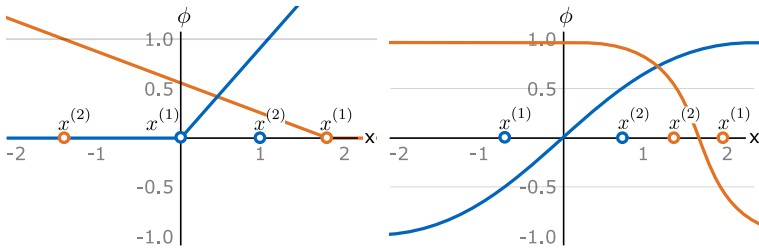

Figure 7: Illustration of the placement of point pairs $x^{(1)}, x^{(2)}$ for activation functions ReLU (left) and tanh (right).

**Example A.1.** We consider the construction of the constants $s_1, s_2$ for ReLU and tanh, as they are the two activation functions used for both the proofs and the experiments. We start by considering the activation function tanh, i.e.,

$$\phi(x) = \frac{\exp^{2x} - 1}{\exp^{2x} + 1}.$$

Consider an arbitrary weight $w$, where we drop the subscripts for simplicity. Setting $s_1 = 2 \cdot s_2$, the input of the corresponding activation function, at the mid-point $x = x^{(1)} + \frac{x^{(2)} - x^{(1)}}{2} = \frac{x^{(2)} + x^{(1)}}{2}$, is

$$\langle w, x \rangle - b = \left\langle w, \frac{x^{(2)} + x^{(1)}}{2} \right\rangle - \langle w, x^{(1)} \rangle - s_2$$

$$= \left\langle w, \frac{x^{(2)} - x^{(1)}}{2} \right\rangle - s_2$$

$$= \frac{2s_2}{\left\| x^{(2)} - x^{(1)} \right\|^2} \cdot \frac{1}{2} \left\langle x^{(2)} - x^{(1)}, x^{(2)} - x^{(1)} \right\rangle - s_2$$

$$= s_2 - s_2 = 0.$$

The output of the neuron corresponding to the input above is then zero, regardless of what the constant $s_2$ is. Another aspect of the constants are that we can decide activation values for the two sampled points. This can be seen with a similar calculation as above,

$$\langle w, x^{(1)} \rangle - b = -s_2.$$

Letting $s_2 = \frac{\ln 3}{2}$, we see that the output of the corresponding neuron at point $x^{(1)}$ is

$$\phi(-s_2) = \frac{\exp^{\ln(3)} - 1}{\exp^{\ln(3)} + 1} = -\frac{1}{2},$$

and for $x^{(2)}$ we get $\frac{1}{2}$. We can also consider the ReLU activation function where we choose to set $s_1 = 1$ and $s_2 = 0$. The center of the real line is then placed at $x^{(1)}$, and $x^{(2)}$ is mapped to one.

## A.1 Universality of sampled networks

In this section we show that universal approximation results also holds for our sampled networks. We start by considering the ReLU function. To separate it from tanh, which is the second activation function we consider, we denote $\phi$ to be ReLU and $\psi$ to be tanh. Networks with ReLU as activation function are then denoted by $\Phi$, and $\Psi$ are networks with tanh activation function. When we use ReLU, we set $s_1 = 1$ and $s_2 = 0$, as already discussed. We provide the notation that is used throughout Appendix A in Table 2.

The rest of this section is structured as follows:

1. We introduce the type of input spaces we are working with.
2. We then aim to rewrite an arbitrary fully connected neural network with one hidden layer into a sampled network.
3. This is done by first constructing neurons that add a constant to parts of the input space while leaving the rest untouched.
4. We then show that we can translate between an arbitrary neural network and a sampled network, if the weights of the former are given. This gives us the first universal approximator result, by combining this step with the former.
5. We go on to construct deep sampled networks with arbitrary width, showing that we can contain the information needed through the first $L - 1$ layers, and then apply the result for sampled networks with one hidden layer, to the last hidden layer.
6. We conclude the section by showing that sampled networks with tanh activation functions are also universal approximators. Concretely, we show that it is dense in the space of sampled networks with ReLU activation function. The reason we need a different proof for the tanh case is that it is not positive homogeneous, a property of ReLU that we heavily depend upon for the proof of sampled networks with ReLU as activation function.

Before we can start with the proof that $\mathcal{F}_{1,\infty}$ is dense in the space of continuous functions, we start by specifying the domain/input space for the functions.

Table 2: Notation used through the theory section of this appendix.

| | |
|---|---|
| $\mathcal{X}'$ | Pre-noise input space. Subset of $\mathbb{R}^D$ and compact for Appendix A.1 and Appendix A.2. |
| $\mathcal{X}$ | Input space. Subset of $\mathbb{R}^D$, $\mathcal{X}' \subset \mathcal{X}$, and compact for Appendix A.1 and Appendix A.2. |
| $\epsilon_I$ | Noise level for $\mathcal{X}'$. Definition 5. |
| $\phi$ | Activation function ReLU, $\phi(x) = \max\{x, 0\}$. |
| $\Phi$ | Neural network with activation function ReLU. |
| $\psi$ | Activation function tanh. |
| $\Psi$ | Neural network with activation function tanh. |
| $\Phi_c$ | Constant block. 5 neurons that combined add a constant value $c$ to parts of the input space $\mathcal{X}$, while leaving the rest untouched. Definition 6. |
| $N_l$ | Number of neurons in layer $l$ of a neural network. $N_0 = D$ and $N_{L+1}$ is the output dimension of network. |
| $w_{l,i}$ | The weight at the $i$th neuron, of the $l$th layer. |
| $b_{l,i}$ | The bias at the $i$th neuron, of the $l$th layer. |
| $\Phi^{(l)}$ | Function that maps $x \in \mathcal{X}$ to the output of the $l$th hidden layer. |
| $\Phi^{(l,i)}$ | The $i$th neuron for the $l$th layer. |
| $\mathcal{X}_l$ | The image of $\mathcal{X}$ after $l$ layers, i.e., $\mathcal{X}_l = \Phi(\mathcal{X})$. We set $\mathcal{X}_0 = \mathcal{X}$. |
| $x_l$ | $x_l = \Phi^{(l)}(x)$. |
| $\mathcal{F}_{L,[N_1,N_2,\dots,N_L]}$ | Space of neural networks with L hidden layers and $N_l$ neurons in layer $l$. |
| $\mathcal{F}_{1,\infty}$ | Space of all neural networks with one hidden layer, i.e., networks with arbitrary width. |
| $\mathcal{F}_{1,\infty}^S$ | Space of all sampled networks with one hidden layer and arbitrary width. |
| $\zeta$ | Function that maps points from $\mathcal{X}$ to $\mathcal{X}'$, mapping to the closest point in $\mathcal{X}'$, which is unique. |
| $\eta$ | Maps $x \in \mathcal{X} \times [0,1]$ to $\mathcal{X}$, by $\eta(x, \delta) = x + \delta(\zeta(x) - x)$. |
| $\|\cdot\|$ | Canonical norm of $\mathbb{R}^D$, with inner product $\langle \cdot \rangle$. |
| $\mathbb{B}_\epsilon(x)$ | The ball $\mathbb{B}_\epsilon(x) = \{x' \in \mathbb{R}^D : \|x - x'\| < \epsilon\}$. |
| $\overline{\mathbb{B}}_\epsilon(x)$ | The closed ball $\overline{\mathbb{B}}_\epsilon(x) = \{x' \in \mathbb{R}^D : \|x - x'\| \leq \epsilon\}$. |
| $\tau$ | Reach of the set $\mathcal{X}'$. |
| $\lambda$ | Lebesgue measure. |

### A.1.1   Input space

We again assume that the input space $\mathcal{X}$ is a subset of $\mathbb{R}^D$. We also assume that $\mathcal{X}$ contains some additional additive noise. Before we can define this noisy input space, we recall two concepts.

Let $(\mathcal{Z}, d)$ be a metric space, and the distance between a subset $A \subseteq \mathcal{Z}$ and a point $z \in \mathcal{Z}$ be defined as

$$d_{\mathcal{Z}}(z, A) = \inf\{d(z, a) : a \in A\}.$$

If $\mathcal{Z}$ is also a normed space, the medial axis is then defined as

$$\mathrm{Med}(A) = \{z \in \mathcal{Z} : \exists p \neq q \in A, \|p - z\| = \|q - z\| = d_{\mathcal{Z}}(z, A)\}$$

The reach of $A$ is defined as

$$\tau_A = \inf_{a \in A} d_{\mathcal{Z}}(a, \mathrm{Med}(A)).$$

Informally, the reach is the smallest distance from a point in the subset $A$ to a non-unique projection of it in the complement $A^c$. This means that the reach of convex subsets is infinite (all projections are unique), while other sets can have zero reach, which means $0 \leq \tau_A \leq \infty$.

Let $\mathcal{Z} = \mathbb{R}^D$, and $d$ be the canonical Euclidean distance.

**Definition 5.** *Let $\mathcal{X}'$ be a nonempty compact subset of $\mathbb{R}^D$ with reach $\tau_{\mathcal{X}'} > 0$. The input space $\mathcal{X}$ is defined as*

$$\mathcal{X} = \{x \in \mathbb{R}^D : d_{\mathcal{Z}}(x, \mathcal{X}') \leq \epsilon_I\},$$

*where $0 < \epsilon_I < \min\{\tau_{\mathcal{X}'}, 1\}$. We refer to $\mathcal{X}'$ as the pre-noise input space.*

*Remark* 2. As $d_{\mathcal{Z}}(x, \mathcal{X}') = 0$ for all $x \in \mathcal{X}'$, we have that $\mathcal{X}' \subset \mathcal{X}$. Due to $d_{\mathcal{Z}}(x, X') \leq \epsilon_I$ we preserve that $\mathcal{X}$ is closed, and as $\epsilon_I < \infty$ means it is bounded, and hence compact. Informally, we have enlarged the input space with new points at most $\epsilon_I$ distance away from $\mathcal{X}'$. This preserves many properties of the pre-noise input space $\mathcal{X}'$, while also being helpful in the proofs to come. We also argue that in practice, we are often faced with "noisy" datasets $\mathcal{X}$ anyway, rather than non-perturbed data $\mathcal{X}'$.

We also define a function that maps elements in $\mathcal{X} \setminus \mathcal{X}'$ down to $\mathcal{X}'$, using the uniqueness property given by the medial axis. That is, $\zeta \colon \mathcal{X} \to \mathcal{X}'$ is the mapping defined as $\zeta(x) = x'$, for $x \in \mathcal{X}$ and $x' \in \mathcal{X}'$, such that $d(x, x') = d_{\mathcal{Z}}(x, \mathcal{X}')$. As we also want to work along the line between these two points, we set $\eta \colon \mathcal{X} \times [0, 1] \to \mathcal{X}$, where $\eta(x, \delta) = x + \delta(\zeta(x) - x)$. We conclude this part with an elementary result.

**Lemma 1.** *Let $x \in \mathcal{X}$, $\delta \in [0, 1]$, and $x' = \eta(x, \delta) \in \mathcal{X}$, then $\|x' - x\| \leq \delta$, with strict inequality when $\delta > 0$. Furthermore, when $\delta > 0$, we have $x' \notin \partial\mathcal{X}$.*

*Proof.* We start by noticing

$$d(\zeta(x), x') = \|\zeta(x) - x'\| = (1 - \delta)\|\zeta(x) - x\| \leq (1 - \delta)\epsilon_I \leq \epsilon_I,$$

which means $x' \in \mathcal{X}$. When $\delta > 0$, we have strict inequality, which implies $x' \in \operatorname{int}\mathcal{X} = \mathcal{X} \setminus \partial\mathcal{X}$. Furthermore,

$$d(x, x') = \|x - x'\| = \delta\|\zeta(x) - x\| \leq \delta \cdot \epsilon_I \leq \delta,$$

where the last inequality holds due to $\epsilon_I < 1$, and is equal only when $\delta = 0$. $\qquad\square$

### A.1.2 Networks with a single hidden layer

We can now proceed to the proof that sampled networks with one hidden layer are indeed universal approximators. The main idea is to start off with an arbitrary network with a single hidden layer, and show that we can approximate this network arbitrarily well. Then we can rely on previous universal approximation theorems [13, 33, 52] to finalize the proof. We start by showing some results for a different type of neural networks, but very similar in form. We consider networks where the bias is of the same form as a sampled network. However, the weight is normalized to be a unit weight, that is, divided by the norm, and not the square of the norm. We show in Lemma 4 that the results also hold for any positive scalar multiple of the unit weight, and therefore our sampled network, where we divide by the norm of the weight squared. To be more precise, the weights are of the form

$$w_{l,i} = \frac{x_{l-1,i}^{(2)} - x_{l-1,i}^{(1)}}{\|x_{l-1,i}^{(2)} - x_{l-1,i}^{(1)}\|},$$

and biases $b_{l,i} = \langle w_{l,i}, x_{l-1,i}^{(1)} \rangle$. Networks with weights/biases in the hidden layers of this form is referred to as *unit sampled network*. In addition, when the output dimension is $N_{L+1} = 1$, we split the bias of the last layer, $b_{L+1}$ into $N_L$ parts, to make the proof easier to follow. This is of no consequence for the final result, as we can always sum up the parts to form the original bias. We write the different parts of the split as $b_{L+1,i}$ for $i = 1, 2, \ldots, N_L$.

We start by defining a constant block. This is crucial for handling the bias, as we can add constants to the output of certain parts of the input space, while leaving the rest untouched. This is important when proving Lemma 2.

**Definition 6.** *Let $c_1 < c_2 < c_3$, and $c \in \mathbb{R}^+$. A constant block $\Phi_c$ is defined as five neurons summed together as follows. For $x \in \mathbb{R}$,*

$$\Phi_c(x) = \sum_{i=1}^{5} f_i(x),$$

*where*

$$f_1(x) = a_1\,\phi(x - c_2), \quad f_2(x) = a_1\,\phi(-(x - c_3)),$$
$$f_3(x) = -a_1\,\phi(x - c_3), \quad f_4(x) = -a_2\,\phi(-(x - c_1))$$
$$f_5(x) = a_3\,\phi(x - c_1),$$

*and $a_1 = \frac{c}{c_3 - c_2}$, $a_2 = a_1\frac{c_1 - c_3}{c_1 - c_2}$, and $a_3 = a_2 - a_1$.*

*Remark* 3. The function $\Phi_c$ are constructed using neurons, but can also be written as the continuous function,

$$\Phi_c(x) = \begin{cases} 0, & x \le c_1 \\ a_3 \cdot x + d, & c_1 < x \le c_2 \\ c, & x > c_2, \end{cases}$$

where $d = a_1 c_3 + a_2 c_2$. Obviously, if $c$ needs to be negative, one can simply swap the sign on each of the three parameters, $a_i$.

We can see by the construction of $\Phi_c$ that we might need the negative of some original weight. That is, the input to $\Phi_c$ is of the form $\langle w_{1,i}, x \rangle$, and for $f_2$ and $f_4$, we require $\langle -w_{1,i}, x \rangle$. In Lemma 3, we shall see that this is not an issue and that we can construct neurons such that they approximate constant blocks arbitrarily well, as long as $c_1, c_2, c_3$ can be produced by the inner product between a weight and points in $\mathcal{X}$, that is, if we can produce the biases equal to the constants $c_1, c_2, c_3$.

Let $\hat{\Phi} \in \mathcal{F}_{1,K}$, with parameters $\{\hat{w}_{l,i}, \hat{b}_{l,i}\}_{l=1,i=1}^{2,K}$, be an arbitrary neural network. Unless otherwise stated, the weights in this arbitrary network $\hat{\Phi}$ are always nonzero. Sampled networks cannot have zero weights, as the point pairs used to construct weights, both for unit and regular sampled networks, are distinct. However, one can always construct the same output as neurons with zero weights by setting certain weights in $W_{L+1}$ to zero.

We start by showing that, given all weights of a network, we can construct all biases in a unit sampled network so that the function values agree. More precisely, we want to construct a network with weights $\hat{w}_{l,i}$, and show that we can find points in $\mathcal{X}$ to construct the biases of the form in unit sampled networks, such that the resulting neural network output on $\mathcal{X}$ equals exactly the values of the arbitrary network $\hat{\Phi}$.

**Lemma 2.** *Let* $\hat{\Phi} \in \mathcal{F}_{1,K} \colon \mathcal{X} \to \mathbb{R}^{N_2}$. *There exists a set of at most* $6 \cdot N_1$ *points* $x_i \in \mathcal{X}$ *and biases* $b_{2,i} \in \mathbb{R}$, *such that a network* $\Phi$ *with weights* $w_{1,i} = \hat{w}_{1,i}$, $w_{2,i} = \hat{w}_{2,i}$, *and biases* $b_{2,i} \in \mathbb{R}$,

$$b_{1,i} = \langle w_{1,i}, x_i \rangle,$$

*for* $i = 1, 2, \ldots, N_1$, *satisfies* $\Phi(x) - \hat{\Phi}(x) = 0$ *for all* $x \in \mathcal{X}$.

*Proof.* W.l.o.g., we assume $N_2 = 1$. For any weight/bias pair $\hat{w}_{1,i}, \hat{b}_{1,i}$, we let $w_{1,i} = \hat{w}_{1,i}$, $w_{2,i} = \hat{w}_{2,i}$, $B_i = \{\langle w_{1,i}, x \rangle \colon x \in \mathcal{X}\}$, $b_\wedge^{(i)} = \inf B_i$, and $b_\vee^{(i)} = \sup B_i$. As $\langle w_{1,i}, \cdot \rangle$ is continuous, means $B_i$ is compact and $b_\wedge^{(i)}, b_\vee^{(i)} \in B_i$.

We have four different cases, depending on $\hat{b}_{1,i}$.

(1) If $\hat{b}_{1,i} \in B_i$, then we simply choose a corresponding $x_i \in \mathcal{X}$ such that $b_{1,i} = \langle w_{1,i}, x_i \rangle = \hat{b}_{1,i}$. Letting $b_{2,i} = \hat{b}_{2,i}$, we have

$$w_{2,i}\phi(\langle w_{1,i}, x \rangle - b_{1,i}) - b_{2,i} = \hat{w}_{2,i}\phi(\langle \hat{w}_{1,i}, x \rangle - \hat{b}_{1,i}) - \hat{b}_{2,i}.$$

(2) If $\hat{b}_{1,i} > b_\vee^{(i)}$, we choose $x_i$ such that $b_{1,i} = \langle w_{1,i}, x_i \rangle = b_\vee^{(i)}$ and $b_{2,i} = \hat{b}_{2,i}$. As $\phi(\langle w_{1,i}, x \rangle - b_{1,i}) = \phi(\langle \hat{w}_{1,i}, x \rangle - \hat{b}_{1,i}) = 0$, for all $x \in \mathcal{X}$, we are done.

(3) If $\hat{b}_{1,i} < b_\wedge^{(i)}$, we choose corresponding $x_i$ such that $b_{1,i} = \langle w_{1,i}, x_i \rangle = b_\wedge^{(i)}$, and set $b_{2,i} = \hat{b}_{2,i} + w_{2,i}\left(\hat{b}_{1,i} - b_\wedge^{(i)}\right)$. We then have

$$\begin{aligned} w_{2,i}\phi(\langle w_{1,i}, x \rangle - b_{1,i}) - b_{2,i} &= w_{2,i}\langle w_{1,i}, x \rangle - w_{2,i} b_{1,i} - b_{2,i} \\ &= w_{2,i}\langle w_{1,i}, x \rangle - \hat{b}_{2,i} - w_{2,i}\hat{b}_{1,i} \pm w_{2,i}b_\wedge^{(i)} \\ &= \hat{w}_{2,i}\langle \hat{w}_{1,i}, x \rangle - \hat{b}_{2,i} - \hat{w}_{2,i}\hat{b}_{1,i} \\ &= \hat{w}_{2,i}\phi(\langle \hat{w}_{1,i}, x \rangle - \hat{b}_{1,i}) - \hat{b}_{2,i}, \end{aligned}$$

where first and last equality holds due to $\langle w_{1,i}, x \rangle > b_{1,i} > \hat{b}_{1,i}$ for all $x \in \mathcal{X}$.

(4) If $b_\wedge^{(i)} < \hat{b}_{1,i} < b_\vee^{(i)}$, and $\hat{b}_{1,i} \notin B_i$, things are a bit more involved. First notice that $B_i^{(1)} = B_i \cap [b_\wedge^{(i)}, \hat{b}_{1,i}]$ and $B_i^{(2)} = B_i \cap [\hat{b}_{1,i}, b_\vee^{(i)}]$ are both non-empty compact sets. We therefore have that supremum and infimum of both sets are members of their respective set, and thus also part of $B_i$. We therefore choose $x_i$ such that $b_{1,i} = \langle w_{1,i}, x_i \rangle = \inf B_i^{(2)}$. To make up for the difference between $\hat{b}_{1,i} < b_{1,i}$, we add a constant to all $x \in \mathcal{X}$ where $\langle w_{1,i}, x \rangle > b_{1,i}$. To do this we add some additional neurons, using our constant block $\Phi_c^{(i)}(\langle w_{1,i}, \cdot \rangle)$. Letting $c = w_{2,i}\left(b_{1,i} - \hat{b}_{1,i}\right)$, $c_1 = \sup B_i^{(1)}$, $c_2 = b_{1,i}$, and $c_3 = b_\vee^{(i)}$. We have now added five more neurons, and the weights and bias in second layer corresponding to the neurons is set to be $\pm a$ and 0, respectively, where both $a$ and the sign of $a$ depends on Definition 6. In case we require a negative sign in front of $\langle w_{1,i}, \cdot \rangle$, we simply set it as we are only concerned with finding biases given weights. We then have that for all $x \in \mathcal{X}$,

$$\Phi_c^{(i)}(\langle w_{1,i}, x \rangle) = \begin{cases} c, & \langle w_{1,i}, x \rangle > b_{1,i} \\ 0, & \text{otherwise.} \end{cases}$$

Finally, by letting $b_{2,i} = \hat{b}_{2,i}$, we have

$$w_{2,i}\phi(\langle w_{1,i}, x \rangle - b_{1,i}) - b_{2,i} + \Phi_c^{(i)}(\langle w_{1,i}, x \rangle)$$
$$= w_{2,i}\langle w_{1,i}, x \rangle - w_{2,i}\, b_{1,i} - \hat{b}_{2,i} + w_{2,i}\, b_{1,i} - \hat{w}_{2,i}\, \hat{b}_{1,i}$$
$$= \langle \hat{w}_{1,i}, x \rangle - \hat{w}_{2,i}\, \hat{b}_{1,i} - \hat{b}_{2,i}$$
$$= \hat{w}_{2,i}\phi(\langle \hat{w}_{1,i}, x \rangle \hat{b}_{1,i}) - \hat{b}_{2,i},$$

when $\langle w_{1,i}, x \rangle > b_{1,i}$, and

$$w_{2,i}\phi(\langle w_{1,i}, x \rangle - b_{1,i}) - b_{2,i} + \Phi_c^{(i)}(\langle w_{1,i}, x \rangle) = b_{2,i} = \hat{b}_{2,i}$$
$$= \hat{w}_{2,i}\phi(\langle w_{1,i}, x \rangle \hat{b}_{1,i}) - \hat{b}_{2,i},$$

otherwise. And thus, $\Phi(x) = \hat{\Phi}(x)$ for all $x \in \mathcal{X}$. As we add five additional neurons for each constant block, and we may need one for each neuron, means we need to construct our network with at most $6 \cdot N_1$ neurons.

$\square$

Now that we know we can construct suitable biases in our unit sampled networks for all types of biases $\hat{b}_{l,i}$ in the arbitrary network $\hat{\Phi}$, we show how to construct the weights.

**Lemma 3.** *Let $\hat{\Phi} \in \mathcal{F}_{1,K} \colon \mathcal{X} \to \mathbb{R}^{N_2}$, with biases of the form $\hat{b}_{1,i} = \langle \hat{w}_{1,i}, x_i \rangle$, where $x_i \in \mathcal{X}$ for $i = 1, 2, \ldots, N_1$. For any $\epsilon > 0$, there exist unit sampled network $\Phi$, such that $\|\Phi - \hat{\Phi}\|_\infty < \epsilon$.*

*Proof.* W.l.o.g., we assume $N_2 = 1$. For all $i = 1, 2, \ldots, N_1$, we construct the weights and biases as follows: If $x_i \notin \partial \mathcal{X}$, then we set $\epsilon' > 0$ such that $\mathbb{B}_{\epsilon'}(x_i) \subset \mathcal{X}$. Let $x_{0,i}^{(1)} = x_i$ and $x_{0,i}^{(2)} = x_{0,i}^{(1)} + \frac{\epsilon'}{2}w_{1,i} \in \mathbb{B}_{\epsilon'}(x_i) \subset \mathcal{X}$. Setting $w_{2,i} = \|\hat{w}_{1,i}\|\hat{w}_{2,i}$, $b_{2,i} = \hat{b}_{2,i}$, and $w_{1,i} = \frac{x_{0,i}^{(2)} - x_{0,i}^{(1)}}{\|x_{0,i}^{(2)} - x_{0,i}^{(1)}\|} = \frac{\hat{w}_{1,i}}{\|\hat{w}_{1,i}\|}$, implies

$$w_{2,i}\phi(\langle w_{1,i}, x - x_{0,i}^{(1)} \rangle) - b_{2,i} = \|\hat{w}_{1,i}\|\hat{w}_{2,i}\phi\left(\left\langle \frac{\hat{w}_{1,i}}{\|\hat{w}_{1,i}\|}, x - x_i \right\rangle\right) - \hat{b}_{2,i}$$
$$= \hat{w}_{2,i}\phi(\langle \hat{w}_i, x - x_i \rangle) - \hat{b}_{2,i},$$

where last equality follows by $\phi$ being positive homogeneous.

If $x_i \in \partial \mathcal{X}$, by continuity we find $\delta > 0$ such that for all $i = 1, 2, \ldots, N_1$ and $x', x \in \mathcal{X}$, where $\|x' - x_i\| < \delta$, we have

$$|\phi(\langle \hat{w}_{1,i}, x - x' \rangle) - \phi(\langle \hat{w}_{1,i}, x - x_i \rangle)| < \frac{\epsilon}{N_1 w_2},$$

where $w_2 = \max\{|\hat{w}_{2,i}|\}_{i=1}^{N_1}$. We set $x_{0,i}^{(1)} = \eta(x_i, \min\{\delta, 1\})$, with $x_{0,i}^{(1)} \in \text{int } \mathcal{X}$ and $\|x_i - x_{0,i}^{(1)}\| < \delta$, due to $\delta > 0$ and Lemma 1. We may now proceed by constructing $x_{0,i}^{(2)}$ as above, and similarly setting $w_{2,i} = \|\hat{w}_{1,i}\|\hat{w}_{2,i}$, $b_{2,i} = \hat{b}_{2,i}$, we have

$$\left| \left( \sum_{i=1}^{N_1} w_{2,i}\phi(\langle w_{1,i}, x - x_{0,i}^{(1)}\rangle) - b_{2,i} \right) - \left( \sum_{i=1}^{N_1} \hat{w}_{2,i}\phi(\langle \hat{w}_{1,i}, x - x_i\rangle) - \hat{b}_{2,i} \right) \right|$$

$$= \left| \sum_{i=1}^{N_1} \hat{w}_{2,i} \left( \phi(\langle \hat{w}_{1,i}, x - x_{0,i}^{(1)}\rangle) - \phi(\langle \hat{w}_{1,i}, x - x_i\rangle) \right) \right|$$

$$\leq \sum_{i=1}^{N_1} \left| w_2 \left( \phi(\langle \hat{w}_{1,i}, x - x_{0,i}^{(1)}\rangle) - \phi(\langle \hat{w}_{1,i}, x - x_i\rangle) \right) \right|$$

$$< \left| \sum_{i=1}^{N_1} w_2 \frac{\epsilon}{N_1 \, w_2} \right| = \epsilon,$$

and thus $\|\Phi - \hat{\Phi}\|_\infty < \epsilon$. $\qquad \square$

Until now, we have worked with weights of the form $w = \frac{x^{(2)} - x^{(1)}}{\|x^{(2)} - x^{(1)}\|}$, however, the weights in a sampled network are divided by the norm squared, not just the norm. We now show that for all the results so far, and also for any other results later on, differing by a positive scalar (such as this norm) is irrelevant when ReLU is the activation function.

**Lemma 4.** *Let $\Phi$ be a network with one hidden layer, with weights and biases of the form*

$$w_{1,i} = \frac{x_{0,i}^{(2)} - x_{0,i}^{(1)}}{\|x_{0,i}^{(2)} - x_{0,i}^{(1)}\|}, \quad b_{1,i} = \langle w_{1,i}, x_{0,i}^{(1)}\rangle,$$

*for $i = 1, 2, \ldots, N_1$. For any weights and biases in the last layer, $\{w_{2,i}, b_{2,i}\}_{i=1}^{N_2}$, and set of strictly positive scalars $\{\omega_i\}_{i=1}^{N_1}$, there exist sampled networks $\Phi_\omega$ where weights and biases in the hidden layer $\{w'_{1,i}, b'_{1,i}\}_{i=1}^{N_1}$ are of the form*

$$w'_{1,i} = \omega_i w_{1,i}, \quad b'_{1,i} = \langle w'_{1,i}, x_{0,i}^{(1)}\rangle,$$

*such that $\Phi_\omega(x) = \Phi(x)$ for all $x \in \mathcal{X}$.*

*Proof.* We set $w'_{2,i} = \frac{w_{2,i}}{\omega_i}$ and $b'_{2,i} = b_{2,i}$. As ReLU is a positive homogeneous function, we have for all $x \in \mathcal{X}$,

$$w'_{2,i}\phi(\langle w'_{1,i}, x\rangle - b'_{1,i}) - b'_{1,i} = \frac{\omega_i w_{2,i}}{\omega_i}\phi(\langle w_{1,i}, x\rangle - b_{1,i})$$

$$w_{2,i}\phi(\langle w_{1,i}, x\rangle - b_{1,i}).$$

$\qquad \square$

The result itself is not too exciting, but it allows us to use the results proven earlier, applying them to sampled networks by setting the scalar to be $\omega_i = \frac{1}{\|x_{0,i}^{(2)} - x_{0,i}^{(1)}\|}$.

We are now ready to show the universal approximation property for sampled networks with one hidden layer. We let $\mathcal{F}_{1,\infty}^S$ be defined similarly to $\mathcal{F}_{1,\infty}$, with every $\Phi \in \mathcal{F}_{1,\infty}^S$ being a sampled neural network.

**Theorem 4.** *Let $g \in C(\mathcal{X}, \mathbb{R}^{N_{L+1}})$. Then, for any $\epsilon > 0$, there exist $\Phi \in \mathcal{F}_{1,\infty}^S$ with ReLU activation function, such that*

$$\|g - \Phi\|_\infty < \epsilon.$$

*That is, $\mathcal{F}_{1,\infty}^S$ is dense in $C(\mathcal{X}, \mathbb{R}^{N_{L+1}})$.*

*Proof.* W.l.o.g., let $N_{L+1} = 1$, and in addition, let $\epsilon > 0$ and $g \in C(\mathcal{X}, \mathbb{R}^{N_{L+1}})$. Using the universal approximation theorem of Pinkus [52], we have that for $\epsilon > 0$, there exist a network $\hat{\Phi} \in \mathcal{F}_{1,\infty}$, such that $\|g - \hat{\Phi}\|_\infty < \frac{\epsilon}{2}$. Let $K$ be the number of neurons in $\hat{\Phi}$. We then create a new network $\Phi$, by first keeping the weights fixed to the original ones, $\{\hat{w}_{1,i}, \hat{w}_{2,i}\}_{i=1}^K$, and setting the biases of $\Phi$ according to Lemma 2 using $\{\hat{b}_{1,i}, \hat{b}_{2,i}\}_{i=1}^K$, adding constant blocks if necessary. We then change the weights of our $\Phi$ with the respect to the new biases, according to Lemma 3 (with the epsilon set to $\epsilon/2$). It follows from the two lemmas that $\|\Phi - \hat{\Phi}\|_\infty < \frac{\epsilon}{2}$. That means

$$\|g - \Phi\|_\infty = \|g - \hat{\Phi} + \hat{\Phi} - \Phi\|_\infty \leq \|g - \hat{\Phi}\|_\infty + \|\hat{\Phi} - \Phi\|_\infty < \epsilon.$$

As the weights of the first layer of $\Phi$ can be written as $w_{1,i} = \frac{x_{0,i}^{(2)} - x_{0,i}^{(1)}}{\|x_{0,i}^{(2)} - x_{0,i}^{(1)}\|^2}$ and bias $b_{1,i} = \langle w_{1,i}, x_{0,i}^{(1)} \rangle$, both guaranteed by Lemma 4, means $\Phi \in \mathcal{F}_{1,\infty}^S$. Thus, $\mathcal{F}_{1,\infty}^S$ is dense in $C(\mathcal{X}, \mathbb{R}^{N_{L+1}})$. □

*Remark* 4. By the same two lemmas, Lemma 2 and Lemma 3, one can show that other results regarding networks with one hidden layer with at most $K$ neurons, also hold for sampled networks, but with $6 \cdot K$ neurons, due to the possibility of one constant block for each neuron. When $\mathcal{X}$ is a connected set, we only need $K$ neurons, as no additional constant blocks must be added; see proof of Lemma 2 for details.

### A.1.3 Deep networks

The extension of sampled networks into several layers is not obvious, as the choice of weights in the first layer affects the sampling space for the weights in the next layer. This additional complexity raises the question, letting $\boldsymbol{N_L} = [N_1, N_2, \ldots, N_{L-1}, N_L]$, is the space $\bigcup_{N_L=1}^\infty \mathcal{F}_{L,\boldsymbol{N_L}}^S$ dense in $C(\mathcal{X}, \mathbb{R}^{N_{L+1}})$, when $L > 1$? We aim to answer this question in this section. With dimensions in each layer being $\boldsymbol{N_L} = [D, D, \ldots, D, N_L]$, we start by showing it holds for $\bigcup_{N_L=1}^\infty \mathcal{F}_{L,\boldsymbol{N_L}}^S$, i.e., the space of sampled networks with $L$ hidden layers, and $D$ neurons in the first $L-1$ layers and arbitrary width in the last layer.

**Lemma 5.** *Let* $\boldsymbol{N_L} = [D, D, \ldots, D, N_L]$ *and* $L > 1$. *Then* $\bigcup_{N_L=1}^\infty \mathcal{F}_{L,\boldsymbol{N_L}}^S$ *is dense in* $C(\mathcal{X}, \mathbb{R}^{N_{L+1}})$.

*Proof.* Let $\epsilon > 0$, and $g \in C(\mathcal{X}, \mathbb{R}^{N_{L+1}})$. Basic linear algebra implies there exists a set of $D$ linearly independent unit vectors $v = \{v_j\}_{j=1}^D$, such that $\mathcal{X} \subseteq \text{span}\{v_j\}_{j=1}^D$, with $v_j \in \mathcal{X}$ for all $v_j \in v$. For $l = 1$ and $i \in \{1, 2, \ldots, N_l\}$, the bias is set to $b_{l,i} = \inf\{\langle v_i, x \rangle : x \in \mathcal{X}\}$. Due to continuity and compactness, we can set $x_{l-1,i}^{(1)}$ to correspond to an $x \in \mathcal{X}$ such that $\langle v_i, x_{l-1,i}^{(1)} \rangle = b_{l,i}$. We must have $x_{l-1,i}^{(1)} \in \partial\mathcal{X}$ — otherwise the inner product between some points in $\mathcal{X}$ and $v_i$ is smaller than the bias, which contradicts the construction of $b_{l,i}$. We now need to show that $\zeta(x_{l-1,i}^{(1)}) - x_{l-1,i}^{(1)} = c\,v_i + x_{l-1,i}^{(1)}$ for $c \in \mathbb{R}_{>0}$, to proceed constructing $x_{l-1,i}^{(2)}$ in similar fashion as in Lemma 3.

Let $U = \overline{\mathbb{B}}_{\epsilon_I}(\zeta(x_{l-1,i}^{(1)}))$, with $x_{l-1,i}^{(1)} \in U$. Also, $U$ is the only closed ball with center $\zeta(x_{l-1,i}^{(1)})$ and $\{x_{l-1,i}^{(1)}\} = U \cap \partial\mathcal{X}$ — otherwise $\zeta$ would not give a unique element, which is guaranteed by Definition 5. We define the hyperplane $\text{Ker} = \{x \in \mathbb{R}^D : \langle v_i, x \rangle - \langle v_i, x_{l-1,i}^{(1)} \rangle = 0\}$, and the projection matrix $P$ from $\mathbb{R}^D$ onto Ker. Let $x' = P\zeta(x_{l-1,i}^{(1)})$. If $x' \neq x_{l-1,i}^{(1)}$, then $x' \in \text{int}U$, as $x_{l-1,i}^{(1)} \in \partial U$ and $P$ is a unique projection minimizing the distance. As $x' \in \text{int}\,U$, means there is an open ball around $x'$, where there are points in $\overline{\mathbb{B}}_{\epsilon_I}(\zeta(x_{l-1,i}^{(1)}))$ separated by Ker, which implies $\langle v_i, x_{l-1,i}^{(1)} \rangle$ is not the minimum. This is a contradiction, and hence $x' = x_{l-1,i}^{(1)}$. This means Ker is a tangent hyperplane, and as any vectors along the hyperplane is orthogonal to $\zeta(x_{l,i}^{(1)}) - x_{l,i}^{(1)}$, implies the angle between $v_i$ and $\zeta(x_{l-1,i}^{(1)}) - x_{l-1,i}^{(1)}$ is 0 or $\pi$. As $\langle v_i, x_{l-1,i}^{(1)} \rangle$ is the minimum, means there exist a $c \in \mathbb{R}_{>0}$, such that $\zeta(x_{l-1,i}^{(1)}) - x_{l-1,i}^{(1)} = c\,v_i + x_{l-1,i}^{(1)}$.

We may now construct $x_{l-1,i}^{(2)} = x_{l-1,i}^{(1)} + c\,v_i$, assured that $x_{l-1,i}^{(2)} \in \mathcal{X}$ due to the last paragraph. We then have $w_{l,i} = \frac{v_i}{\|v_i\|}$ and $b_{l,i} = \langle v_i, x_{l-1,i}^{(1)} \rangle$ for all neurons in the first hidden layer. The image

after this first hidden layer $\Psi^{(l)}(\mathcal{X})$ is injective, and therefore bijective. To show this, let $u_1, u_2 \in \mathcal{X}$ such that $u_1 \neq u_2$. As the vectors in $v$ spans $\mathcal{X}$, means there exists two unique set of coefficients, $c_1, c_2 \in \mathbb{R}^D$, such that $u_1 = \sum c_1^{(i)} v_i$ and $u_2 = \sum c_2^{(i)} v_i$. We then have

$$\Phi^{(l)}(u_1) = V c_1 - b_l \neq V c_2 - b_l = \Phi^{(l)}(u_2),$$

where $V$ is the Gram matrix of $v$. As vectors in $v$ are linearly independent, $V$ is positive definite, and combined with $c_1 \neq c_2$, this implies the inequality. That means $\Phi^{(l)}$ is a bijective mapping of $\mathcal{X}$. As the mapping is bijective and continuous, we have that for any $x \in \mathcal{X}'$, there is an $\epsilon_I^{(l)} > 0$, such that $\overline{\mathbb{B}}_{\epsilon_I'}(\Phi^{(l)}(x)) \subseteq \mathcal{X}$.

For $1 < l < L$, we repeat the procedure, but swap $\mathcal{X}$ with $\mathcal{X}_{l-1}$. As $\Phi^{(l-1)}$ is a bijective mapping, we may find similar linear independent vectors and construct similar points $x_{l-1,i}^{(1)}, x_{l-1,i}^{(2)}$, but now with noise level $\epsilon_I^{(l)}$. For $l = L$, as we have a subset of $\mathcal{X}$ that is a closed ball around each point in $\Phi^{l-1}(x)$, for every $x \in \mathcal{X}'$, means we can proceed by constructing the last hidden layer and the last layer in the same way as explained when proving $Theorem\ 4$. The only caveat is that we are approximating a network with one hidden layer with the domain $\mathcal{X}_{l-1}$, and the function we approximate is $\tilde{g} = g \circ [\Phi^{(L-1)}]^{-1}$. Given this, denoting $\Phi^{(L:L+1)}$ as the function of last hidden layer and the last layer, there exists a number of nodes, weights, and biases in the last hidden layer and the last layer, such that

$$\|g - \Phi\|_\infty = \|\tilde{g} - \Phi^{(L:L+1)}\|_\infty < \epsilon,$$

due to construction above and Theorem 4. As $\Phi$ is a unit sampled network, it follows by Lemma 4 that $\bigcup_{N_L=1}^{\infty} \mathcal{F}_{L,\boldsymbol{N_L}}^S$ is dense in $C(\mathcal{X}, \mathbb{R}^{N_L+1})$. $\qquad\square$

We can now prove that sampled networks with $L$ layers, and different dimensions in all neurons, with arbitrary width in the last hidden layer, are universal approximators, with the obvious caveat that each hidden layer $l = 1, 2, \ldots, L - 1$ needs at least $D$ neurons, otherwise we will lose some information regardless of how we construct the network.

**Theorem 5.** *Let $\boldsymbol{N_L} = [N_1, N_2, \ldots, N_{L-1}, N_L]$, where $\min\{N_l : l = 1, 2, \ldots, L - 1\} \geq D$, and $L \geq 1$. Then $\bigcup_{N_L=1}^{\infty} \mathcal{F}_{L,\boldsymbol{N_L}}^S$ is dense in $C(\mathcal{X}, \mathbb{R}^{N_L+1})$.*

*Proof.* Let $\epsilon > 0$ and $g \in C(\mathcal{X}, \mathbb{R}^{N_L+1})$. For $L = 1$, Theorem 4 is enough, and we therefore assume $L > 1$. We start by constructing a network $\tilde{\Phi} \in \bigcup_{N_L=1}^{\infty} \mathcal{F}_{L,\tilde{\boldsymbol{N_L}}}^S$, where $\tilde{\boldsymbol{N_L}} = [D, D, \ldots, D, N_L]$ according to Lemma 5, such that $\|\tilde{\Phi} - g\|_\infty < \epsilon$. To construct $\Phi \in \bigcup_{N_L=1}^{\infty} \mathcal{F}_{L,\boldsymbol{N_L}}^S$, let $l = 1$, and start by constructing weights/biases for the first $D$ nodes according to $\tilde{\Phi}$. For the additional nodes, in the first hidden layer, select an arbitrary direction $w$. Let $X_1 = \{\tilde{x}_{1,i}^{(j)} : i = 1, 2, \ldots, D$ and $j = 1, 2\}$ be the set of all points needed to construct the $D$ neurons in the next layer of $\tilde{\Phi}$. Then for each additional node $i = D + 1, \ldots, N_1$, we set

$$x_{0,i}^{(1)} = \arg\max\{\langle w, x \rangle : x \in \mathcal{X} \text{ and } \tilde{\Phi}^{(1)}(x) \in X_1\}.$$

and choose $x_{0,i}^{(2)} \in \mathcal{X}$ such that $x_{0,i}^{(2)} - x_{0,i}^{(1)} = aw$, where $a \in \mathbb{R}_{>0}$, similar to what is done in the proof for Lemma 3. Using these points to define the weights and biases of the last $N_1 - D$ nodes, the following space $\mathcal{X}_1$ now contains points $[\tilde{x}_{1,i}^{(j)}, 0, 0, \ldots, 0]$, for $j = 1, 2$ and $i = 1, 2, \ldots, D$. For $1 < l < L$ repeat the process above, setting $x_{l,i}^{(j)} = [\tilde{x}_{l,i}^{(j)}, 0, 0, \ldots, 0]$ for the first $D$ nodes, and construct the weights of the additional nodes as described above, but with sampling space being $\mathcal{X}_{l-1}$. When $l = L$, set number of nodes to the same as in $\tilde{\Phi}$, and choose the points to construct the weights and biases as $x_{L-1,i}^{(j)} = [\tilde{x}_{L-1,i}^{(j)}, 0, 0, \ldots, 0]$, for $j = 1, 2$ and $i = 1, 2, \ldots, N_L$. The weights and biases in the last layer are the same as in $\tilde{\Phi}$. This implies,

$$\|\Phi - g\|_\infty = \|\tilde{\Phi} - g\|_\infty < \epsilon,$$

and thus $\bigcup_{N_L=1}^{\infty} \mathcal{F}_{L,\boldsymbol{N_L}}^S$ is dense in $C(\mathcal{X}, \mathbb{R}^{N_L+1})$. $\qquad\square$

*Remark* 5. Important to note that the proof is only showing existence, and that we expect networks to have a more interesting representation after the first $L-1$ layers. With this theorem, we can conclude that stacking layers is not necessarily detrimental for the expressiveness of the networks, even though it may alter the sampling space in non-trivial ways. Empirically, we also confirm this, with several cases performing better under deep networks — very similar to iteratively trained neural networks.

**Corollary 1.** $\mathcal{F}^S_{\infty,\infty}$ *is dense in* $C(\mathcal{X}, \mathbb{R}^{N_{L+1}})$.

### A.1.4 Networks with a single hidden layer, tanh activation

We now turn to using tanh as activation function, which we find useful for both prediction tasks, and if we need the activation function to be smooth. We will use the results for sampled networks with ReLU as activation function, and show we can arbitrarily well approximate these. The reason for this, instead of using arbitrary network with ReLU as activation function, is that we are using weights and biases of the correct form in the former, such that the tanh networks we construct will more easily have the correct form. We set $s_2 = \frac{\ln(3)}{2}$, $s_1 = 2 \cdot s_2$ — as already discussed — and let $\psi$ be the tanh function, with $\Psi$ being neural networks with $\psi$ as activation function — simply to separate from the ReLU $\phi$, as we are using both in this section. Note that $\Phi$ is still network with ReLU as activation function and $s_1 = 1, s_2 = 0$.

We start by showing how a sum of tanh functions can approximate a set of particular functions.

**Lemma 6.** *Let* $f \colon [c_0, c_{M+1}] \to \mathbb{R}^+$*, defined as* $f(x) = \sum_{i=1}^{M} a_i \, \mathbf{1}_{[c_i, c_{M+1}]}(x)$*, with* $\mathbf{1}$ *being the indicator function,* $c_0 < c_1 < \cdots < c_M < c_{M+1}$*, and for all* $i = 0, 1, \ldots, M+1$*,* $c_i \in \mathbb{R}$ *and* $a_i \in \mathbb{R}_{>0}$*. Then there exists strictly positive scalars* $\omega = \{\omega_i\}_{i=1}^{M}$ *such that*

$$g(x) = \sum_{i=1}^{M} g_i(x) = \sum_{i=1}^{M} \frac{a_i}{2} \left[ \psi(\omega_i(x - c_i) - s_2) + 1 \right],$$

*fulfills* $f(c_{i-1}) < g(x) < f(c_{i+1})$ *whenever* $x \in [c_{i-1}, c_{i+1}]$*, for all* $i = 1, 2, \ldots, M$*.*

*Proof.* We start by observing that both functions, $f$ and $g$, are increasing, with the latter strictly increasing. We also have that $f(c_0) = 0 < g(x) < \sum_{i=1}^{M} a_i = f(c_M)$, for all $x \in [c_0, c_{M+1}]$, regardless choice of $\omega$. We then fix constants

$$0 < \delta_i < \frac{\frac{3}{4} a_i}{M - i} \qquad 0 < \epsilon_{i+1} < \frac{\frac{a_i}{4}}{i},$$

for $i = 1, 2, \ldots, M-1$. We have, due to $s_2$, that $g_i(c_i) = \frac{a_i}{4}$, for all $i = 1, 2, \ldots, M$. In addition, we can always increase $\omega_i$ to make sure $g_i(c_{i+1})$ is large enough, and $g_i(c_{i-1})$ small enough for our purposes, as $\psi$ is bijective and strictly increasing. We set $\omega_1$ large enough such that $g_1(c_j) > a_1 - \epsilon_j$, for all $j = 2, 3, \ldots, M-1$. For $i = 2, 3, \ldots, M-1$, we set $\omega_i$ large enough such that $g_i(c_j) < \delta_j$, where $j = 1, 2, \ldots, i-1$, and $g_i(c_j) > a_i - \epsilon_j$, where $j = i+1, i+2, \ldots, M-1$. Finally, let $\omega_M$ be large enough such that $g_M(c_j) < \delta_j$, for $j = 1, 2, \ldots, M-1$. With the strictly increasing property of every $g_i$, we see that

$$g(c_i) = \sum_{j=1}^{i-1} g_j(c_i) + \frac{a_i}{4} + \sum_{j=i+1}^{M} g_j(c_i)$$

$$< \sum_{j=1}^{i-1} a_j + \frac{a_i}{4} + \sum_{j=i+1}^{M} \delta_j$$

$$= \sum_{j=1}^{i-1} a_j + \frac{a_i}{4} + \frac{3a_i}{4} = f(c_i),$$

and

$$g(c_i) = \sum_{j=1}^{i-1} g_j(c_i) + \frac{a_i}{4} + \sum_{j=i+1}^{M} g_j(c_i)$$

$$> \sum_{j=1}^{i-1}(a_j - \epsilon_i) + \frac{a_i}{4}$$

$$= \sum_{j=1}^{i-1} a_j - \frac{a_i}{4} + \frac{a_i}{4} = f(c_{i-1}).$$

Combing the observations at the start with $f(c_{i-1}) < g(c_i) < f(c_i)$, for $i = 1, 2, \ldots, M$, and the property sought after follows quickly. $\qquad\square$

We can now show that we can approximate a neuron with ReLU $\phi$ activation function and with unit sampled weights arbitrarily well.

**Lemma 7.** *Let $\hat{x}^{(1)}, \hat{x}^{(2)} \in \mathcal{X}$ and $\hat{w}_2 \in \mathbb{R}$. For any $\epsilon > 0$, there exist a $M \in \mathbb{N}_{>0}$, and $M$ pairs of distinct points $\{(x_i^{(2)}, x_i^{(1)}) \in \mathcal{X} \times \mathcal{X}\}_{i=1}^{M}$, such that*

$$\left| \hat{w}_2 \phi(\langle \hat{w}_1, x - \hat{x}^{(1)} \rangle) - \sum_{i=1}^{M} \tilde{w}_i \left[ \psi\left( \left\langle w_i, x - x_i^{(1)} \right\rangle - s_2 \right) + 1 \right] \right| < \epsilon,$$

*where $\hat{w}_1 = \frac{\hat{x}^{(2)} - \hat{x}^{(1)}}{\|\hat{x}^{(2)} - \hat{x}^{(1)}\|}$, $\tilde{w}_i \in \mathbb{R}$, and $w_i = s_1 \frac{x_i^{(2)} - x_i^{(1)}}{\|x_i^{(2)} - x_i^{(1)}\|^2}$.*

*Proof.* Let $\epsilon > 0$ and, w.l.o.g., $\hat{w}_2 > 0$. Let $B = \{\langle \hat{w}_1, x \rangle \colon x \in \mathcal{X}\}$, as well as $\hat{f}(x) = \hat{w}_2 \phi(\langle \hat{w}_1, x - \hat{x}^{(1)}\rangle)$. We start by partitioning $\hat{f}$ into $\epsilon/4$-chunks. More specifically, let $c = \max B$, and $M' = \left\lceil \frac{4\hat{w}_2 |c|}{\epsilon} \right\rceil$. Set $d_k = \frac{(k-1)c}{M'}$, for $k = 1, 2, \ldots, M', M'+1$. We will now define points $c_j$, with the goal of constructing a function $f$ as in Lemma 6. Still, because we require $c_j \in B$ to define biases in our tanh functions later, we must define the $c_j$s iteratively, by setting $c_1 = 0$, $k = j = 2$, and define every other $c_j$ as follows:

1. If $d_k = c$, we are done, otherwise proceed to 2.

2. Set
$$d_k' = \sup B \cap [c_{j-1}, d_k] \quad d_k'' = \inf B \cap [d_k, c].$$

3. If $d_k' = d_k''$, set $c_j = d_k$, and increase $j$ and $k$, and go to 1.

4. If $d_k' > c_{j-1}$, set $c_j = d_k'$, and increase $j$, otherwise discard the point $d_k'$.

5. Set $c_j = d_k''$, and increase $j$. Set $k = \arg\min\{d_k - c_j \colon d_k - c_j > 0\}$ and go to 1.

We have $M < 2 \cdot M'$ points, and can now construct the $a_j$s of $f$. For $j = 1, 2, \ldots, M$, with $c_{M+1} = c$, let

$$a_j = \begin{cases} \hat{f}(c_{j+1}) - \hat{f}(c_j), & c_{j+1} - c_j \leq \frac{c}{M'} \\ \hat{f}(\rho(c_j)) - \hat{f}(c_j), & \text{otherwise,} \end{cases}$$

where $\rho(c_j) = \arg\min\{d_k - c_j \colon d_k - c_j \geq 0 \text{ and } k = 1, 2, \ldots, M'+1\}$. Note that $0 < c - c_M \leq \frac{c}{M'}$, by Definition 5 and continuity of the inner product $\langle \hat{w}_1, \cdot - x^{(1)} \rangle$. We then construct $f$ as in Lemma 6, namely,

$$f(x) = \sum_{i=1}^{M} a_i \mathbf{1}_{[c_i, c_{M+1}]}(x).$$

Letting $C = \{[c_j, c_{j+1}] \colon j = 0, 1, \ldots, M$ and $c_{j+1} - c_j \leq \frac{c}{M'}\}$, it is easy to see

$$|f(x) - \hat{f}(x)| < \frac{\epsilon}{4},$$

for all $x \in \bigcup_{c' \in C} c'$. For any $x$ outside said set, we are not concerned with, as it is not part of $B$, and hence nothing from $\mathcal{X}$ is mapped to said points.

Construct $\omega = \{\omega_i\}_{i=1}^M$ according to Lemma 6. We will now construct a sum of tanh functions, using only weights/biases allowed in sampled networks. For all $i = 1, 2, \ldots, M$, define $\tilde{w}_i = \frac{a_i}{2}$ and set $x_i^{(1)} = \eta(x, \delta_i)$, with $\delta_i \geq 0$ and $x \in \mathcal{X}$, such that $\langle \hat{w}_1, x \rangle = c_i$ — where $\delta_i = 0$ iff $x \notin \partial \mathcal{X}$. We specify $\delta_i$ and $\epsilon_i' > 0$ such that $\frac{s_1}{\epsilon_i'} \geq \omega_i$, $\mathbb{B}_{2\epsilon_i'}(x_i^{(1)}) \subseteq \mathcal{X}$, and

$$\left| \psi(\langle w_i, x - x_i^{(1)} \rangle - s_2) - \psi(\langle w_i, x \rangle - c_i - s_2) \right| < \frac{\epsilon}{4|\tilde{w}_i|M},$$

with $w_i = s_1 \frac{x_i^{(2)} - x_i^{(1)}}{\|x^{(2)} - x^{(1)}\|^2}$, $x_i^{(2)} = x_i^{(1)} + \epsilon_i' \hat{w}_1$, and for all $x \in \mathcal{X}$. It is clear that $x_i^{(1)}, x_i^{(2)} \in \mathcal{X}$. We may now rewrite the sum of tanh functions as

$$
\begin{aligned}
g(x) &= \sum_{i=1}^M \tilde{w}_i \psi\left( \langle w_i, x - x_i^{(1)} \rangle - s_2 \right) + \tilde{w}_i \\
&= \sum_{i=1}^M \tilde{w}_i \psi\left( \frac{s_1}{\|\epsilon_i' \hat{w}_1\|} \left\langle \frac{\epsilon_i' \hat{w}_1}{\|\epsilon_i' \hat{w}_1\|}, x - x_i^{(1)} \right\rangle - s_2 \right) + \tilde{w}_i \\
&= \sum_{i=1}^M \tilde{w}_i \psi\left( \frac{s_1}{\epsilon_i'} \langle \hat{w}_1, x \rangle - \tilde{c}_i - s_2 \right) + \tilde{w}_i \\
&= \sum_{i=1}^M \frac{a_i}{2} \left[ \psi\left( \tilde{\omega}_i \langle \hat{w}_1, x \rangle - \tilde{c}_i - s_2 \right) + 1 \right],
\end{aligned}
$$

where $\tilde{c}_i = \langle \hat{w}_1, x_i^{(1)} \rangle$. As $\omega_i \leq \tilde{\omega}_i$ for all $i = 1, 2, \ldots, M$, it follows from Lemma 6 and the construction above that

$$
\begin{aligned}
|g(x) - \hat{f}(x)| &\leq \left| g(x) - \sum_{i=1}^M \frac{a_i}{2} \left[ \psi\left( \tilde{\omega}_i \langle \hat{w}_1, x \rangle - c_i - s_2 \right) + 1 \right] \right| \\
&\quad + \left| \sum_{i=1}^M \frac{a_i}{2} \left[ \psi\left( \tilde{\omega}_i \langle \hat{w}_1, x \rangle - c_i - s_2 \right) + 1 \right] - f(x) \right| \\
&\quad + |f(x) - \hat{f}(x)| \\
&< \frac{\epsilon}{4} + \frac{\epsilon}{2} + \frac{\epsilon}{4} = \epsilon.
\end{aligned}
$$

$\square$

We are now ready to prove that sampled networks with one hidden layer with tanh as activation function are universal approximators.

**Theorem 6.** *Let $\mathcal{F}_{1,\infty}^S$ be the set of all sampled networks with one hidden layer of arbitrary width and activation function $\psi$. $F_{1,\infty}^S$ is dense in $C(\mathcal{X}, \mathbb{R}^{N_2})$, with respect to the uniform norm.*

*Proof.* Let $g \in C(X, \mathbb{R}^{N_2})$, $\epsilon > 0$, and w.l.o.g., $N_2 = 1$. By Theorem 4, we know there exists a network $\Phi$, with $\hat{N}_1$ neurons and parameters $\{\hat{w}_{1,i}, \hat{w}_{2,i}, \hat{b}_{1,i}, \hat{b}_{2,i}\}_{i=1}^{\hat{N}_1}$, and ReLU as activation function, such that $\|\Phi - g\|_\infty < \frac{\epsilon}{2}$. We can then construct a new network $\Psi$, with $\psi$ as activation function, where for each neuron $\Phi^{(1,n)}$, we construct $M_i$ neurons in $\Psi$, according to Lemma 7, with $\frac{\epsilon}{2\hat{N}_1}$. Setting the biases in last layer of $\Psi$ based on $\Phi$, i.e., for every $i = 1, 2, \ldots, \hat{N}_1$, $b_{2,j} = \frac{\hat{b}_{2,i}}{M_i}$,

where $j = 1, 2, \ldots, M_i$. We then have, letting $N_1$ be the number of neurons in $\Psi$,

$$|\Psi(x) - \Phi(x)| = \left| \left( \sum_{i=1}^{N_1} w_{2,i} \Psi^{(1,i)}(x) - b_{2,i} \right) - \left( \sum_{i=1}^{\hat{N}_1} \hat{w}_{2,i} \Phi^{(1,i)}(x) - \hat{b}_{2,i} \right) \right|$$

$$= \left| \sum_{i=1}^{\hat{N}_1} \left( \sum_{j=1}^{M_i} w_{2,j} \psi(\langle w_{1,j}, x \rangle - b_{1,j}) \right) - \hat{w}_{2,i} \phi(\langle \hat{w}_{1,i}, x \rangle - \hat{b}_{1,i}) \right|$$

$$\leq \sum_{i=1}^{\hat{N}_1} \left| \left( \sum_{j=1}^{M_i} w_{2,j} \psi(\langle w_{1,j}, x \rangle - b_{1,j}) \right) - \hat{w}_{2,i} \phi(\langle \hat{w}_{1,i}, x \rangle - \hat{b}_{1,i}) \right|$$

$$< \sum_{i=1}^{\hat{N}_1} \frac{\epsilon}{2\hat{N}_1} = \frac{\epsilon}{2},$$

for all $x \in \mathcal{X}$. The last inequality follows from Lemma 7. This implies that

$$\|\Psi - g\|_\infty \leq \|\Psi - \Phi\|_\infty + \|\Phi - g\|_\infty < \frac{\epsilon}{2} + \frac{\epsilon}{2} = \epsilon,$$

and $F_{1,\infty}^S$ is dense in $C(\mathcal{X}, \mathbb{R}^{N_2})$. $\square$

## A.2 Barron spaces

Working with neural networks and sampling makes it very natural to connect our theory to Barron spaces [2, 20]. This space of functions can be considered a continuum analog of neural networks with one hidden layer of arbitrary width. We start by considering all functions $f \colon \mathcal{X} \to \mathbb{R}$ that can be written as

$$f(x) = \int_\Omega w_2 \phi(\langle w_1, x \rangle - b) d\mu(b, w_1, w_2),$$

where $\mu$ is a probability measure over $(\Omega, \Sigma_\Omega)$, with $\Omega = \mathbb{R} \times \mathbb{R}^D \times \mathbb{R}$. A Barron space $\mathcal{B}_p$ is equipped with a norm of the form,

$$\|f\|_{\mathcal{B}_p} = \inf_\mu \{ \mathbb{E}_\mu[|w_2|^p (\|w_1\|_1 + |b|)^p]^{1/p} \}, \quad 1 \leq p \leq \infty,$$

taken over the space of probability measure $\mu$ over $(\Omega, \Sigma_\Omega)$. When $p = \infty$, we have

$$\|f\|_{\mathcal{B}_\infty} = \inf_\mu \max_{(b, w_1, w_2) \in \text{supp}(\mu)} \{ |w_2|(\|w_1\|_1 + |b|) \}.$$

The Barron space can then be defined as

$$\mathcal{B}_p = \{ f \colon f(x) = \int_\Omega w_2 \phi(\langle w_1, x \rangle - b) d\mu(b, w_1, w_2) \text{ and } \|f\|_{\mathcal{B}_p} < \infty \}.$$

As for any $1 \leq p \leq \infty$, we have $\mathcal{B}_p = \mathcal{B}_\infty$, and so we may drop the subscript $p$ [20]. Given our previous results, we can easily show approximation bounds between our sampled networks and Barron functions.

**Theorem 7.** *Let $f \in \mathcal{B}$ and $\mathcal{X} = [0,1]^D$. For any $N_1 \in \mathbb{N}_{>0}$, $\epsilon > 0$, and an arbitrary probability measure $\pi$, there exist sampled networks $\Phi$ with one hidden layer, $N_1$ neurons, and ReLU activation function, such that*

$$\|f - \Phi\|_2^2 = \int_{\mathcal{X}} |f(x) - \Phi(x)|^2 d\pi(x) < \frac{(3 + \epsilon)\|f\|_{\mathcal{B}}^2}{N_1}.$$

*Proof.* Let $N_1 \in \mathbb{N}_{>0}$ and $\epsilon > 0$. By E et al. [20], we know there exists a network $\hat{\Phi} \in \mathcal{F}_{1,N_1}$, where $\hat{\Phi}(\cdot) = \sum_{i=1}^{N_1} \hat{w}_{2,i} \phi(\langle \hat{w}_{1,i}, \cdot \rangle - \hat{b}_{1,i})$, such that $\|f - \hat{\Phi}\|_2^2 \leq \frac{3\|f\|_{\mathcal{B}}^2}{N_1}$. By Theorem 4, letting

$\mathcal{X}' = [\epsilon_I, 1 - \epsilon_I]^D$, we know there exists a network $\Phi \in \mathcal{F}_{1,N_1}^S$, such that $\|\Phi - \hat{\Phi}\|_\infty < \sqrt{\frac{\epsilon\|f\|_{\mathcal{B}}^2}{N_1}}$. We do not need constant blocks, because $\mathcal{X}$ is connected. We then have

$$\|\Phi - f\|_2^2 \leq \int_{\mathcal{X}} |\Phi(x) - \hat{\Phi}(x)|^2 d\pi(x) + \int_{\mathcal{X}} |\hat{\Phi}(x) - f(x)|^2 d\pi(x)$$
$$< \frac{\epsilon\|f\|_{\mathcal{B}}^2}{N_1} \int_{\mathcal{X}} d\pi(x) + \frac{3\|f\|_{\mathcal{B}}^2}{N_1} = \frac{(3+\epsilon)\|f\|_{\mathcal{B}}^2}{N_1}.$$

$\square$

### A.3   Distribution of sampled networks

In this section we prove certain invariance properties for sampled networks and our proposed distribution. First, we define the distribution to sample the pair of points, or equivalently, the parameters. Note that $\mathcal{X}$ is now any compact subset of $\mathbb{R}^D$, as long as $\lambda_D(\mathcal{X}) > 0$, where $\lambda_D$ the $D$-dimensional Lebesgue measure.

As we are in a supervised setting, we assume access to values of the true function $f$, and define $\mathcal{Y} = f(\mathcal{X})$. We also choose a norm $\|\cdot\|_{\mathcal{Y}}$ over $\mathbb{R}^{N_{L+1}}$, and norms $\|\cdot\|_{\mathcal{X}_{l-1}}$ over $\mathbb{R}^{N_{l-1}}$, for each $l = 1, 2, \ldots, L$. For the experimental part of the paper, we choose the $L^\infty$ norm for $\|\cdot\|_{\mathcal{Y}}$ and for $l = 1, 2, \ldots, L$, we choose the $L^2$ norm for $\|\cdot\|_{\mathcal{X}_{l-1}}$. We also denote $\bar{N} = \sum_{l=1}^L N_l$ as the total number of neurons in a given network, and $\bar{N}_l = \sum_{k=1}^l N_k$. Due to the nature of sampled networks and because we sample each layer sequentially, we start by giving a more precise definition of the conditional density given in Definition 2. As a pair of points from $\mathcal{X}$ identifies a weight and bias, we need a distribution over $\mathcal{X}^{2\bar{N}}$, and for each layer $l$ condition on sets of $2\bar{N}_{l-1}$ points, which then parameterize the network $\Phi^{(l-1)}$ by constructing weights and biases according to Definition 4.

**Definition 7.** *Let $\mathcal{X}$ be compact, $\lambda_D(\mathcal{X}) > 0$, and $f \colon \mathbb{R}^D \to \mathbb{R}^{N_{L+1}}$ be Lipschitz-continuous w.r.t. the metric spaces induced by $\|\cdot\|_{\mathcal{Y}}$ and $\|\cdot\|_{\mathcal{X}}$. For any $l \in \{1, 2, \ldots, L\}$, setting $\epsilon = 0$ when $l = 1$ and otherwise $\epsilon > 0$, we define*

$$q_l^\epsilon\left(x_0^{(1)}, x_0^{(2)} \mid X_{l-1}\right) = \begin{cases} \dfrac{\|f(x_0^{(2)}) - f(x_0^{(1)})\|_{\mathcal{Y}}}{\max\{\|x_{l-1}^{(2)} - x_{l-1}^{(1)}\|_{\mathcal{X}_{l-1}}, \epsilon\}}, & x_{l-1}^{(1)} \neq x_{l-1}^{(2)} \\ 0, & otherwise, \end{cases}$$

*where $x_0^{(1)}, x_0^{(2)} \in \mathcal{X}$, $x_{l-1}^{(1)} = \Phi^{(l-1)}(x_0^{(1)})$, and $x_{l-1}^{(2)} = \Phi^{(l-1)}(x_0^{(2)})$, with the network $\Phi^{(l-1)}$ parameterized by pairs of points in $\mathcal{X}^{\bar{N}_{l-1}}$. Then, we define the integration constant $C_l = \int_{\mathcal{X} \times \mathcal{X}} q_l^\epsilon d\lambda$. The l-layered density $p_l^\epsilon$ is defined as*

$$p_l^\epsilon = \begin{cases} \frac{q_l^\epsilon}{C_l}, & if\ C_l > 0 \\ \frac{1}{\lambda_{2D}(\mathcal{X} \times \mathcal{X})}, & otherwise. \end{cases} \tag{6}$$

*Remark* 6. The added $\epsilon$ is there to ensure the density is bounded, but is not needed when considering the first layer, due to the Lipschitz assumption. Adding $\epsilon > 0$ for $l = 1$ is both unnecessary and affects equivariant/invariant results in Theorem 8. We drop the $\epsilon$ superscript wherever it is unambiguously included.

We can now use this definition to define the probability of the whole parameter space $\mathcal{X}^{2\bar{N}}$, i.e., given an architecture provide a distribution over all weights and biases the network require. Let $\bar{\mathcal{X}} = \mathcal{X}^{2\bar{N}}$, i.e., the sampling space of $P$ with the product topology, and $\bar{D} = 2\bar{N}D$ as the dimension of the space. Since all the parameters is defined through points of $\mathcal{X} \times \mathcal{X}$, we may choose an arbitrary ordering of the points, which means one set of weights and biases for the whole network can be written as $\{x_i^{(1)}, x_i^{(2)}\}_{i=1}^{\bar{N}}$.

**Definition 8.** *The probability distribution $\rho_f$ over $\bar{\mathcal{X}}$ have density $p$,*

$$p\left(x_1^{(1)}, x_1^{(2)}, x_2^{(1)}, x_2^{(2)}, \ldots, x_{\bar{N}}^{(1)}, x_{\bar{N}}^{(2)}\right) = \prod_{l=1}^L \prod_{i=1}^{N_l} p_l\left(x_{\bar{N}_{l-1}+i}^{(1)}, x_{\bar{N}_{l-1}+i}^{(2)} \mid X_{l-1}\right),$$

*with $X_{l-1} = \bigcup_{k=1}^{l-1} \bigcup_{j=1}^{N_k} \{x_{\bar{N}_k+j}^{(1)}, x_{\bar{N}_k+j}^{(2)}\}$.*

It is not immediately clear from above that $P$ is valid distribution, and in particular, that the density is integrable. This is what we show next.

**Proposition 1.** *Let $\mathcal{X} \subseteq \mathbb{R}^D$ be compact, $\lambda_D(\mathcal{X}) > 0$, and $f$ be Lipschitz continuous w.r.t. the metric spaces induced by $\|\cdot\|_\mathcal{Y}$ and $\|\cdot\|_\mathcal{X}$. For fixed architecture $\{N_l\}_{l=1}^L$, the proposed function $p$ is integrable and*

$$\int_{\bar{\mathcal{X}}} p \, d\lambda_{\bar{D}} = 1.$$

*It therefore follows that $P$ is a valid probability distribution.*

*Proof.* We will show for each $l = 1, 2, \ldots, L$ that $p_l$ is bounded a.e. and is nonzero for at least one subset with nonzero measure. There exist $A \subseteq \mathcal{X} \times \mathcal{X}$ such that $p_l(A) \neq 0$ and $\lambda_{2D}(A) > 0$, as either $C_l > 0$ or $p_l$ is the uniform density by Definition 7 and $\lambda_{2D}(\mathcal{X} \times \mathcal{X}) > 0$ by assumption and the product topology.

For $l = 1$, let $K_l > 0$ be the Lipschitz constant. Then $q_l(x^{(1)}, x^{(2)})$ by assumption of $f$ for all $x^{(1)}, x^{(2)} \in \mathcal{X}$. When $l > 1$, for all $X_{l-1} \in \mathcal{X}^{2\bar{N}_{l-1}}$, we have that there exist a constant $K_l > 0$ due to continuity and compactness, such that

$$p_l(x^{(1)}, x^{(2)} \mid X_l) \leq \max\left\{\frac{K_l}{\epsilon}, \frac{1}{\lambda_{2D}(\mathcal{X} \times \mathcal{X})}\right\}.$$

As $p$ is a multiplication of a finite set of elements, we end up with

$$0 < \int_{\bar{\mathcal{X}}} p \, d\lambda < \int_{\bar{\mathcal{X}}} \prod_{l=1}^L N_l \left(\frac{K_l}{\epsilon} + K_l + \frac{1}{\lambda_{2D}(\mathcal{X} \times \mathcal{X})}\right) d\lambda_{\bar{D}}$$

$$< L \max\left\{N_l \left(\frac{K_l}{\epsilon} + K_l + \frac{1}{\lambda_{2D}(\mathcal{X} \times \mathcal{X})}\right)\right\}_{l=1}^L \lambda_{\bar{D}}(\bar{\mathcal{X}}) < \infty.$$

using the fact that $0 < \lambda_D(\mathcal{X}) < \infty$ and $\lambda_{\bar{D}}$ being the product measure, implies $0 < \lambda_{\bar{D}}(\bar{\mathcal{X}}) < \infty$. Due to the normalization constants added to $q_l$, we see $p_l$ integrates to one. This means $P$ is a valid distribution of $\bar{\mathcal{X}}$, with implied independence between the neurons in the same layer. $\square$

One special property of sampled networks, and in particular of the distribution $P$, is their invariance under both linear isometries and translation (together forming the Euclidean group, i.e., rigid body transformations), as well as scaling. We denote the set of possible transformations as $\mathcal{H}(D) = \mathbb{R} \setminus \{0\} \times O(D) \times \mathbb{R}$, with $O(D)$ being the orthogonal group of dimension $D$. We then denote $(a, A, c) = H \in \mathcal{H}(D)$ as $H(x) = aAx + c$, where $x \in \mathcal{X}$. The space $\mathcal{H}_f(D) \subseteq \mathcal{H}$ are all transformations such that $f \colon \mathcal{X} \to \mathbb{R}^{N_{L+1}}$ is equivariant with respect to the transformations, with the underlying metric space given by $\|\cdot\|_\mathcal{Y}$. That is, for any $H \in \mathcal{H}_f(D)$, there exists a $H' \in \mathcal{H}(N_{L+1})$, such that $f(H(x)) = H'(f(x))$, where $x \in \mathcal{X}$, and the orthogonal matrix part of $H'$ is isometric w.r.t. $\|\cdot\|_\mathcal{Y}$. Note that often the $H'$ will be the identity transformation, for example by having the same labels for the transformed data. When $H'$ is the identity function, we say $f$ is invariant with respect to $H$. In the next result, we assume we choose norms $\|\cdot\|_{\mathcal{X}_0}$ and $\|\cdot\|_\mathcal{Y}$, such that the orthogonal matrix part of $H$ is isometric w.r.t. those norms and the canonical norm $\|\cdot\|$, as well as continue the assumption of Lipschitz-continuous $f$.

**Theorem 8.** *Let $H \in \mathcal{H}_f(D)$, $\Phi, \hat{\Phi}$ be two sampled networks with the same number of layers $L$ and neurons $N_1, \ldots, N_L$, where $\Phi \colon \mathcal{X} \to \mathbb{R}^{N_{L+1}}$ and $\hat{\Phi} \colon H(\mathcal{X}) \to \mathbb{R}^{N_{L+1}}$, and $f \colon \mathcal{X} \to \mathbb{R}^{N_{L+1}}$ is the true function. Then the following statements hold:*

*(1) If $\hat{x}_{0,i}^{(1)} = H(x_{0,i}^{(1)})$ and $\hat{x}_{0,i}^{(2)} = H(x_{0,i}^{(2)})$, for all $i = 1, 2, \ldots, N_1$, then $\Phi^{(1)}(x) = \hat{\Phi}^{(1)}(H(x))$, for all $x \in \mathcal{X}$.*

*(2) If $f$ is invariant w.r.t. $H$: $\Phi \in \mathcal{F}_{L,[N_1,\ldots N_L]}^S(\mathcal{X})$ if and only if $\hat{\Phi} \in \mathcal{F}_{L,[N_1,\ldots N_L]}^S(H(\mathcal{X}))$, such that $\Phi(x) = \hat{\Phi}(x)$ for all $x \in \mathcal{X}$.*

*(3) The probability measure $P$ over the parameters is invariant under $H$.*

*Proof.* Let $H = (a, A, c)$. Assume we have sampled $\hat{x}_{0,i}^{(1)} = H(x_{0,i}^{(1)})$ and $\hat{x}_{0,i}^{(2)} = H(x_{0,i}^{(2)})$, for all $i = 1, 2, \ldots, N_1$. The points sampled determines the weights and biases in the usual way, giving

$$
\begin{aligned}
\phi\left(\left\langle w_{1,i}, x - x_{0,i}^{(1)}\right\rangle\right) &= \phi\left(\left\langle \frac{x_{0,i}^{(2)} - x_{0,i}^{(1)}}{\|x_{0,i}^{(2)} - x_{0,i}^{(1)}\|^2}, x - x_{0,i}^{(1)}\right\rangle\right) \\
&= \phi\left(\left\langle A\frac{x_{0,i}^{(2)} - x_{0,i}^{(1)}}{\|A(x_{0,i}^{(2)} - x_{0,i}^{(1)})\|^2}, A(x - x_{0,i}^{(1)})\right\rangle\right) \\
&= \frac{a \cdot a}{a^2}\phi\left(\left\langle A\frac{x_{0,i}^{(2)} - x_{0,i}^{(1)}}{\|A(x_{0,i}^{(2)} - x_{0,i}^{(1)})\|^2}, A(x - x_{0,i}^{(1)})\right\rangle\right) \\
&= \phi\left(\left\langle aA\frac{x_{0,i}^{(2)} - x_{0,i}^{(1)}}{\|aA(x_{0,i}^{(2)} - x_{0,i}^{(1)})\|^2}, aA(x - x_{0,i}^{(1)})\right\rangle\right) \\
&= \phi\left(\left\langle aA\frac{x_{0,i}^{(2)} + c - x_{0,i}^{(1)} - c}{\|aA(x_{0,i}^{(2)} + c - x_{0,i}^{(1)} - c)\|^2}, aA(x + c - x_{0,i}^{(1)} - c)\right\rangle\right) \\
&= \phi\left(\left\langle \frac{H(x_{0,i}^{(2)}) - H(x_{0,i}^{(1)})}{\|H(x_{0,i}^{(2)}) - H(x_{0,i}^{(1)})\|^2}, H(x) - H(x_{0,i}^{(1)})\right\rangle\right) \\
&= \phi\left(\left\langle \hat{w}_{1,i}, \hat{x} - \hat{x}_{0,i}^{(1)}\right\rangle\right)
\end{aligned}
$$

for all $x \in \mathcal{X}$, $\hat{x} \in H(\mathcal{X})$, and $i = 1, 2, \ldots, N_1$. Which implies that (1) holds.

Assuming $f$ is invariant w.r.t. $H$, then for any $\Phi \in \mathcal{F}_{L,[N_1,\ldots N_L]}^S(\mathcal{X})$, let $X = \{x_{1,i}^{(1)}, x_{1,i}^{(2)}\}_{i=1}^{N_1}$, we can then choose $H(X)$ as points to construct weights and biases in the first layer of $\hat{\Phi}$, and by (1), we have $\mathcal{X}_1 = \Phi^{(1)}(\mathcal{X}) = \hat{\Phi}^{(1)}(H(\mathcal{X})) = \hat{\mathcal{X}}_1$. As the sampling space is the same for the next layer, we see that we can choose the points for the weights and biases of $\hat{\Phi}$, such that $\mathcal{X}_l = \hat{\mathcal{X}}_l$, where $l = 1, 2, \ldots, L$. As the input after the final hidden layer is also the same, by the same argument, means the weights in the last layer must be the same, due to the loss function $\mathcal{L}$ in Definition 4 is the same due to the invariance assumption. Thus, $\Phi(x) = \hat{\Phi}(H(x))$ for all $x \in \mathcal{X}$. As $H$ is bijective, means the same must hold true starting with $\hat{\Phi}$, and constructing $\Phi$, and we conclude that (2) holds.

For any distinct points $x^{(1)}, x^{(2)} \in \mathcal{X}$, letting $(a', A', c')$ be the set such that $g(aAx + c) = a'A'g(x) + c'$, and $\hat{C}_l, C_l$ be the normalization constants over the conditional density $p_l$, for $H(\mathcal{X})$ and $\mathcal{X}$ resp. We have for the conditional density when $l = 1$ is,

$$
\begin{aligned}
p_1(H(x^{(1)}), H(x^{(2)})) &= \frac{1}{\hat{C}_1}\frac{\|f(aAx^{(2)} + c) - f(aAx^{(1)} + c)\|_{\mathcal{Y}}}{\|aAx^{(2)} + c - aAx^{(1)} - c\|_{\mathcal{X}}} \\
&= \frac{1}{\hat{C}_1}\frac{\|a'A'f(x^{(2)}) + c' - a'A'f(x^{(1)}) - c'\|_{\mathcal{Y}}}{\|aAx^{(2)} - aAx^{(1)}\|_{\mathcal{X}}} \\
&= \frac{|a'|}{|a|\hat{C}_1}\frac{\|f(x^{(2)}) - f(x^{(1)})\|_{\mathcal{Y}}}{\|x^{(2)} - x^{(1)}\|_{\mathcal{X}}} = \frac{|a'|}{|a|\hat{C}_1}p_1(x^{(1)}, x^{(2)}).
\end{aligned}
$$

With similar calculations, we have

$$
\frac{|a|}{|a'|}\hat{C}_1 = \frac{|a|}{|a'|}\int_{\mathcal{X}\times\mathcal{X}}p_1(H(x), H(z))dxdz = \frac{|a|}{|a'|}\frac{|a'|}{|a|}\int_{\mathcal{X}\times\mathcal{X}}p_1(x, z)dxdz = C_1.
$$

Hence, the conditional probability distribution for the first layer is invariant under $H$, and then by (1) and (2), the possible sampling spaces are equal for the following layers, and therefore the conditional distributions for each layer is the same, and therefore $P$ is invariant under $H$. $\qquad\square$

*Remark* 7. Point (2) in the theorem requires $f$ to be invariant w.r.t. $H$. This is due to the fact that the parameters in the last layer minimizes the implicitly given loss function $\mathcal{L}$, seen in Definition 4. We have rarely discussed the loss function, as it depends on what function we are approximating.

Technically, (2) also holds as long as the loss function is not altered by $H$ in a way that the final layer alters, but we simplified it to be invariant, as this is also the most likely case to stumble upon in practice. The loss function also appears in the proofs given in Appendix A.1 and Appendix A.2. Here both uses, implicitly, the loss function based on the uniform norm.

# B Illustrative example: approximating a Barron function

All computations for this experiment were performed on the Intel Core i7-7700 CPU @ 3.60GHz × 8 with 32GB RAM.

We compare random Fourier features and our sampling procedure on a test function for neural networks [64]: $f(x) = \sqrt{3/2}(\|x - a\|_{\ell^2} - \|x + a\|_{\ell^2})$, with the constant vector $a \in \mathbb{R}^d$ defined by $a_j = 2j/d - 1$. For all dimensions $d$, we sample 10,000 points uniformly at random in the cube $[-1, 1]^d$ for training (i.e., sampling and solving the last, linear layer problem), and another 10,000 points for evaluating the error. We re-run the same experiment with five different random seeds, but with the same train/test datasets. This means the random seed only influences the weight distributions in sampled networks and random feature models, not the data. Figure 8 shows the relative $L^2$ error in the full hyperparameter study, i.e. in dimensions $d \in \{1, 2, 3, 4, 5, 10\}$ (rows) and for tanh (left column) and sine activation functions (right column). For the random feature method, we always use sine activation, because we did not find any data-agnostic probability distributions for tanh. Figure 9 shows the fit times for the same experiments, demonstrating that sampling networks are not slower than random feature models. This is obvious from the complexity analysis in Appendix F, and confirmed experimentally here. Interesting observations regarding the full hyperparameter study are that for one dimension, $d = 1$, random feature models outperform sampled networks and can even use up to two hidden layers. In higher dimensions, and with larger widths / more layers, sampled networks consistently outperform random features. In $d = 10$, a sampled network with even a single hidden layer is about one order of magnitude more accurate than the same network with normally distributed weights. The convergence rate of the error of sampled networks with respect to increasing layer widths is consistent over all dimensions and layers, even sometimes outperforming the theoretical bound of $O(N^{-1/2})$.

# C Classification benchmark from OpenML

The Adam training was done on the GeForce 4x RTX 3080 Turbo GPU with 10 GB RAM, while sampling was performed on the Intel Core i7-7700 CPU @ 3.60GHz × 8 with 32GB RAM.

We use all 72 datasets in the "OpenML-CC18 Curated Classification benchmark" from the "OpenML Benchmarking Suites" (CC-BY) [4], which is available on openml.org: `https://openml.org/search?type=benchmark&sort=tasks_included&study_type=task&id=99`.

We use the OpenML Python API (BSD-3 Clause) [22] to download the benchmark datasets. The hyperparameters used in the study are listed in Table 3. From all datasets, we use at most 5000 points. This was done for all datasets before applying any sampled or gradient-based approach, because we wanted to reduce the training time when using the Adam optimizer. We pre-process the input features using one-hot encoding for categorical variables, and robust whitening (removal of mean, division by standard deviation using the `RobustScaler` class from `scikit-learn`). Missing features are imputed with the median of the feature. All layers of all networks always have $500$ neurons. For Adam, we employ early stopping with `patience=3`, monitoring the loss value. The least squares regularization constant for sampling is $10^{-10}$. We split the data sets for 10-fold cross-validation using stratified k-fold (`StratifiedKFold` class from `scikit-learn`), and report the average of the accuracy scores and fit times over the ten folds (Figure 4 in the main paper).

# D Deep neural operators

## D.1 Dataset

To generate an initial condition $u_0$, we first sample five Fourier coefficients for the lowest frequencies. We sample both real and imaginary parts from a normal distribution with zero mean and standard

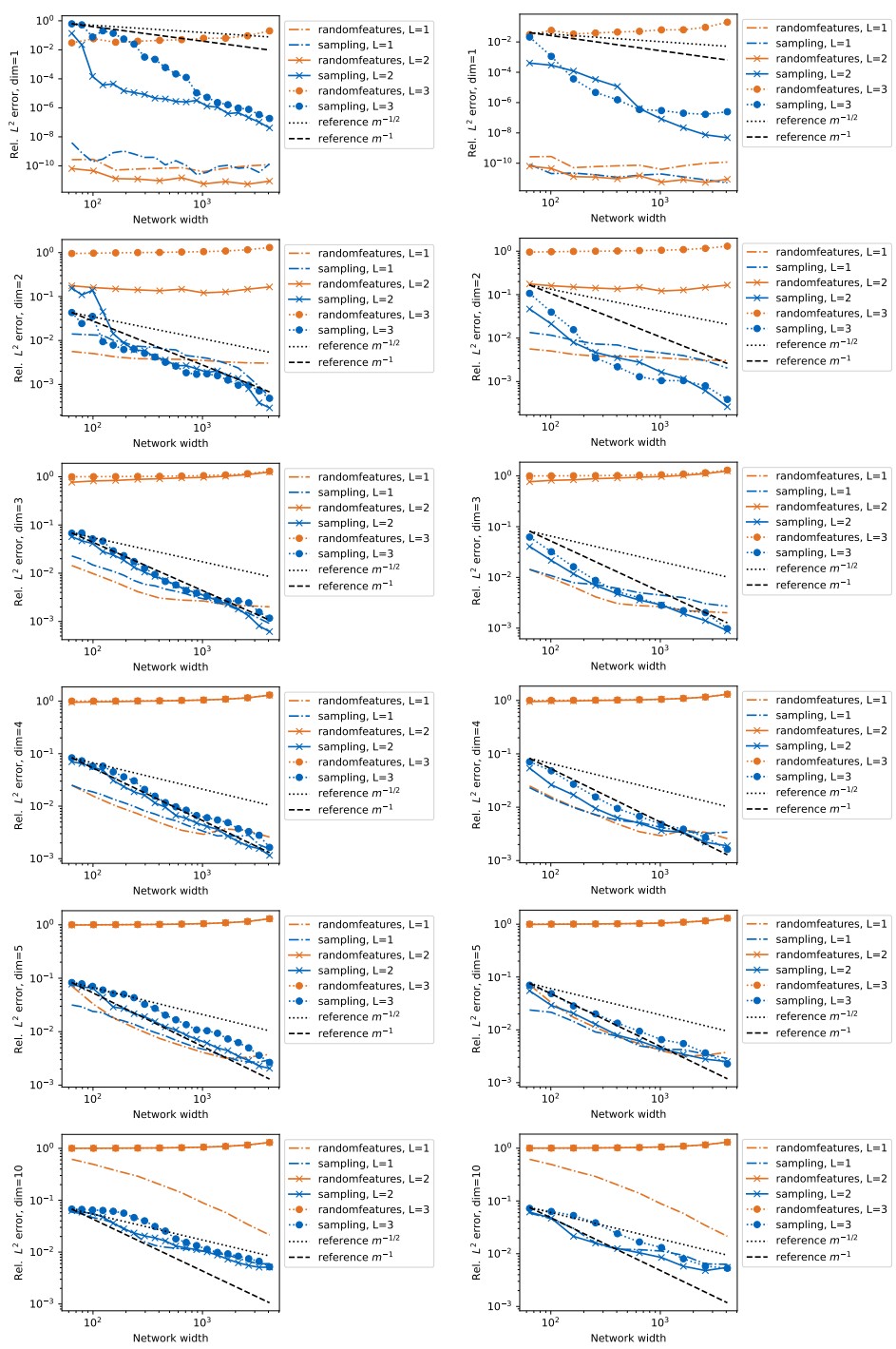

Figure 8: Relative $L^2$ approximation error for random features (using sine activation) and sampling (left: using tanh activation, right: using sine activation), with input dimensions $d = 1, 2, 3, 4, 5$ and 10. Results are averaged over five runs with different random seeds.

deviation equal to five. We then scale the $k$-th coefficient by $1/(k + 1)^2$ to create a smoother initial condition. For the first coefficient corresponding to the zero frequency, we also set the imaginary part to zero as our initial condition should be real-valued. Then, we use real inverse Fourier transform with orthonormal normalization (`np.fft.irrft`) to generate an initial condition with the

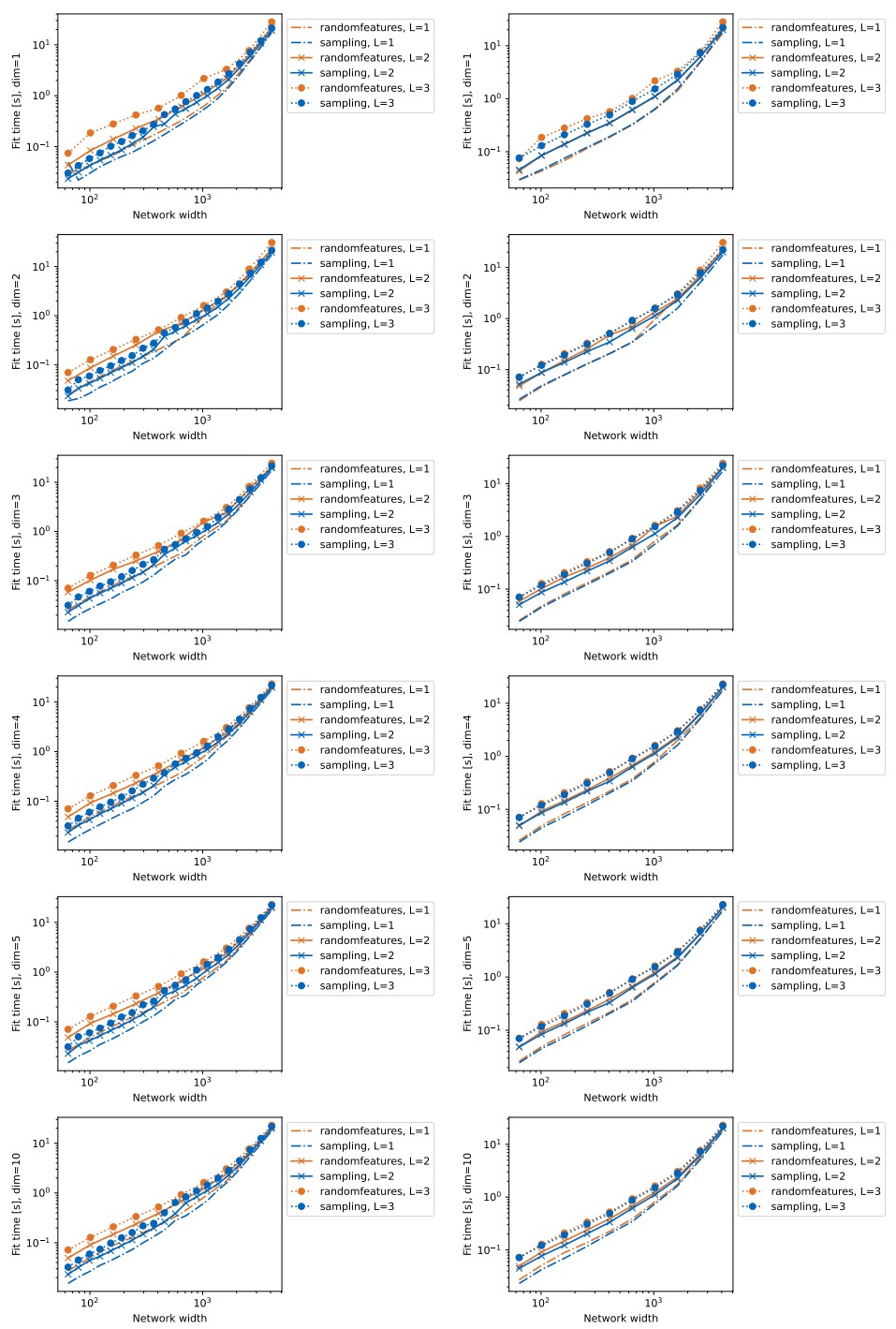

Figure 9: Fit times in seconds for random features (using sine activation) and sampling (left: using tanh activation, right: using sine activation), with input dimensions $d = 1, 2, 3, 4, 5$ and $10$. Results are averaged over five runs with different random seeds.

discretization over 256 grid points. To solve the Burgers' equation, we use a classical numerical solver (`scipy.integrate.solve_ivp`) and obtain a solution $u_1$ at $t = 1$. This way, we generate 15000 pairs of $(u_0, u_1)$ and split them between train (9000 pairs), validation (3000 pairs), and test sets (3000 pairs).

Table 3: Network hyperparameters used in the OpenML benchmark study.

|  | Parameter | Value |
|---|---|---|
| Shared | Number of layers | $[1, 2, 3, 4, 5]$ |
|  | Layer width | 500 |
|  | Activation | tanh |
| Sampling | $L^2$-regularization | $10^{-5}$ |
| Adam | Learning rate | $10^{-3}$ |
|  | Max. epochs | 100 |
|  | Early stopping patience | 3 |
|  | Batch size | 64 |
|  | Loss | mean-squared error |

Table 4: Network hyperparameters used to train deep neural operators.

|  | Parameter | Values | Note |
|---|---|---|---|
| Shared | Number of layers | $[1, 2, 4]$ |  |
|  | Layer width | $[256, 512, 1024, 2048, 4096]$ |  |
|  | Number of modes | $8, 16, 32$ |  |
|  | Number of channels | $16, 32$ |  |
|  | Activation | tanh |  |
| Sampling | $L^2$-regularization | $10^{-10}, 10^{-8}, 10^{-6}$ |  |
| Adam | Learning rate | $10^{-5}, 5 \cdot 10^{-5}, 10^{-4}, 5 \cdot 10^{-4}$ | FCNs and POD-DeepONet |
|  |  | $10^{-4}, 5 \cdot 10^{-4}, 10^{-3}, 5 \cdot 10^{-3}$ | FNO |
|  | Weight decay | 0 |  |
|  | Max. epochs | 2000 | FNO, FCN in signal space |
|  |  | 5000 | FCN in Fourier space |
|  |  | 90000 | POD-DeepONet |
|  | Patience | 100 | FNO, FCN in signal space |
|  |  | 200 | FCN in Fourier space |
|  |  | 4500 | POD-DeepONet |
|  | Batch size | 64 | FCNs and FNO |
|  |  | full data | POD-DeepONet |
|  | Loss | mean-squared error | FCNs and POD-DeepONet |
|  |  | mean relative $H^1$-loss | FNO |

## D.2 Experiment setup

The Adam training was done on the GeForce 4x RTX 3080 Turbo GPU with 10 GB RAM, while sampling was performed on the 1x AMD EPYC 7402 @ 2.80GHz × 8 with 256GB RAM. We use the `sacred` package (https://github.com/IDSIA/sacred, MIT license, [37]) to conduct the hyperparameter search.

We perform a grid search over several hyperparameters (where applicable): the number of layers/Fourier blocks, the width of layers, the number of hidden channels, the regularization scale, the learning rate, and the number of frequencies/modes to keep. The full search grid is listed in Table 4.

**Adam** With the exception of FNO, we use the mean squared error as the loss function when training with Adam. For FNO, we train with the default option, that is, with the mean relative $H^1$-loss. This loss is based on the $H^1$-norm for a one-dimensional function $f$ defined as

$$\|f\|_{H^1} = \|f\|_{L^2} + \|f'\|_{L^2}.$$

For all the experiments, we use the mean relative $L^2$ error as the validation metric. We perform an early stopping on the validation set and restore the model with the best metric.

**DeepONet** We use the `deepxde` package (https://github.com/lululxvi/deepxde, LGPL-2.1 License, [45]) to define and train POD-DeepONet for our dataset. After defining the model, we apply an output transform by centering and scaling the output with $1/p$, where $p$ is the number of modes we

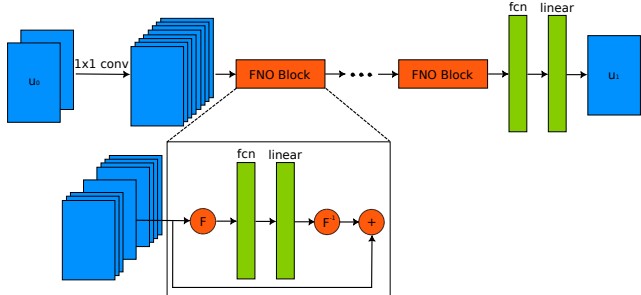

Figure 10: The architecture of the sampled FNO.

keep in the POD. To compute the POD, we first center the data and then use `np.linalg.eigh` to obtain the modes. We add two callbacks to the standard training: the early stopping and the model checkpoint. We use the default parameters for the optimization and only set the learning rate. The default training loop uses the number of iterations instead of the number of epochs, and we train for 90000 iterations with patience set to 4500. By default, POD-DeepONet uses all the data available as a batch. We ran several experiments with the batch size set to 64, but the results were significantly worse compared to the default setting. Hence, we stick to using the full data as one batch.

**FNO**   We use the `neuralop` (https://github.com/neuraloperator/neuraloperator, MIT license, [42, 38]) package with slight modifications. We changed the `trainer.py` to add the early stopping and to store the results of experiments. Apart from these changes, we use the functionality provided by the library to define and run experiments. We use batch size 64 and train for 2000 epochs with patience 100. In addition to the hyperparameters considered for other architecture, we also did a search over the number of hidden channels (32 or 64). We use the same number of channels for lifting and projection operators.

**FCN**   For both fully-connected networks in our experiments, we use PyTorch framework to define the models. For training a network in Fourier space, we prepend a forward Fourier transform before starting the training. During the inference, we applied the inverse Fourier transform to compute the validation metric. We considered using the full data as a batch, but several experiments indicated that this change worsens the final metric. Hence, we use batch size 64 for both architectures.

**Sampling FNO**   Here, we want to give more details about sampled FNO. First, we start with input $u_0$ and append the coordinate information to it as an additional channel. Then, we lift this input to a higher dimensional channel space by drawing $1 \times 1$ convolution filters from a normal distribution. Then, we apply a Fourier block which consists of fully-connected networks in Fourier space for each channel. After applying the inverse Fourier transform, we add the input of the block to the restored signal as a skip connection. After applying several Fourier blocks, we learn another FCN in signal space to obtain the final prediction. We can see the schematic depiction of the model in Figure 10. We note that to train fully-connected networks in the Fourier space, we apply Fourier transform to labels $u_1$ and use it during sampling as target functions. Similarly to the Adam-trained FNO, we search over the number of hidden channels we use for lifting.

### D.3   Results

We can see the results of the hyperparameter search in Figure 11. We see that sampled FNO and FCN in Fourier space perform well even with a smaller number of frequencies available, while DeepONet requires more modes to achieve comparable accuracy. We also see that adding more layers is not beneficial for sampled networks. A more important factor is the width of the layers: all of the sampled networks achieved the best results with the largest number of neurons.

For the Adam-trained models, we note that we could not significantly improve the results by increasing the widths of the layers. Possibly, this is due to a limited range of learning rates considered in the experiments.

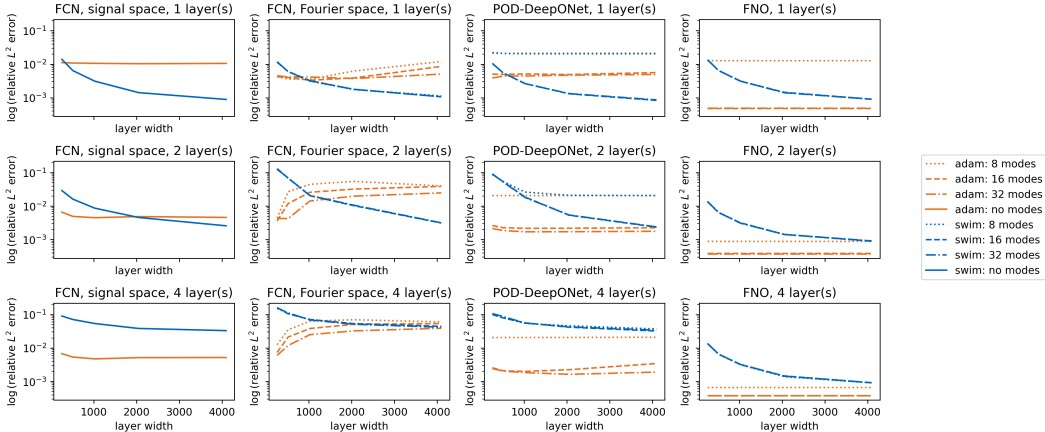

Figure 11: Comparison of different deep neural operators trained with Adam and with SWIM algorithm. FNO with Adam has no dependency on a layer width. Thus, we show results for FNO trained with Adam as a straight line in all rows. For the hyperparameters not present in the plot, we chose the values that produce the lowest error on the validation set. We repeat each experiment three times and average the resulting metrics for the plotting. The error is computed on the validation set.

# E  Transfer learning

This section contains details about the results shown in the main paper and the supporting hyper-parameter search concerning the transfer learning task. The source and target tasks of transfer learning, train-test split, and pre-trained models are the same as in the main paper. We describe the software, hardware, data pre-processing, and details of the sampling and the Adam training approach. Note that the results shown here are not averaged over five runs, as was done for the plots in the main paper.

**Software and hardware**    We performed all the experiments concerning the transfer learning task in Python, using the TensorFlow framework. We used a GeForce 4x RTX 3080 Turbo GPU with 10 GB RAM for the Adam training and a 1x AMD EPYC 7402 socket with 24 cores @ 2.80GHz for sampling. We use the pre-trained models from Keras Applications which are licensed under the Apache 2.0 License which can be found at `https://github.com/keras-team/keras/blob/v2.12.0/LICENSE`.

**Pre-processing of data**    The size of images in the ImageNet data set is larger than the ones in CIFAR-10, so we use a bicubic interpolation to upscale the CIFAR-10 images to dimensions $224 \times 224 \times 3$. We pre-process the input data for each model the same way it was pre-processed during the pre-training on ImageNet. Moreover, we apply some standard data augmentation techniques before training, such as vertical and horizontal shifts, rotation, and horizontal flips. After pre-processing, the images are first passed through the pre-trained classification layers and a global average pooling layer. The output of the global average pooling layer and classification labels serve as the input and output data for the classification head respectively.

**Details of the Adam training and sampling approaches**    We find the weights and biases of the hidden layer of the classification head using the proposed sampling algorithm and the usual gradient-based training algorithm.

First, in the Adam-training approach, we find the parameters of the classification head by iterative training with the Adam optimizer. We use a learning rate of $10^{-3}$, batch size of 32, and train for 20 epochs with early-stopping patience of 10 epochs. We store the parameters that yield the lowest loss on test data. We use the tanh activation function for the hidden layer, the softmax activation function for the output layer, and the categorical cross-entropy loss function with no regularization.

Second, in the sampling approach, we use the proposed sampling algorithm to sample parameters of the hidden layer of the classification head. We use the tanh activation function for the hidden layer.

Once the parameters of the hidden layers are sampled, an optimization problem for the coefficients of a linear output layer is solved. For this, the mean squared error loss function without regularization is used.

Unless mentioned otherwise, the above-mentioned setup is used for all the experiments in this section.

## E.1 Sampling Vs Adam training

Figure 12 compares the train and test accuracy for different widths (number of neurons in the hidden layer of the classification head) using three pre-trained neural network architectures for the Adam training and sampling approaches. As in the main paper, we observe that for all the models, the sampling approach results in a higher test accuracy than the Adam training approach for sufficiently higher widths.

The sampling algorithm tends to over-fit for very high widths. The loss on the train data for the iterative training approach decreases with width, particularly for the Xception network. This suggests that an extensive hyper-parameter search for the iterative training approach could yield higher classification accuracy on the train data. However, as shown in the main paper, the iterative training approach can be orders of magnitude slower than the sampling approach.

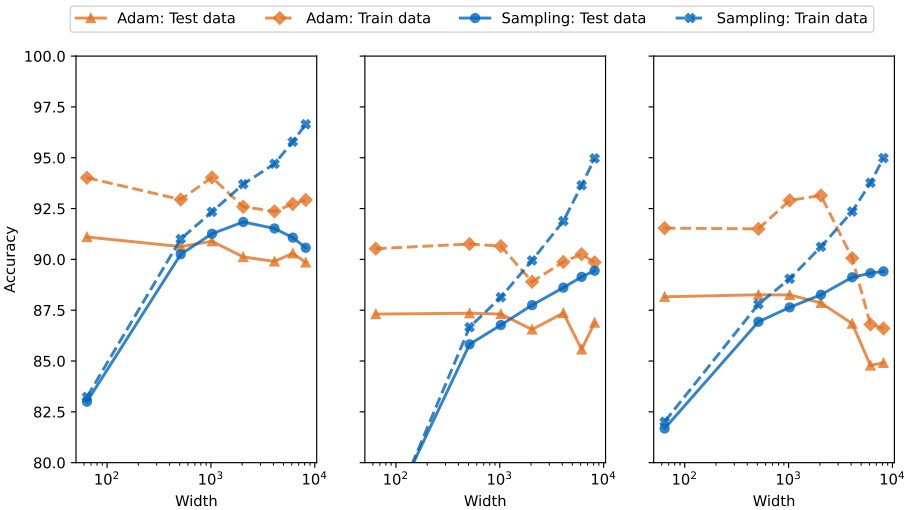

Figure 12: Left: ResNet50. Middle: VGG19. Right: Xception.

## E.2 Sampling with tanh Vs ReLU activation functions

This sub-section compares the performance of the sampling algorithm used to sample the weights of the hidden layer of the classification head for tanh and ReLu activation functions.

Figure 13 shows that for ResNet50, the test accuracy with tanh is similar to that obtained with ReLU for a width smaller than $2048$. As the width increases beyond 4096, test accuracy with ReLU is slightly better than with tanh. However, for VGG19 and Xception, test accuracy with tanh is higher than or equal to that with ReLU for all widths. Thus, we find the sampling algorithm with the tanh activation function yields better results for classification tasks than with ReLU.

On the training dataset, tanh and ReLU yield similar accuracies for all widths for ResNet50 and Xception. For VGG19, using the ReLU activation function yields much lower accuracies on the train and test data sets, especially as the width of the fully connected layer is increased. Thus, we observe that the tanh activation function is more suitable for classification tasks.

## E.3 Fine-tuning

There are two typical approaches to transfer learning: feature extraction and fine-tuning. In feature extraction, there is no need to retrain the pre-trained model. The pre-trained models capture the

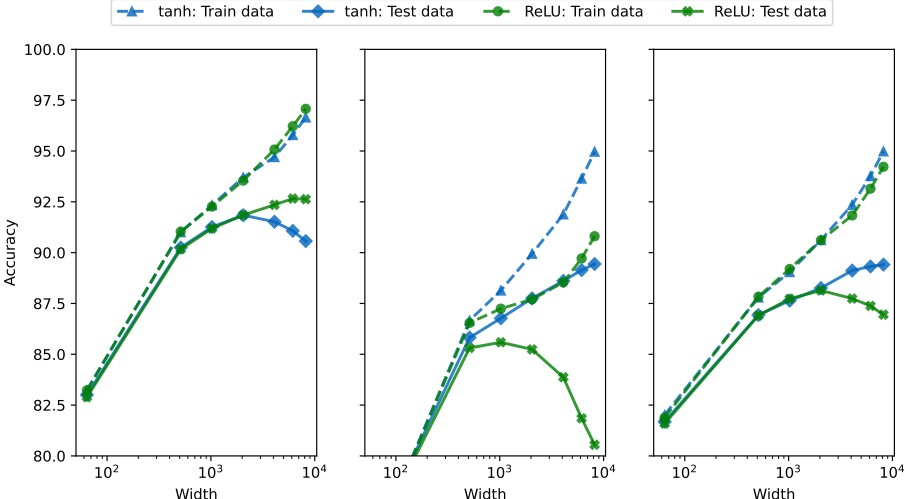

Figure 13: Left: ResNet50. Middle: VGG19. Right:Xception

essential representation/features from a similar task. One only needs to add a new classifier (one or more fully connected layers) and find the parameters of the new classifier.

In fine-tuning, after the feature extraction, some or all the parameters of the pre-trained model (a few of the last layers or the entire model) are re-trained with a much smaller learning rate using typical gradient-based optimization algorithms. Fine-tuning is not the focus of this work. Nevertheless, it is essential to verify that the weights of the last layer sampled by the proposed sampling algorithm can also be trained effectively in the fine-tuning phase. The results are shown in Figure 14.

In Figure 14, we observe that for certain widths (512, 1024, and 4096), sampling the last layer followed by fine-tuning the entire model yields slightly higher test accuracies than training the last layer followed by fine-tuning. For widths 2048, 6144, and 8192, for some models, sampling the last layer, followed by fine-tuning, is better; for others, training the last layer, followed by fine-tuning, is better.

Nevertheless, these experiments validate that the parameters of the last layer sampled with the proposed algorithm serve as a good starting point for fine-tuning. Moreover, the test accuracy after fine-tuning is comparable irrespective of whether the last layer was sampled or trained. However, as we show in the main paper, sampling the weights in the feature extraction phase takes much less time and gives better accuracy than training with the Adam optimizer for appropriate widths.

### E.4 One vs two hidden layers in the classification head

The goal of this sub-section is to explore whether adding an additional hidden layer in the classification head leads to an improvement in classification accuracy. We keep the same width for the extra layer in this experiment.

Figure 15 compares the train and test accuracies obtained with one and two hidden layers in the classification head for different widths.

Figure 15 (left) shows that unless one chooses a very high width with the sampling approach >= 6148, adding an extra hidden layer yields a lower test accuracy. On the train data, the sampling approach with 2 hidden layers results in lower accuracy for all widths in consideration.

Figure 15 (right) shows that with the Adam training approach, adding an extra hidden layer yields a lower test accuracy for all the widths in consideration. For lower widths, adding more layers results in over-fitting. We believe that the accuracy on the train data for higher widths could be improved with an extensive hyper-parameter search. However, adding an extra layer increases the training time too.

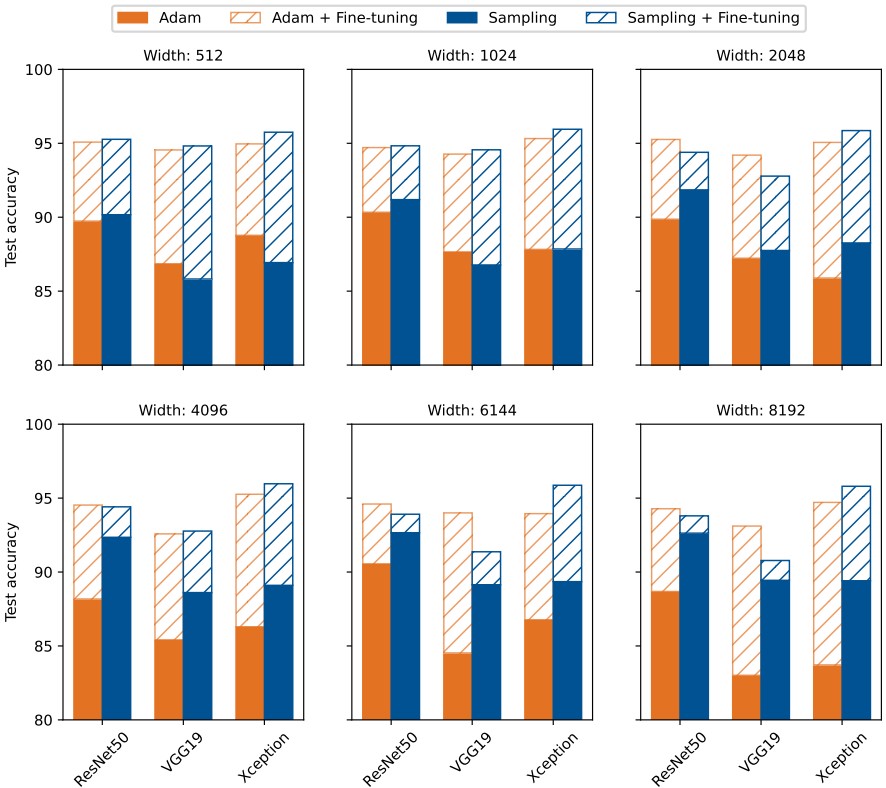

Figure 14: Comparison of test accuracies obtained by the sampling approach followed by fine-tuning and Adam training approach followed by fine-tuning for different widths

# F  Code repository and algorithm complexity

The code to reproduce the experiments from the paper can be found at

$$\texttt{https://gitlab.com/felix.dietrich/swimnetworks-paper.}$$

An up-to-date code base is maintained at

$$\texttt{https://gitlab.com/felix.dietrich/swimnetworks.}$$

Python 3.8 is required to run the computational experiments and install the software. Installation works using "`pip install -e .`" in the main code repository. The python packages required can be installed using the file `requirements.txt` file, using `pip`. The code base is contained in the folder `swimnetworks`, the experiments and code for hyperparameter studies in the folder `experiments`.

We now include a short complexity analysis of the SWIM algorithm (Algorithm 1 in the main paper), which is implemented in the code used in the experiments. The main points of the analysis can be found in Table 5.

There are two separate parts that contribute to the runtime. First, we consider sampling the parameters of the hidden layers. Letting $N = \max\{N_0, N_1, N_2, \ldots, N_L\}$, i.e., the maximum of the number of neurons in any given layer and the input dimension, and $M$ being the size of the training set. The size of the training set also limits the number of possible weights / biases that we can sample, namely $M^2$. The time complexity to sample the parameters of the hidden layers is $\mathcal{O}(L \cdot M(\min\{\lceil N/M \rceil, M\} + N^2))$. We can further refine the expression with $\mathcal{O}(\sum_{l=1}^{L} M \cdot (\min\{\lceil N_l/M \rceil, M\} + N_l \cdot N_{l-1}))$. Several factors contribute to this runtime. (1) The

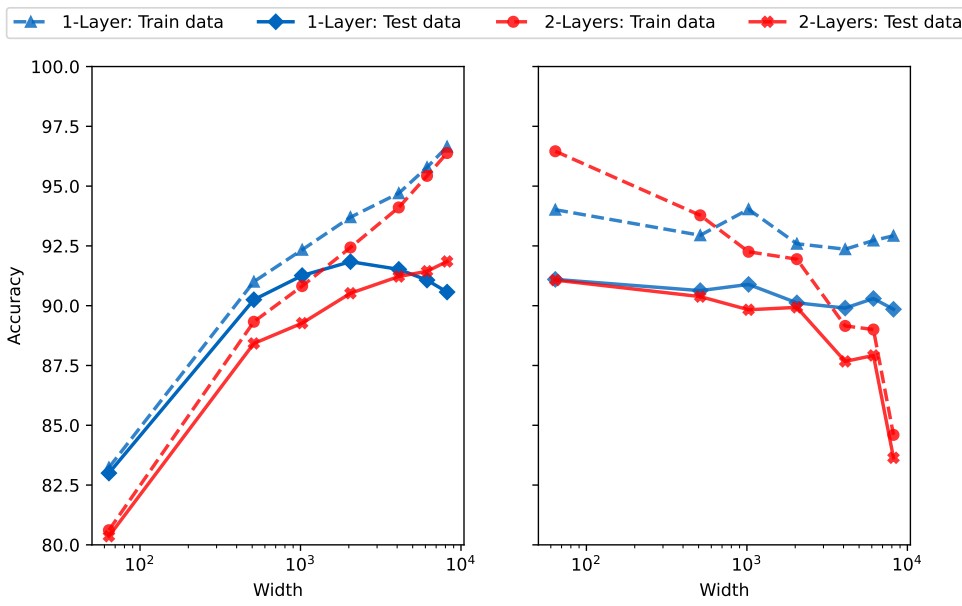

Figure 15: Left: Sampling, Right: Adam training

Table 5: Runtime and memory requirements for training a sampled neural networks with the SWIM algorithm, where $N = \max\{N_0, N_1, N_2, \ldots, N_L\}$. Assumption (I) assume the output dimension is less than or equal to $N$. Assumption (II) assumes in addition that $N < M^2$, i.e., number of neurons and input dimension is less than the size of dataset squared. Assumption (III) assumes a fixed architecture.

|  | Runtime | Memory |
|---|---|---|
| Assumption (I) | $\mathcal{O}(L \cdot M(\min\{\lceil N/M \rceil, M\} + N^2))$ | $\mathcal{O}(M \cdot \min\{\lceil N/M \rceil, M\} + LN^2)$ |
| Assumption (II) | $\mathcal{O}(L \cdot M(\lceil N/M \rceil + N^2))$ | $\mathcal{O}(M \cdot \lceil N/M \rceil + LN^2)$ |
| Assumption (III) | $\mathcal{O}(M)$ | $\mathcal{O}(M)$ |

number of layers increases the number of samples. (2) Computing the output after $l - 1$ layers for the entire dataset to compute the space from which we construct the weights, hence the term $N_l \cdot N_{l-1} \cdot M$. (3) Computing $M \cdot \tilde{M}_l$ probabilities, where $\tilde{M}_l = \min\{\lceil N_l/M \rceil, M\}$. When the size of the hidden layer is less than the number of training points — which is often the case — we compute $2 \cdot M$ probabilities — depending on a scaling constant. On the other hand, when the size of the layer is greater than or equal to $M^2$, we compute in worst case all possible probabilities, that is, $M^2$ probabilities. The last contributing factor is to sample $N_l$ pair of points, that in the expression is dominated by the term $N_l \cdot N_{l-1} \cdot M$. We are often interested in a fixed architecture, or at least to bound the number of neurons in each hidden layer to be less than the square of the number of training points, i.e., $N < M^2$. Adding the latter as an assumption, we end up with the runtime $\mathcal{O}(L \cdot M(\lceil N/M \rceil + N^2))$.

For the second part, we optimize the weights/biases $W_{L+1}, b_{L+1}$ to map from the last hidden layer to the output. Assume that we use the SVD decomposition to compute the pseudo-inverse and subsequently solve the least squares problem. The time complexity in general is then $\mathcal{O}(M \cdot N_L \cdot \min\{M, N_L\} + N_L \cdot M \cdot N_{L+1})$. Again, if we make the reasonable assumption that the number of training points is larger than the number of hidden layers, and that the output dimension is smaller than the dimension of the last hidden layer, we find $\mathcal{O}(MN_L^2)$, which can be rewritten to $\mathcal{O}(MN^2)$. With these assumptions in mind, the run time for the *full* training procedure, from start to finish, is $\mathcal{O}(L \cdot M(\lceil N/M \rceil + N^2))$, and when the architecture is fixed, we have $\mathcal{O}(M)$ runtime for the full training procedure.

In terms of memory, at every layer $l$, we need to store the probability matrix, the output of the training set when mapped through the previous $l - 1$ layers, and the number of points we sample. This means the memory required is $\mathcal{O}(M \cdot \lceil N_l/M \rceil + N_{L+1} \cdot N_L)$. In the last layer, we only need the image of the data passed through all the hidden layers, as well as the weights/biases, which leads to $\mathcal{O}(M + N_{L+1} \cdot N_L)$. We end up with the required memory for the SWIM algorithm is $\mathcal{O}(M \cdot \lceil N/M \rceil + LN^2)$.

