# OpenReview forum: "Sampling weights of deep neural networks"
_NeurIPS.cc/2023/Conference — NeurIPS 2023 poster_

### Official Review · Reviewer_b7pE · 2023-07-05

**Soundness:** 3 good
**Presentation:** 3 good
**Contribution:** 3 good
**Rating:** 6
**Confidence:** 3

**Summary:**

This article introduces an alternative to random features for sampling weights of neural networks. Their method relies on data points (both inputs and outputs) and activations to build iteratively the weights and biases of one layer after the other, in opposition with data-agnostic/purely random methods. It proves several results in term of function approximation by their sampled networks. Finally, they compare accuracy, training speed and size of model needed on a classification benchmark, an ODE approximation problem and a vision classification fine-tuning problem.

**Strengths:**

- I think the method is original, interesting and easy to understand.
- The method is more robust to depth than standard Random Features.

**Weaknesses:**

- The presentation of the paper is not perfect ( the algorithm is not easily readable, there is an indent missing for the for loop over $l$, the theory section is quite dense).
- The comparison against random features is only on a toy example. It is also not clear which algorithm is used on top on the RFs (linear regression, SGD,...)?
- The use in modern tasks and architectures seems very limited.
- In the experiments, it seems that we need quite wide networks to reach Adam networks.

**Questions:**

- I am not an expert in deep neural operators, but it is not clear to me how to use the method in the presence of a time-series? How to take into account the time-dependency of the data in the sampling process?
- It seems to me that this method may have a link with Bayesian neural networks (a field I am not an expert too), i.e. we have a distribution probability over layers. Could the authors comment on this, and maybe add a small paragraph in the text?
- The authors did not comment the case where we have outliers in the data? How would the method perform in that case? How easily these outliers would impact the target function? The method seems quite sensitive to them.

**Limitations:**

Limitations were addressed in the broader impact.
I am willing to modify my score accordingly to how the weaknesses and questions sections are addressed.

---

> ### Author Rebuttal · Authors · 2023-08-09
>
>
> **Regarding weaknesses:**
>  * [W1] We hope that an additional page after - potentially - accepting the paper can help to make Algorithm 1 more readable, and we can also add more explanations to the theoretical section.
>  * [W2] We also commented on a similar question in our answer to reviewer KCHD: In our experience, random feature models only perform comparatively well for layers with much more neurons - but we would be happy to extend the experiments to
> demonstrate it (there was not enough time in one week, unfortunately). For the last layer of random feature models, we use exactly the same algorithm (least squares, same regularization) as we use for the last layer in sampling. Only the hidden layer is sampled differently.
>  * [W3] We argue that many modern tasks involve supervised learning problems (surrogate modeling, learning classifiers and solving regression problems on tabular data, machine learning potentials for molecular dynamics), and we are currently experimenting with using the sampled weights as a "good basis" even for unsupervised tasks (modeling dynamical systems, solving partial differential equations). We demonstrate in Section 4.3 that more complicated architectures can be decomposed into smaller sequences of supervised learning problems as well, which can then be solved by the proposed algorithm without any modifications.
>
>     Furthermore, we argue that with our work we encourage constructing neural networks that use data much more directly to create their parameters. This may spur interesting ideas and extensions to different tasks and architectures, also the ones trained with gradient descent. This type of thinking also very easily leads to many ways to construct interesting densities and emphasizes a more probabilistic approach to neural networks that are also very efficient computationally. Interpretability of our networks is also a benefit that may help the acceptance of machine learning methods in practice.
>  * [W4] Indeed. The main reason wider networks are needed is that we randomly sample the hidden layers' weights and biases, so a certain amount of the neurons will not be very useful in the final layer. In followup work, we have started experimenting with $L^1$ optimization of the last layer (instead of $L^2$), which leads to many coefficients in the last layer being exactly zero. This means we can easily prune the sampled networks after solving the last layer, too, something we will explore in the future.
>
> **Regarding questions:**
>
>  * [Q1] The experiments presented in the paper do not use time series as input data. Models in Section 4.3 approximate an operator that transforms an initial condition into the solution at a particular time step. More specifically, the task’s input is function values at $t = 0$, while the target is function values at $t = 1$. Thus, we do not employ any time dependency in input data during sampling.
>
> If we had time-series data, we could attempt to learn the underlying dynamics of it, for example, an ordinary differential equation (ODE). To do this, we would compute finite differences (in time) in the input time series first, and then train a network to predict those in a supervised manner. This way, we could approximate the right-hand side of the underlying ODE and then solve it using classical integration methods.
>  * [Q2] The manuscript already contains a brief discussion in "related work" (lines 65-68). Note that we effectively connect training data points and weights, while Bayesian neural networks mostly use distributions over the weights without a direct connection to the data. We have expanded our discussion in the manuscript based on the reviewer's question.
>  * [Q3] Indeed, analyzing the behavior of the algorithm in a stochastic setting, or with measurement noise, was left for future work. Experiments in Section 4.2 (OpenML) and Section 4.4 (transfer learning) demonstrate empirically that the method is not very sensitive to noise: we use real data for the experiments, which always contains a certain noise level. If we would find a specific sensitivity to noise in a certain setting, it would be possible to study robust methods for linear systems (e.g., robust least-squares) as a replacement for the current standard least-squares method we use in the last layer. Sampling hidden weights and biases may also be affected by noise (mostly by poor function values, like wrong labels, for example). This would cause a less optimal selection of internal weights, which may still be mitigated by more robust methods for the last layer.

---

> > ### Comment · Reviewer_b7pE · 2023-08-12
> > **Answer to rebuttal**
> >
> > I would like to thank the authors for their answer. I think this paper is an interesting proof of concept, and that the experiments shown here are interesting. I am however left with the question on whether this method would scale to real tasks.
> > I think this line of work is promising and worth exploring, and I am therefore increasing my rating from 5 to 6.

---

> > > ### Author Response · Authors · 2023-08-16
> > >
> > > Thank you!
> > > Regarding the question on scaling to real tasks: We interpret this as a question on how our algorithm scales to large data sets, as they are enountered in real, big-data settings. Section F in the supplemental material contains a complexity analysis of the algorithm. It is mostly based on complexity analysis for solving linear systems. For a fixed network, the convergence to a solution is linear in the number of training data points, which is not worse than classical neural network training and should be sufficient for big-data settings.

---

### Official Review · Reviewer_KCHD · 2023-07-05

**Soundness:** 3 good
**Presentation:** 2 fair
**Contribution:** 3 good
**Rating:** 6
**Confidence:** 2

**Summary:**

The paper proposes a novel approach and analysis to sample weights of neural networks that can potentially address backprop limitations.
The method is based on computing differences between data points and can be scaled to deep networks (by computing the difference of data point activations). The paper introduces a rigor mathematical formulation supported by several experiments showing some benefits of the approach compared to backprop.

**Strengths:**

1. The problem of sampling weights better than using a data-agnostic random distribution is very interesting and solving it might have a lot of practical implications.
2. The method is supported by rigor mathematical formulation.
3. The experiments are extensive and diverse, including experiments with large networks, showing some benefits of the method.

**Weaknesses:**


1. In L24, it says "we introduce a data-driven sampling scheme to construct weights and biases close to gradients". However, in Section 3, the connection of the proposed approach to gradients is missing. Is the difference between data points related to gradients? In that sense, is there a connection of this submission with Forward Gradient approaches, e.g. "Gradients without Backpropagation. Baydin et al., 2022." or "Learning by Directional Gradient Descent. Silver et al., 2022"? In Forward Gradient, the gradients are often estimated by perturbing the inputs/weights a little bit. Even if it's directly related, I believe since this submission and Forward Gradient papers are both alternative approaches to backprop, it should be discussed at least in Related Work.

2. An important baseline for the Fig. 3 and 5 experiments that is missing would be to keep the first layers initialized randomly, while still apply arg min L for the last layer. This baseline would show more clearly the benefit of sampling weights. It may be that the main benefit is coming from solving argmin L.

3. It's a bit unclear why the proposed approach scales poorly with width (Fig 5, right). It seems that in Algorithm 1 most of the loops, specifically for k=1,2,...N_l, can be run in parallel for all neurons, so it should scale well with width if implemented efficiently. Perhaps, the comparison to Adam is not very fair as the authors are probably using a very efficient Adam implementation.

4. In algorithm 1, ||y_i - y_j|| must be a constant (same for all i,j) for classification problems (assuming y is a one hot vector and i, j are of different classes), so it's not very clear what's the purpose of this term. If labels are not that important, it could be beneficial for the paper to claim that the approach works without the need of labels (which are often expensive to obtain), perhaps except for the last layer.

5. Some visualizations of sampled vs trained weights would be useful. In particular, for tasks such as MNIST, where sampled first layer weights can be easy to interpret. In general, it remains a bit mysterious how the algorithm actually samples weights that are better than random weights. So some visualization like in Fig. 1 but for actual optimization tasks would be useful.

6. The paper claims improved "Interpretability" in the Introduction, however, this claim was not supported empirically.

I will be willing to raise my score if the weaknesses are addressed.

**Questions:**

1. L213: "Pre-processing failed in 11 of the 72 datasets" - what kind of pre-processing and what exactly means "failed"? Was it applied to both the Adam and proposed approach? Are those 11 datasets ignored in Figure 3? Were the Adam hyperparameters (learning rate, weight decay, etc.) tuned?

2. In Algorithm 1, arg min L is a linear optimization problem. Is it solved using some kind of least-square in a closed form/gradient-based way? Does this step dominate time complexity in large experiments (ResNet, VGG, Xception) in Fig. 5 right?

**Limitations:**

Limitations are discussed, but not all (e.g. see Weakness 3 about scalability above).

**I updated the rating from 4 to 6 based on the author response and other reviews**

---

> ### Author Rebuttal · Authors · 2023-08-09
>
> **Regarding weaknesses:**
>  * [W1] In both papers, "Gradients without Backpropagation" and "Learning by Directional Gradient Descent", the authors propose methods to compute the gradient of the network with respect to its parameters (weights and biases). In contrast, we propose to choose data point pairs that are close to regions with steep gradients *of the target function* with respect to the input. We choose this so that the corresponding network activation functions $\phi(\langle w,x\rangle-b)$ are "placed" close to these steep slopes. We hope the new illustration in the general answer to all reviewers clarifies this. We still added a brief discussion of the two papers in the related work section of the revised manuscript.
>  * [W2] In Section 4.1, for random features, we do indeed initialize randomly all layers except the last one. Then, we use a least squares solver to find the weights and biases of this last layer, exactly in the same way as for our sampled network.
>     We did not also add this comparison in Figures 3 and 5 (corresponding to Section 4.2 "OpenML" and Section 4.4 "Transfer learning"). In our experience, random feature models only perform comparatively well for layers with much more neurons - but we would be happy to extend the experiments to demonstrate it.
>  * [W3] The scaling with width may be clarified with our complexity analysis (Section F in the supplemental material). Essentially, the *minimum* of the (a) width of the last layer and (b) number of training data points determines how much work the linear solver for the last layer must do; it scales cubically with this number. Note that this is the time complexity *until convergence* for our sampled network, which usually is very hard to obtain in general for methods like ADAM. The sampling of the hidden layer weights is negligible for the computation time. It is true that we also compare a CPU-based research code against an efficient implementation running on the GPU. We also want to highlight that in Figure 5, our approach outperforms the Adam optimizer on the test set already at around 1000 neurons, where our approach is still around ten times faster. It is not clear how long the ADAM optimizer would have to run to reach the same test accuracy.
>  * [W4] The function value differences are indeed constant (equal one) if the classes are one-hot encoded. However, for data pairs inside the same class, the difference is zero. The purpose of the term thus is that only pairs that are in different classes are selected. Furthermore, points that are closer to the decision boundary are selected with higher probability, because they correspond to different classes, but have smaller distances - and the distances are included in the denominator (Equation 2). Figure 1 in the answer to all reviewers should visualize this.
>  * [W5] We added two figures related to the interpretability of the weights in sampled networks to the answer to all reviewers. The examples are constructed on actual tasks (one classification task in two dimensions, one classification using MNIST).
>  * [W6] We hope that the visualization in the answer to all reviewers helps to emphasize how the weights and biases of networks can now be interpreted better. In fact, in the proof of Theorem 1, we show that any neural network can be transformed into a sampled network (with identical values on the input space, but potentially not away from it). This means the sampling method may be useful to interpret even other neural networks that were not obtained with our algorithm, as we can take a trained network as input and output a network with weights and biases of the form given by Definition 1. Once this is done, we can use the information to interpret which datapoints in the training set have been essential to create the neural network which was trained by ADAM.
>
> **Regarding questions:**
>  * [Q1] The datasets from OpenML must be pre-processed before they can be used by standard feed forward neural networks (they contain nominal variables and missing values). The pre-processing steps are listed in the supplemental material (lines 496-499). This pre-processing was applied before we construct the networks, the same way for sampling and Adam training. The datasets where pre-processing "failed" were excluded from all evaluations. We re-ran the code after the submission period ended, and it seems that there may have been an issue with the openml package downloading the data. With a newer version, all 72 datasets work correctly now (no change to our code was needed, just an update to the package). The results of all missing datasets are now included in the new manuscript, they do not change the results shown in Figure 3 significantly.
>
>     We did not modify the standard hyperparameters of Adam, and except for neural architecture search (number of layers), we did not perform hyperparameter search for the OpenML experiment. This hyperparameter search would have skewed the time for the Adam algorithm even more in favor of sampling - as there is just a single hyperparameter to tune for sampling (regularization of the least-squares solve), as compared to many for Adam. Of course, the accuracy for Adam may have improved.
>
>  * [Q2] Indeed, we solve it using the closed form of a least-squares optimization problem (Tikhonov regularization), which is the key source to the scaling. For the scaling factors of the network, we have also included a time and space complexity analysis in the supplemental material, Table 4 (p. 25). Note that the scaling in Figure 5 is due to the increasing, large widths of the networks, not the number of datapoints.

---

> > ### Comment · Reviewer_KCHD · 2023-08-10
> >
> > Thank you for addressing the concerns in detail. In particular, I found the random features experiments [W2] and the new visualizations for 2d and MNIST [W5] very useful and convincing. I'm generally satisfied with the response and will raise my score accordingly (once this option becomes available to reviewers).
> > One minor note is that there are several recent works studying the idea of learning a representation of neural network weights to sample new  weights from the latent distribution, e.g. [A, B]. It would be useful to see some discussion in the Related Work w.r.t. this kind of papers.
> >
> > [A] Hyper-Representations as Generative Models: Sampling Unseen Neural Network Weights, NeurIPS 2022
> >
> > [B] Learning to Learn with Generative Models of Neural Network Checkpoints, 2022

---

> > > ### Author Response · Authors · 2023-08-11
> > >
> > > Thank you! We will cite [A,B] (and potentially others), and discuss learning weights from existing models in the related work section. It may be interesting to consider our sampled weights as training set in this context.

---

### Official Review · Reviewer_ReAz · 2023-07-19

**Soundness:** 2 fair
**Presentation:** 2 fair
**Contribution:** 2 fair
**Rating:** 4
**Confidence:** 3

**Summary:**

This study proposes a sampling learning method for deep ReLU networks. The proposed distribution is data dependent and the sampled network is shown to have universality.

**Strengths:**

The idea of sampling parameters is well-investigated, for example, such as Bayesian NNs, sampling-based dimension reduction for kernel methods, random Fourier features, mean-field theory and Langevin dynamics to mimic SGD learning dynamics, and ridgelet transform for Barron-type integral representation theory. One of the major shortcomings of these methods is that these theories are often limited to shallow networks. This is because the math behind these algorithms is an integral representation of a neural network, and it is essentially a model of a single hidden layer with infinite units. Despite this difficulty, this study has developed and proposed data-dependent proposal parameter distributions for multiple layers.

**Weaknesses:**

However, I could not figure out if the proposed distribution, Eq.2, is well-defined. Since $\Phi^{(l-1)}$ is a piecewise linear map, the graph of Eq.2 may have (1) a constant direction, and (2) line singularities as $|x_1 - x_2| \to 0$. These characteristics suggest that the proposed function is not generally *integrable*, thus the well-definedness is not trivial. Nevertheless, theorems are proved without regularity conditions, so I consider the theory as incomplete.

Additionally, the design of experiments are not consistent to the theory, since (1) Figure 2 draws reference lines $m^{-1/2}$ and $m^{-1}$, but there is no guarantee that the proposed algorithm converges at these rate, and (2) Section 4.3 deals with neural operators, but the architecture is not considered with theory.

Furthermore, Section 2 “Related work” is rather a compressed list of related works than literature overview since it lacks reviewing on what problems remain open in the past studies and how the authors addressed them.
Several closely related works are omitted, for example, such as
- attempts to use (Quasi) Monte Carlo computation of integral transforms (that describe data-dependent parameter distributions):
  - https://arxiv.org/abs/1902.00648
  - https://jmlr.csail.mit.edu/papers/volume22/20-1300/20-1300.pdf
  - https://jmlr.org/papers/volume18/15-178/15-178.pdf
- strong lottery ticket hypothesis (particularly the edge-pop algorithm and its universality):
  - https://arxiv.org/pdf/2111.11146.pdf
- and representer theorem for deep ReLU nets:
  - https://jmlr.org/papers/v20/18-418.html


**Questions:**

Please refer to the weakness section

---

> ### Author Rebuttal · Authors · 2023-08-09
>
> **Regarding the sampling distribution:** Equation 2 uses both the map $\Phi^{(l-1)}$ (ReLU) in the denominator, and the target function values $f(x)$ in the nominator. Based on this review, we refined the definition of our probability density and now assume the following:
>  * Compactness of the data domain. This is already stated in the paper, and in the supplemental material, Definition 3.
>  * Non-constant, Lipschitz-continuous target function $f$. This was not stated before, and we are thankful to the reviewer for pointing it out. If the function $f$ is constant on $\Phi^{(l-1)}(\mathcal{X})$ or the map $\Phi$ maps all points to a single point, we define $p$ to be uniform.
>  * Regularization with a small constant $\epsilon>0$ of the denominator for every hidden layers except the first one. We already state this (for all layers) in Algorithm 1, but did not do so in the definition of the probability density. In the numerical experiments, we used $\epsilon=10^{-10}$.
>
> The new definition of the density is now the following (slightly adapted because not all LaTeX features are available here):
>
> **Definition 2.** Let $\mathcal{X}$ be the compact input space, and let $f$ be a Lipschitz-continuous, non-constant function on $\mathcal{X}$. Let $\mathcal{Y} = f(\mathcal{X})$ be given function values, and let $P$ be a continuous probability distribution defined through its conditional distribution with density $p$, for $l=1,2,\dots,L$ and $x^{(1),0}$, $x^{(2),0}$ $\in\mathcal{X}$, setting $x^{(1),l-1}$,  $x^{(2),l-1} = \Phi^{(l-1)}(x^{(1),0}), \Phi^{(l-1)}(x^{(2),0})$, where $p(x^{(1),0}, x^{(2),0}| \{W_{j}, b_{j}\}, j=1,\dots,l-1)$ is proportional
> to
> $ | f(x^{(2),0}) - f(x^{(1),0})| / \text{max}\lbrace|  x^{(2),l-1}- x^{(1),l-1} |, \epsilon\rbrace,$ when $x^{(1),l-1}\neq  x^{(2),l-1}$ and $0$ otherwise.
>
>   Here, $\Phi^{(l-1)}(\cdot)$ is the sampled network up to layer $l-1$ with parameters {${W_{j}, b_{j}}$}, $j=1,\dots,l-1$, $\mathcal{X}^{l-1}= \Phi^{(l-1)}(\mathcal{X})$, and $\epsilon$ is a regularization constant that we set to zero for $l=1$ and larger than zero for $l>1$.
>   If $f$ is constant or $p$ above would be zero a.e., we define $p$ to be the uniform distribution over $\mathcal{X}\times\mathcal{X}$.
>
> With these additions, most of them being incorporated in the experimental part already, we can now guarantee the proposed density induces a proper distribution. By leaving the conditional density for the first hidden layer unregularized, also all the theory, including Theorem 3, is now consistent.
>
> **Regarding the convergence rate:** we prove (Theorem 2) that there exist sampled networks that achieve convergence rates $m^{-1/2}$, which is why we added the line in the plots for the experiments. Indeed, it is not clear if the specific sampling probability leads to this convergence rate, we only demonstrate that it does so empirically. The architectures we use in Section 4.3 can be mostly considered pre-processing techniques for the sampled networks we study analytically. For example, mapping the target function to Fourier space before constructing a sampled network does not change the theory surrounding it, just the target values. Of course, there are still a lot of open questions (beyond the scope of this paper), e.g., convergence in Sobolev spaces of the sampled networks. The third paper cited by the reviewer may help here.
>
> **Regarding related work:** We greatly appreciate the pointers to additional literature. We comment on them below, and also incorporated them together with a more detailed review in the related work section of the manuscript.
>  * Monte-Carlo / kernels:
>    * Paper1: We construct an inverse mapping from the given target function to the distribution in parameter space of (deep) networks. Even though we currently cannot prove the same exponential convergence rates, our sampling algorithm works for networks with more than one layer. We also introduce a duality between training data and network parameters, which makes the sampled parameters easily interpretable.
>    * Paper2: The ridgelet prior is constructed with normally distributed weights in the first hidden layer, so it "does not require access to any part of the dataset" (quote). Thus the prior does not utilize the information to construct weights, as we do here. The analysis in the paper offers a path toward Bayesian framework for our sampling procedure.
>    * Paper3: There is a strong relation between random features and  kernel functions. The cited paper covers many examples and provides a unifying view on weights for kernel quadrature and random feature approximation of functions. Our work differs in that we do not start with a kernel and decompose it into random features, but we start with a practical and interpretable construction of random features and then discuss their approximation properties. Exploring the kernel (and related RKHS / Bayesian framework) that corresponds to "our" sampled features is an exciting future work that we already started to pursue.
>  * Lottery ticket: The strong lottery ticket hypothesis, where the "winning" subnetworks are not trained but selected from a randomly initialized starting network, is similar to our approach of randomly choosing data pairs and then selecting highly probable ones for weights and biases. The two approaches are still not easily comparable: in the edge-pop algorithm, iterative gradient-based updates of the weight scores are required, and the remaining weights after pruning cannot be interpreted as easily as they can with our algorithm (see our general answer).
>  *  Deep ReLU nets: The paper states that "Designing an algorithm that can effectively deal with this issue [of selecting proper spline points] will be a very valuable contribution to the field." Our sampling procedure may offer a path toward such an algorithm. Even though we cannot ensure that we only find optimal spline points, at least we offer a solution to constructing useful ones based on given data.

---

### Official Review · Reviewer_5eGE · 2023-07-19

**Soundness:** 3 good
**Presentation:** 4 excellent
**Contribution:** 3 good
**Rating:** 6
**Confidence:** 3

**Summary:**

This paper presents an approach to training deep neural networks by introducing a probability distribution for weights and biases, which significantly reduces the necessity for iterative optimization or gradient computations. The proposed sampling scheme is data-driven, factoring in the input and output training data to sample both shallow and deep networks. The paper demonstrates the universality of the constructed networks as well as their invariance to rigid transformations and scaling of the input data. The robustness and speed of the method are shown through various test trials, including a classification benchmark from OpenML, sampling of neural operators to represent maps in function spaces, and transfer learning using well-known architectures. Overall, this approach provides a valuable direction in neural network training, promoting the efficiency of training and the interpretability of the model.

**Strengths:**

1.  This is a well-written paper with clear descriptions and detailed formula derivations.
2.  The data-driven sampling scheme demonstrated in this paper addresses several challenges posed by random feature models compared to iterative optimization methods in supervised learning. Numerical experiments highlight its superiority in training time, accuracy, and interpretability against the ADAM optimizer. Further, its application in transfer learning indicates its potential for broader tasks.

**Weaknesses:**

1. The experiments in the paper only compare this method with iterative optimization methods, specifically ADAM. However, it lacks comparative experiments with related non-iterative training methods, thus not fully showcasing its potential advantages in non-iterative training tasks.
2. There is a notable lack of analysis regarding the convergence rate within the paper.
3. The technique of using data pairs to build model weights is akin to the approach by Galanis et al.[21]. The paper, though, does not provide a sufficient discussion about this similarity, which could potentially undersell the novelty of the methodology.

**Questions:**

1. In Algorithm 1, the L2 Loss function is consistently used. Can this method be adapted to accommodate other types of loss functions?
2. Could you elucidate on the process used to select the data pairs?
3. In the paper, you mention that this method can serve as a good starting point for fine-tuning. However, in Figure 5, even though the initial accuracy of the Sampling method surpasses that of ADAM, the test accuracy after fine-tuning falls short compared to ADAM + Fine-tuning. Can you offer an explanation for this outcome?


**Limitations:**

This paper adequately discusses the limitations, primarily focusing on the following aspects: the sampling strategy is not well-suited for convolutional or transformer networks, the method faces challenges in handling implicit problems, and a theoretical analysis of convergence rates is not provided.

---

> ### Author Rebuttal · Authors · 2023-08-09
>
> Regarding weaknesses:
>  * [W1] Indeed, comparisons to non-iterative methods are still missing, e.g. particle-based or simulated annealing. We expect that these methods sometimes may lead to more accurate solutions (if they find a global solution that we did not), but are probably still orders of magnitude slower - because the time complexity of our method is the same as solving a single linear problem (see our complexity analysis in Section F of the supplemental material).
>  * [W2] Theorem 2 states convergence rates with respect to functions in the Barron space. There still remains questions relating specific sampling distribution to convergence rates. This also reflects the state of research around neural networks and how its often limited to very few results in terms of convergence (except for strong assumptions such as convexity of the loss), and Theorem 2 builds upon the current existing theory on neural networks approximating Barron spaces. However, the theory behind random feature methods are more rich in this instance, and we aim to connect more theory between the networks we propose and random feature methods theory, to provide stronger and probabilistic convergence rates.
>  * [W3] Galaris et al. [21] propose to use the data to find the direction and center of the weights for the sigmoid activation function in a shallow neural network, applied to numerical bifurcation analysis of PDEs from a certain type of simulations. We greatly extend their initial work, which we now also emphasize more in the manuscript:
>
>    *  We define a non-uniform probability density over the data pairs, which forces weights to be close to steep gradients of the target function.
>    *  We extend the idea to different activation functions, by adding scaling factors (the square of the norm) and change the bias. We show that these changes are essential in theory and experiments, particularly for tanh activation functions.
>    *  We extend the construction to deep neural networks.
>    *  Our work brings the key insight from Galaris et al. into a broader machine learning framework, with all the added theoretical investigation (including convergence), to ensure we do not lose anything by enforcing the weights and biases to a restrictive space.
>
> Regarding questions:
>  * [Q1] Yes - please see our general answer to this (several reviewers asked this question).
>  * [Q2] We hope the figures in the general answer to all reviewers also help to illustrate how we choose pairs.
>
>     In theory, we start by sampling the weights of the first hidden layer by sampling from the joint distribution over the square of the dataset, with density given by Equation 2 of the paper. After we have sampled a pair of points for each neuron in the first hidden layer, we construct the weight according to Definition 1. If we have more than one hidden layer, we pass the whole dataset through the first hidden layer and proceed to sample as before, but from the dataset transformed by the first hidden layer. This continues until we reach the last layer, the coefficients of which we then approximate using least squares (in case of mean squared loss).
>
>     In practice, due to the complexity of computing the density of every pair of points, we first uniformly draw a number of data pairs. This number is only proportional to the data set size, not the square of it. From this set, we then draw again, randomly, a number of pairs equal to the number of desired weights in the current hidden layer, but now with a probability that is proportional to the finite difference computed for each pair (and the corresponding target function values on this pair). This means pairs associated with steeper gradients of the target functions are chosen more likely than pairs in flat regions.
>
>  * [Q3] In the paper, we state, 'The sampled weights also provide a good starting point for the fine-tuning of the entire model.' Here, we mean that even though fine-tuning the entire network after sampling breaks the direct connection of weights and biases to data pairs, the ADAM optimizer can still improve the test loss. Essentially, we demonstrate that sampling provides a good starting point for fine-tuning in weight space, i.e., the sampled weights are not a "bad" local minimum from which it is difficult to escape with ADAM.
>
> In our experiments, regardless of whether the weights of the classification head are sampled or trained, the difference in the test accuracy after retraining the whole network is rather small. Thus, for comparable test accuracy, the computational efficiency of sampling is higher.
>
> Nevertheless, we list possible reasons for the small differences:
>  * (a) Note that for the fine-tuning after Sampling, we use the ADAM optimizer. This suffers from comparatively high variance, mostly from iterative optimization and mini-batching, i.e. stochastic approximation of the local gradient. We believe the differences after fine-tuning are mostly due to this variance and not caused by sampling before.
>  * (b) The accuracy after fine-tuning also changes with the chosen architecture. For ResNet50, 'Adam training + finetuning' yields a slightly higher test accuracy as compared to 'sampling + finetuning'. However, for Xception architecture, we observe that 'sampling  + finetuning' yields a higher test accuracy (which is also the highest overall test accuracy in our experiments).

---

> > ### Comment · Reviewer_5eGE · 2023-08-18
> >
> > Thank you for providing the rebuttal and the additional experiments. I agree with the authors' answers to my questions, but the discussion regarding the comparison of non-iterative methods and convergence rates in the weakness section is still insufficient. Despite this, the paper still presents a promising method, and I will keep my rating of 6.

---

> > > ### Author Response · Authors · 2023-08-21
> > >
> > > Thank you! Technically, experiment 4.1 in the paper already is an empirical comparison to a non-iterative method (random features). Still, we agree that there is more research and discussion possible, both toward convergence rates and non-iterative methods.

---

### Official Review · Reviewer_jKG3 · 2023-07-25

**Soundness:** 3 good
**Presentation:** 3 good
**Contribution:** 3 good
**Rating:** 6
**Confidence:** 4

**Summary:**

This work provides a method to sample weights of a neural network in a data-driven manner, such that expensive iterative gradient calculations and optimization can be avoided. The proposed sampling technique sets weights and biases of fully connected networks with ReLU and tanh activations based on pairs of training data points. This mechanism is used for all but the last layer of the network, which is then trained in a gradient-based manner at the very end. This method constructs neural network predictors orders of magnitude faster than some iterative training techniques, while matching them in performance. Experiments include a classification becnhmark, deep neural operators and a transfer learning setting.

**Strengths:**

- The idea of data-driven sampling of neural networks, as opposed to the random features model, is an interesting and original one. This paper sufficiently demonstrates the ability of their method to compete with iterative training methods while being much more efficient.
- The invariance properties of the sampling scheme remove the need for several data preprocessing techniques, which often require significant domain knowledge to determine.
- The theoretical analysis and experimental verification are sound. Proof sketches provided in the main paper are useful to gain insight into the method.
- Given the computational expense of training neural networks currently, extensions of this work to larger scales would have high impact. I find the demonstration and proof of concept shown in this work to be significant.

**Weaknesses:**

- All the analysis seems to be done using mean squared error as the loss function. Stating whether this can be generalized to other losses and/or providing intuition on how that can be done would be useful.
- The main iterative training technique that the proposed sampling technique is compared to is the Adam optimizer. Results using different optimizers/training techniques would strengthen the claims of the paper.

**Questions:**

- As above, how would the theory and/or experiments change for different loss functions and different optimizers used as comparisons?
- The authors provide a sampling method for fully connected layers and state that architectures like convolutions or transformers cannot be sampled with their method yet. Since convolutions ultimately implement linear transformations, is it possible to leverage that view of the convolution operator to extend this sampling scheme? If not, what are the concrete challenges to be solved for extending such a method to convolutional networks or transformers?
- Similarly, what are the challenges to constructing sampled networks for unsupervised or self-supervised tasks?
- The method intuitively provides greater interpretability since the mechanism of sampling weights is given. Do the authors have any thoughts on how this may help compute things like influence functions to answer questions like "what is the influence of a given training point on a given prediction"?

**Limitations:**

Authors clearly and satisfactorily state the limitations and impact of their work.

---

> ### Author Rebuttal · Authors · 2023-08-09
>
> Weaknesses:
>
>  * [W1] Please see our general answer on the topic of choosing different loss functions (multiple reviewers asked for this).
>  * [W2] Different optimizers other than Adam should not drastically change the results in our chosen experiments, as long as they are iterative and gradient based. It may be that for some of the classification problems in Section 4, other optimizers like stochastic gradient descent or particle based methods result in slightly better results compared to Adam, but it is unlikely that they are much faster, given the highly optimized implementations in TensorFlow.
>
>  Questions:
>
>  * [Q1] See W1.
>  * [Q2] Regarding convolutional network sampling: We have tried to extend sampling to convolution kernels, but as of now, it seems that sampling kernels is fundamentally different than sampling weights in a feed-forward network. One of the main challenges is that convolution kernels do not have a one-to-one correspondence to individual data points, so it is not easy to extend the idea of "one data pair per weight" into "one pair of images per convolution kernel". If we would use all possible sub-images of each image pair to determine good kernels, this would quickly lead to an unmanageable amount of kernel pairs. Remark 3 in (https://jmlr.csail.mit.edu/papers/volume22/20-1300/20-1300.pdf) (a paper cited by reviewer ReAz) contains a brief discussion how to convert convolutional networks to feed-forward networks, maybe this can be used as a path forward.
>
>  Regarding transformers: A crude simplification of a transformer block is to apply an embedding to each token in a sequence, in such a way that the covariance matrix of the embedded sequence helps in the prediction of the output sequence. It also seems to be important to stack transformer blocks several times. At this point, the main challenge to extend sampling to transformers is how one would sample good token embedding maps without already knowing the embedding or at least the covariance (unsupervised setting). It certainly is possible to sample a transformer's last layer(s), much like our transfer learning experiment with the convolution backbones (Section 4.4).
>
>  * [Q3] Regarding unsupervised, self-supervised tasks: The main challenge for our sampled networks in unsupervised tasks is that we require the target function values to fit the coefficients of the last layer. Still, the main benefit of our sampling of weights and biases, even without fitting the last layer, is to create a "good" set of basis functions on the data. This set can then be used in downstream tasks, including unsupervised learning, even if the data pairs are "just" chosen uniformly at random (i.e., not taking into account the target function values). A direction we are currently exploring is the solution of partial differential equations, where the solution is unknown (unsupervised setting), but "good" ansatz functions can help to solve the equation easily. Self-supervised tasks may similarly benefit from sampling. One could sample a set of weights/biases without knowledge of the target function (by sampling uniformly over pairs), solve the last layer with gradient descent, and then refine the sampled weights once a new approximation of the function is available.
>
>  * [Q4] Regarding interpretability: The direct connection of weights and biases to data pairs is indeed what we think helps most to interpret the network "internals". This connection answers the question "What is the influence of certain pairs of training data points on all predictions?". Given, individual predictions, as asked by the reviewer, are more difficult to interpret because individual predictions often need multiple activation functions to be "active" (meaning non-zero, in the case of ReLU activation). A simple answer may be: if the training point is part of the "active" ReLUs weights, it is "important", otherwise not. How important exactly should be determined by the magnitude of the connected weight, i.e., the gradient of the ReLU function - which, also using our algorithm, means "how close the training point is to its paired point". For classification problems, this means "how close is the training point to the decision boundary", because only data pairs with differing classes are considered (otherwise the probability of choosing them is zero).

---

> > ### Comment · Reviewer_jKG3 · 2023-08-21
> >
> > Thank you to the authors for their clarifications and explanations regarding my questions. I will keep my score of 6.

---

### Official Review · Reviewer_LFwn · 2023-07-27

**Soundness:** 2 fair
**Presentation:** 1 poor
**Contribution:** 2 fair
**Rating:** 6
**Confidence:** 1

**Summary:**

This work presents a method for sampling the parameters of a deep neural network, including the final task-specific layer. In contrast to prior work that leverages Bayesian deep learning or generative models to learn the sampling distribution, this work presents a sampling algorithm that defines a data-dependent sampling distribution, so no gradient descent is used to train any parameters.

This sampling algorithm is compared with conventionally trained networks (with Adam) on a number of different tasks, where the sampled networks are shown to perform on par or somewhat better than the trained counterparts. Importantly, the sampling procedure faster than iterative training. Currently, this sampling algorithm is restricted to fully connected neural networks.

**Strengths:**

Being familiar with the (sometimes fraught) work on Bayesian neural networks, I appreciate that this work designs an algorithm for effective sampling of neural network weights that is not computationally intractable. This is a new perspective to me, and I found the theoretical results, in particular theorems 1 & 2 to be quite useful in establishing that the sampling algorithm is sound.

I found the empirical results to be interesting, especially the transfer learning / CIFAR-10 experiment that showed comparable test-time performance to an Adam trained network while being far more efficient to "train".

**Weaknesses:**

I'll start by saying that I have very unfamiliar with this line of work, though work in learning to sample neural network parameters is relevant to me. I am quite certain that the main novelty of this paper was missed by me.

That being said, I found this paper to be impenetrable even at a high level. Any confusion I have with the sampling distribution for example, is not alleviated by any satisfactory explanation in the text. I think the only people who will be able to digest this paper in its current form are those completely familiar with this line of work.

**Questions:**

I'm curious if there is any connection to NTK or to NNGP models here? Both employ a similar property of converging on a solution as the width of the network tends to infinity -- and are data dependent, though in a different way.

**Limitations:**

I think the limitations were adequately addressed, as this work applies to small neural networks there shouldn't be any concerns.

---

> ### Author Rebuttal · Authors · 2023-08-09
>
> This high-level explanation of the paper may help: instead of training all parameters of hidden layers with gradient-based methods, we construct each combination of weight and bias using pairs of data points from the training set, i.e., $(x_1,x_2)$. As there are usually many more pairs of points in a dataset than we need for the number of neurons (each associated with a weight vector and bias value), we assign probabilities to pairs. The probability for a pair to be chosen is proportional to the finite difference $|f(x_2)-f(x_1)|/|x_2-x_1|$, where $f$ is the function we want to approximate. The higher the absolute value of the finite difference, the more likely we pick that data pair for a weight in the network. We use finite differences so that weights are more likely distributed around high gradients of the target function than in flat regions.
> In the context of classification and using tanh as an activation function, an informal explanation behind the theory is that we pick two points $x_1$ and $x_2$ for each neuron, where the two points belong to different classes. In addition, we try to pick them as close to the decision boundary as possible. Then, by letting the bias be set as described in the paper, we end up assigning positive values to points that are closer to $x_1$ and negative values to points closer to $x_2$. Then, the last set of weights can use this information to perform the final classification, as the decision boundary separates the two points.
>
> Regarding NNGP/NTK:
> There certainly is a relation, but not an immediate one. Our sampling distribution is one of the differences from the models considered in the NNGP/NTK literature. Classically, the weights of a neural network are initialized with a normal distribution. In our case, we use the distribution from Equation 2. Yet, as we sample parameters i.i.d., we can still consider infinitely wide sampled networks and apply the central limit theorem to get a process similar to NNGP.
> The NTK describes the training dynamics of a network under gradient descent. Sampled networks do not use gradient optimization for hidden layers, so we cannot define such a kernel evolution. However, one could consider only the last layer of a sampled network and assume gradient descent optimization for its weights. This would make it possible to recover a variation of NTK.

---

> > ### Comment · Reviewer_LFwn · 2023-08-15
> >
> > I appreciate the additional summary and feedback by the authors. I also found the additional experiments (particularly figure 1) to be illuminating with regard to the advantages of the sampling distribution over other approaches. I am happy with the authors responses, as it cleared up multiple pain points for me, such as the reasons behind the limitations with sampling convolutional networks. I will adjust my score accordingly from a 5 to a 6.

---

> > > ### Author Response · Authors · 2023-08-16
> > >
> > > Thank you very much!

---

### Author Rebuttal · Authors · 2023-08-09

We very much appreciate all the constructive criticism, feedback, and suggestions. We replied to every review individually. Several questions were raised on (a) the loss function we use for the last layer, and (b) the interpretability of the sampled networks, so we would like to comment on these here. We added two figures in the attached PDF file, which we refer to in the text.

Regarding different loss functions for the final layer:
We only use mean squared error as the loss function because there is a closed-form solution to the linear system in the final layer (using the least squares method, with Tikhonov regularization). This leads to very fast approximation and clear time and space complexity of the algorithm (see Section F in the supplemental material).
If approximation time is not a concern, loss functions used in standard neural network training can also be used - for example, cross-entropy loss for classification tasks. If no closed-form solution is available, the last layer must be trained iteratively. This should still be much faster than training the entire network iteratively.
According to Theorem 1, any loss function based on the $L_p$ norm, for $1\leq p \leq \infty$ is admissible. This covers most of the norms, and therefore losses, that are used to study universal approximations and convergence of neural networks.

Regarding the interpretability of sampled networks:
We argue that sampled networks are inherently more interpretable than iteratively trained ones. This is because we associate a pair of data points in the training set with every weight and bias pair in the network. Note that the pairs are not assigned after training; the network parameters are *constructed* using these data pairs. This allows us to interpret what part of the data domain the network is using most (cf. Figure 1, experiment A), and which data pairs are most important for predictions (cf. Figure 2, experiment B).
In experiment A, we first sample $10^6$ data points in $[0,1]^2$. From those, we sample $10^6$ data pairs with uniform distribution (as suggested in Galaris et al., [21]) and separately with our proposed distribution, using the class as target function values  $(A=1,B=0)$. Finally, we plot the density of all points that were sampled (Figure 1, center and right plots). Obviously, the uniform distribution does not take into account the classification task. In contrast, our distribution mostly concentrates the data pairs around the decision boundaries.
In experiment B, we construct three very small networks with five neurons in one hidden layer each and train them on the MNIST image data set (50,000 training images, ten classes, one-hot encoded). All of the networks are classical feed-forward networks, so the input images (dimension $28\times 28$ pixels) are first flattend to vectors of length 784. The last layer is always constructed using least squares (Tikhonov regularization). In Figure~2, we illustrate the five hidden weights for each neuron by reshaping them from $784$ back to a $28\times 28$ pixel image again. The first network uses random features (weights distributed in a standard normal distribution, biases uniformly in $[0,1]$). The second network uses uniform sampling of pairs, and the third network uses our proposed distribution. Classification accuracy on the test set (10,000 images) is 24% (random features), 43% (uniform), and 47% (ours), respectively. Note that the goal of this experiment is not to achieve high accuracy - the networks have only five neurons each in the hidden layer - but to highlight the differences in interpretability. For random features, the sampled weights are hard to interpret. For the second and third network (second and third row in the figure), the weights are directly computed from the data points. With this knowledge, it is easy to understand why, for example, classes 2 and 4 are poorly classified: none of them are included in the weights. Also, the third weight of the uniformly sampled network does not add a lot of information, because it is constructed from a pair with both points in the same class.
We would also like to point out that the proof of Theorem 1 is constructive, which means we provide a method to convert any given network to a sampled network-effectively by re-constructing data pairs the weights and biases are associated to. This means even networks trained iteratively could now be interpreted using our construction (as long as the training data set contains a large set of points).

---

### Decision · Program_Chairs · 2023-09-21

**Decision:**

Accept (poster)

**Comment:**

The paper proposes a sampling strategy for weights and biases of neural networks using a Bayesian approach. The method is data-driven and does not require further gradient descent steps for optimization. The authors did an excellent work addressing the reviewers' concerns and the reviewers increase their scores post-rebuttal. However, Reviewer ReAz is still concerned with the validity of the theoretical results. It is acceptable if the theory is used only as a guide for developing the technique. However, the authors must update the proofs and fix the bugs before the final version. Also, there are some references that do not follow the `last name (year)` format, which need to be fixed.